



# Warm Greenland during the last interglacial: the role of regional changes in sea ice cover

Niklaus Merz[1,2], Andreas Born[1,2], Christoph C. Raible[1,2], and Thomas F. Stocker[1,2]

[1]Climate and Environmental Physics, University of Bern, Bern, Switzerland
[2]Oeschger Centre for Climate Change Research, University of Bern, Bern, Switzerland

*Correspondence to:* N. Merz (merz@climate.unibe.ch)

**Abstract.** The last interglacial, the Eemian, is characterized by warmer than present conditions at high latitudes and is therefore often considered as a possible analogue for the climate in the near future. Simulations of Eemian surface air temperatures (SAT) in the Northern Hemisphere, however, show large variations between different climate models and it has been hypothesized that this model spread relates to diverse representations of the Eemian sea ice cover. Here we use versions 3 and 4 of the

Community Climate System Model (CCSM3 and CCSM4), to highlight the crucial role of sea ice and sea surface temperatures during the Eemian, in particular for SAT in the North Atlantic sector and in Greenland. A substantial reduction in sea ice cover results in an amplified atmospheric warming and, thus, a better agreement with Eemian proxy records. Sensitivity experiments with idealized lower boundary conditions reveal that warming over Greenland is mostly due to a sea ice retreat in the Nordic Seas. In contrast, sea ice changes in the Labrador Sea have a limited local impact. Changes in sea ice cover in either region

are transferred to the overlying atmosphere through anomalous surface energy fluxes. The large-scale warming simulated for the sea ice retreat in the Nordic Seas further relates to anomalous heat advection. Diabatic processes play a secondary role, yet distinct changes in the hydrological cycle are possible. Our results imply that temperature and accumulation records from Greenland ice cores are sensitive to sea ice changes in the Nordic Seas but insensitive to sea ice changes in the Labrador Sea. Moreover, our simulations suggest that the uncertainty in the Eemian sea ice cover accounts for 1.6°C of the Eemian warming

at the NEEM ice core site. The estimated Eemian warming of 5°C above present-day based on the NEEM $\delta^{15}$N record can be reconstructed by the CCSM4 model for the scenario of a substantial sea ice retreat in the Nordic Seas combined with a reduced Greenland ice sheet.

## 1 Introduction

The last interglacial (ca. 130–116 ka) also known as the Eemian is often regarded as a possible analogue for future climate as

it stands for the most recent period in the past characterized by a warmer than present climate in many regions on the globe, particularly at high latitudes. In contrast to the future warming controlled by rising greenhouse gas (GHG) concentrations, the Eemian climate was largely driven by anomalous orbital forcing that led to enhanced summer insolation at high latitudes. A warmer than present Eemian climate has been observed in various proxy records (CAPE Last Interglacial Project Members, 2006; Turney and Jones, 2010; Capron et al., 2014) and also simulated in climate model experiments (e.g., Nikolova et al.,



2013; Lunt et al., 2013; Merz et al., 2014a). However, model-data comparison studies revealed rather poor agreement between simulations and data, with climate models generally underestimating the magnitude of warming inferred from proxy records (Lunt et al., 2013; Otto-Bliesner et al., 2013; Capron et al., 2014). In the Northern Hemisphere (NH), the models reasonably capture the distinct warming in summer which is a direct result of increased summer insolation. In contrast, the models mostly

fail to simulate a clear warming for winter, but rather generate lower temperatures due to the decrease in winter insolation (Lunt et al., 2013). This disagreement either originates from missing feedbacks in the model simulations and/or misconceptions in the interpretation of the proxy records. The reasonable coherence among various Eemian proxy records, however, strongly suggests model deficiencies to be the major problem.

Besides the lack of agreement of climate models with proxy signals, the simulated Eemian warming can further substantially

vary among different fully-coupled climate models themselves, in particular in the NH mid- and high latitudes (Lunt et al., 2013; Otto-Bliesner et al., 2013; Nikolova et al., 2013). Those studies hypothesized that model-dependent changes in sea ice are a primary cause for the diverse temperature response, however, without testing the role of sea ice cover in detail. Here we will do so, as we use sea ice and SSTs from two different fully-coupled simulations of the Eemian (Lunt et al., 2013) to force an atmospheric model. In addition, we design a set of idealized sea ice sensitivity experiments embedded in Eemian climate

conditions. More precisely, we investigate the influence of sea ice changes on the temperature in and around Greenland in order to facilitate the interpretation of temperature records from Greenland ice cores. Hence, this study complements work by Merz et al. (2014a, b) who showed that changes in the Greenland ice sheet configuration can lead to distinct Greenland climate signals that are of local rather than large-scale (e.g., hemispheric or global) dimension. Here, we make an effort to show how a reduction in NH sea ice cover can lead to a substantial warming in central Greenland, which is recorded by ice cores such as

NEEM (NEEM community members, 2013), without being necessarily related to a hemispheric-scale temperature anomaly.

The question whether and to which extent the sea ice around Greenland was different during the Eemian compared to the present interglacial is difficult to answer. Firstly, no direct sea ice measurements or sea ice proxies are available for the Eemian. Besides, climate models simulate diverse sea ice covers for the Eemian (e.g., Otto-Bliesner et al., 2013; Nikolova et al., 2013). Moreover, sea surface temperature (SST) proxy records also show a rather complex response in the region of the North

Atlantic basin to the external forcing during the Eemian: near the East Greenland coast marine and terrestrial records indicate summer temperatures that are about 2-3°C higher than the Holocene optimum indicating unfavorable conditions for sea ice (Funder et al., 1998). In contrast, sediment samples from a core southeast of the Fram Strait indicate colder Nordic Seas conditions compared to the Holocene optimum (Van Nieuwenhove et al., 2011).

Furthermore, the peak warming in the Nordic Seas during the Eemian is not in phase with more southerly regions of

the North Atlantic, possibly due to anomalous ocean currents and delayed influx of relatively warm Atlantic water masses (Bauch et al., 2012; Born et al., 2010). Even less is known about sea surface conditions of the Labrador Sea and the Baffin Bay during the Eemian. Various terrestrial records from coastal Baffin Island, however, point at clearly above present temperatures (Axford et al., 2011, and references therein) and therefore imply conditions which suggest a reduced Labrador Sea ice cover.

Although little is known about the precise NH sea ice extent before the modern era, the impact of sea ice on the climate

of the past has been investigated with respective climate model experiments for the Greenland/North Atlantic region. A com-



mon approach are sensitivity experiments where the sea ice concentration (SIC) and SSTs in the ice-containing grid cells vary among a set of simulations with all other boundary conditions held constant. For example, Smith et al. (2003) demonstrated that there are significant changes, primarily in winter, in North Atlantic surface temperature, sea level pressure, and snowfall, when changing from modern to what they assume as minimum/maximum Holocene sea ice coverage. Furthermore,

Li et al. (2005, 2010) showed for glacial conditions that a substantial sea ice retreat in the North Atlantic results in distinct Greenland temperature and accumulation anomalies reflecting observed signals associated with Dansgaard-Oeschger cycles in Greenland ice cores. The majority of NH sea ice sensitivity experiments, however, have been conducted for present and future climate conditions (e.g., Alexander et al., 2004; Higgins and Cassano, 2009; Petoukhov and Semenov, 2010; Deser et al., 2010; Screen et al., 2013). These studies showed that the ongoing reduction in Arctic sea ice has a seasonally diverse im-

pact on the local surface climate (Deser et al., 2010; Screen et al., 2013). Moreover, sea ice changes also affect the large-scale atmospheric circulation (Petoukhov and Semenov, 2010) and the atmospheric modes of variability such as the North Atlantic Oscillation (Alexander et al., 2004; Kvamstø et al., 2004) and thus also can have a significant impact on more distant areas. The atmospheric response to a sea ice retreat is further found to be sensitive to the geographical location of the ice loss (Rinke et al., 2013). Thereby the authors focused on different areas within the Arctic ocean. For this study, however, we concentrate on a

possible sea ice loss in the areas adjacent to Greenland.

The manuscript is structured as follows: Sect. 2 describes the climate model simulations followed by Sect. 3 explaining the design of idealized "sea ice shift" experiments that simulate a sea ice retreat located either west (i.e., in the Labrador Sea/Baffin Bay) or east (i.e., in the Nordic Seas) of Greenland. In Sect. 4 we investigate existing fully-coupled Eemian simulations as well as newly created atmospheric simulations that use simulated Eemian sea ice extent and SSTs as prescribed lower boundary

conditions. These simulations enable us to quantify the contribution of sea ice to the Eemian warming and demonstrate how differences in regional sea ice cover and SSTs can be responsible for a large part of the spread in the simulated Eemian warming found in Lunt et al. (2013), Otto-Bliesner et al. (2013) and Nikolova et al. (2013). In Sect. 5, we analyze the idealized sea ice shift experiments with a focus on changes in surface climate (e.g., surface air temperature (SAT) and precipitation) and their relation to concurrent changes in the atmospheric heat and moisture budget. The results are discussed and interpreted with

respect to possible consequences for Greenland ice core signals in Sect. 6. Finally, a summary is given in Sect. 7.

## 2   Model description and experiments

The study is based on model simulations with versions 3 and 4 of the Community Climate System Model (CCSM) provided by the National Center for Atmospheric Research (NCAR). Both model versions include components for atmosphere, ocean, land and sea ice, which are connected by a coupler exchanging state information and fluxes.

### 30   2.1   CCSM3 simulations

We use four existing fully-coupled simulations generated with CCSM3 (Collins et al., 2006): (i) a pre-industrial control simulation (Merkel et al., 2010) and (ii) 30 years of output at 125 ka from a transient (130–115 ka) orbitally accelerated Eemian



simulation (Varma et al., 2015) both using the low horizontal resolution of T31 in the atmosphere/land and approximately 3° grid spacing in the ocean/sea ice component. Furthermore, we analyze (iii) a pre-industrial control simulation and (iv) an Eemian simulation with perpetual 125 ka forcing both at a resolution of T85 in the atmosphere/land and approximately 1° in the ocean/sea ice (Otto-Bliesner et al., 2013). Hence, we can compute two realizations of the Eemian minus pre-industrial

climate anomaly (denoted as EEM-PI) based on the same CCSM3 model but differing in horizontal resolution.

## 2.2   CCSM4 simulations

Additionally, a set of simulations is generated employing CCSM4 (Gent et al., 2011) with $0.9° \times 1.25°$ resolution in the atmosphere and land surface with prescribed time-varying monthly SSTs and sea ice cover. This CCSM4 setup is termed atmosphere-land only and comprises the Community Atmosphere Model version 4 (CAM4; Neale et al., 2010) and the Com-

munity Land Model version 4 (Oleson et al., 2010) but no dynamic representation of the ocean and sea ice. Besides the benefit of being computationally cost-efficient compared to fully-coupled simulations, this setup is convenient for sea ice sensitivity experiments, as one can simply compute the atmospheric response to any prescribed change in sea ice (and SSTs). As a drawback, these simulations do not allow feedbacks with the ocean and sea ice components. A general model validation of the CCSM4 atmosphere-land-only setup is given by Evans et al. (2013).

In total, we perform 12 simulations with CCSM4 of which the 6 simulations listed in Table 1 build the core of this study whereas the remainder of the simulations will be shortly discussed in Sect. 5.4. Each simulation has a length of 30 years plus a 3-year spin-up phase and the external forcing is held constant throughout the simulation.

### 2.2.1   Eemian and pre-industrial experiments with prescribed SSTs/sea ice

The first set of CCSM4 experiments consists of two pre-industrial simulations with 1850 AD external forcing and two Eemian

simulations with 125 ka external forcing. The atmosphere-land-only setup requires appropriate SST and sea ice fields as input data. We use the output of the respective fully-coupled CCSM3 simulations mentioned above: the CCSM4 $PI_{lowRes}$ and $EEM_{lowRes}$ use output of the pre-industrial and Eemian simulations generated with the $T31 \times 3°$ CCSM3 whereas the CCSM4 $PI_{highRes}$ and $EEM_{highRes}$ use output of the $T85 \times 1°$ CCSM3, respectively. Note that the CCSM4 simulations themselves all use the same horizontal resolution of $0.9° \times 1.25°$; the lowRes/highRes suffixes solely attribute the origin of the lower boundary

conditions.

 With the two pairs of PI and EEM atmosphere-land-only CCSM4 simulations we create equivalents to the existing fully-coupled CCSM3 simulations. Hence, we can compute two realizations of the EEM-PI climate anomaly based on the exact same CCSM4 model but differing in terms of prescribed SSTs and sea ice. Consequently, this setup eliminates uncertainties arising from different model physics and parameterizations at different resolutions (as it is the case in the fully-coupled CCSM3

simulations). This enables a more robust analysis of the impact of sea ice and SSTs.



### 2.2.2 Sea ice sensitivity experiments

A second set of CCSM4 experiments is designed to analyze the atmospheric response to an idealized sea ice retreat in a specific geographical area. As it will be shown in Sect. 4 both the Labrador Sea/Baffin Bay (LabS) and the Nordic Seas (NordS) region are reasonable candidates for a distinct Eemian warming induced by a local sea ice reduction. In order to evaluate the

importance of these two areas separately, we design both the scenario of a sea ice retreat in the LabS area (simulation denoted as $EEM_{LabS}$) and a sea ice retreat in the NordS area (simulation denoted as $EEM_{NordS}$). As shown in Table 1, $EEM_{LabS}$ and $EEM_{NordS}$ are identical to $EEM_{lowRes}$ with the exception of the modified sea ice and SSTs used at the lower boundary, thus being classical sea ice sensitivity experiments embedded in an Eemian background climate.

### 2.3 Definition of climate anomalies

Based on our set of simulations we define a few climate anomalies, which will be frequently used throughout this manuscript (definitions see Table 2). The EEM-PI anomaly simply refers to the change in Eemian climate with respect to pre-industrial. For both the CCSM3 and CCSM4 simulations, two EEM-PI anomalies are possible using either the lowRes or highRes simulations. The difference between the two EEM-PI anomalies themselves is referred to as $EEM\text{-}PI_{diff}$. Moreover, we use the terms LabS-shift and NordS-shift for the comparison of the Eemian experiments including a regional (either in LabS or NordS) shift in

lower boundary conditions compared to their reference.

### 3 A new type of an idealized sea ice sensitivity experiment

Various types of sea ice reduction experiments have been presented in previous studies (e.g., Smith et al., 2003; Deser et al., 2010; Petoukhov and Semenov, 2010). A prominent approach is to implement an observed or simulated minimum sea ice cover (e.g., Smith et al., 2003; Alexander et al., 2004) or an altered sea ice climatology that exhibits a retreated sea ice cover

compared to its reference (e.g., Higgins and Cassano, 2009; Deser et al., 2010). An alternative option is to artificially reduce the SIC in a target region to a certain percentage (e.g., Petoukhov and Semenov, 2010). These experimental designs have in common that they use a repeating seasonal cycle of SICs (and SSTs) and thus are not accounting for inter-annual variability. The absence of inter-annual variability in the ocean/sea ice representation, however, can be a drawback with respect to atmospheric dynamics, e.g., causing a degraded representation of the mid-latitude stormtrack (Raible and Blender, 2004).

To avoid this deficiency and also to be consistent with the pre-industrial and Eemian CCSM4 simulations, which use time-varying SSTs and sea-ice fields (including inter-annual variability), the "sea ice shift" approach is applied (illustrated in Fig. 1). We take the monthly varying lower boundary conditions previously used for CCSM4 $EEM_{lowRes}$ and modify the values in the target region by shifting them along a certain axis. For the $EEM_{LabS}$ simulation we shift all SIC values in the LabS domain northwestward (see Fig. 1a). In technical terms, all values within the solid rectangle in Fig. 1a are replaced point-by-point by

the values within the dashed rectangle. Values in the green shaded area are linearly interpolated to guarantee a smooth transition with the adjacent regions. Similarly, for $EEM_{NordS}$ we shift all SIC values in the NordS domain (dashed rectangle in Fig. 1b)



northwards. As illustrated by the 50% sea ice contour lines in Fig. 1a,b this approach results in a local sea ice retreat in the perturbed (dashed contour) compared to the reference simulation (solid contour). Note that in all cases we only change the sea ice area, whilst the sea ice thickness is fixed at 2 m throughout the Arctic which is the default for CCSM4 simulations with prescribed lower boundary conditions.

A key consideration in all types of sea ice sensitivity experiments is the prescription of corresponding SST changes. For example, grid cells becoming ice-free are exposed to solar radiation and thus local SSTs likely increase compared to the typical freezing point temperature of $-1.8°C$ of an ocean grid cell covered by ice. Vice-versa, the sea ice retreat itself can be caused by a warming of the surface ocean, hence a reduction in SIC is usually accompanied by an increase in SSTs. This strong relationship between SST and SIC in marginal sea ice areas is also found in the input data used for $EEM_{lowRes}$ (solid

lines in Fig. 1c,d) along the transects A→B and C→D cutting through our two target regions. In order to account for this strong link between the sea ice cover and SSTs, we shift the SSTs in the same way as the SICs (see dashed lines in Fig. 1c,d). This approach seems particularly reasonable for the LabS region where we find gradual changes along the transect (Fig. 1c). Hence, the northwestward LabS-shift in $EEM_{LabS}$ can be understood as a warm water inflow into the LabS area (see SSTs in LabS in Fig. 1a compared to Fig. 1b) resulting in a coherent sea ice retreat. In contrast, the situation in the Nordic Seas is more

complex (see Fig. 1d) as the northward shift in SSTs corresponds to a displacement of local ocean currents with a nonparallel orientation to the C→D axis along which we apply the shift. For example, the northward NordS-shift results in a removal of the cold East Greenland current in $EEM_{NordS}$ (see SSTs in NordS in Fig. 1b compared to Fig. 1a). Consequently, our sea ice shift experiments are of idealized nature but, nevertheless, result in SIC and SST anomalies that resemble the simulated EEM-PI changes as will be shown in Sect. 5.

Additionally, we generate a second pair of LabS- and NordS-shift experiments (termed $EEM_{LabS\ ICE}$ and $EEM_{NordS\ ICE}$) for which we only shift the SIC (equivalently to $EEM_{LabS}$ and $EEM_{NordS}$) but not the SSTs. Hence, this second approach avoids a possibly unrealistic warming of the surface ocean but, on the other hand, violates the obvious SST-SIC relationship revealed in Fig. 1c,d. Thus, $EEM_{LabS\ ICE}$ and $EEM_{NordS\ ICE}$ can be understood as experiments providing the lower range in terms of atmospheric response to a prescribed sea ice retreat. A detailed discussion of the atmospheric response to different

experimental designs is presented in Sect. 5.4.

## 4    Simulated Eemian warming: importance of sea ice and SSTs

### 4.1    Atmospheric temperature response in CCSM3 simulations

The starting point of our analysis is the model-intercomparison study by Lunt et al. (2013) which showed that the EEM-PI annual mean atmospheric warming (Fig. 5 therein) strongly varies among different climate models. One particularly striking

finding is the disagreement between two EEM-PI temperature anomalies generated by two simulations with the same climate model (denoted as CCSM3_Bremen and CCSM3_NCAR in Lunt et al. (2013), reproduced here in Fig. 2a,c). The EEM-$PI_{highRes}$ SAT change (Fig. 2a) is based on the high resolution CCSM3 simulations corresponding to CCSM3_NCAR whereas the EEM-$PI_{lowRes}$ SAT change (Fig. 2c) is based on the low resolution CCSM3 simulations corresponding to CCSM3_Bremen.





Hence, the differences between EEM-PI$_{highRes}$ and EEM-PI$_{lowRes}$ are due to different horizontal resolutions and other distinctions in the model setups, e.g., only CCSM3_Bremen includes a dynamic vegetation module. CCSM3 EEM-PI$_{highRes}$ exhibits a distinct warming in the NH high latitudes with the strongest signal occurring in an area including the Arctic, Greenland and the North Atlantic (Fig. 2a). Significant warming but of smaller magnitude is further found in Europe and most of North America.

On the contrary, the CCSM3 EEM-PI$_{lowRes}$ warming is very limited in terms of magnitude and spatial expansion (Fig. 2c). In fact, large areas of the NH experience an annual mean cooling. The difference between the two EEM-PI warming patterns (Fig. 2e) illustrates a stronger warming of EEM-PI$_{highRes}$ than EEM-PI$_{lowRes}$ in almost the entire NH, but most distinctively over the Arctic and the North Atlantic ocean.

The reasons for this remarkable discrepancy of EEM-PI warming among the two pairs of CCSM3 simulations can be
diverse. As the horizontal resolution differs in all components (i.e., atmosphere, land, ocean, sea ice), which themselves all interact with each other, it is not a priori clear where the disparity in the SAT response has its origin. The ocean and sea ice are likely candidates as a too cold North Atlantic with and an excessive NH sea ice cover are well-known model biases in the low resolution version of CCSM3 that are less distinct in higher resolution CCSM3 versions (Yeager et al., 2006). Therefore, we investigate the role of SSTs and the sea ice cover with a set of new model simulations: we use the SSTs and sea ice of
both pre-industrial and both Eemian fully-coupled CCSM3 simulations as boundary conditions for a corresponding set of atmosphere-land-only CCSM4 simulations (see Sect. 2.2 for details on the model setup). With this approach, we test whether the atmospheric model (of CCSM4) driven by respective lower boundary conditions is able to reproduce the different EEM-PI warming identified among the two pairs of fully-coupled CCSM3 simulations.

## 4.2   Atmospheric temperature response in CCSM4 simulations

The similarities of the CCSM4 simulations and their CCSM3 equivalents are remarkable (Fig. 2). The CCSM4 EEM-PI$_{highRes}$ (Fig. 2b) largely exhibits the same distinct high latitude warming as its CCSM3 counterpart (Fig. 2a), and there is also a high agreement for EEM-PI$_{lowRes}$ CCSM4 and CCSM4 (compare Fig. 2c and 2d). Eventually, the CCSM4 EEM-PI$_{diff}$ SAT pattern (Fig. 2f) strongly suggests that large parts of the spread between the two diverse fully-coupled CCSM3 EEM-PI responses (shown in Fig. 2e) originate from differences in SSTs and sea ice. Note that the two pairs of CCSM4 simulations (i.e., PI$_{lowRes}$,
PI$_{highRes}$ and EEM$_{lowRes}$, EEM$_{highRes}$, respectively) use identical experimental setups, so all CCSM4 EEM-PI$_{diff}$ differences (Fig. 2f) necessarily result from differences in the prescribed lower boundary conditions. The strongest impact of the lower boundary conditions is simulated for the area around Greenland and the North Atlantic but also expanding to Europe and parts of continental Asia. In contrast, the influence of the lower boundary conditions on low latitude regions is of smaller magnitude. In the following we will focus on the distinct EEM-PI$_{diff}$ SAT signal in the Greenland/North Atlantic region and analyze in
detail its relation with the underlying sea ice cover and SSTs.

The EEM-PI change in SSTs and sea ice simulated by the highRes and lowRes fully-coupled CCSM3 simulations is shown in Fig. 3. EEM-PI$_{highRes}$ shows a warming of the North Atlantic and a retreat of the sea ice cover in all seasons. In winter (DJF) and spring (MAM), the main reduction in sea ice is confined to the Labrador Sea whereas in summer and autumn the sea ice cover in the Nordic Seas is reduced as well. The strongest increase in SSTs (>4°C anomaly) is found south of Greenland



corresponding to a strengthening of the Atlantic subpolar gyre. A strong subpolar gyre during the Eemian due to less sea ice export from the Arctic is in agreement with previously published results based on two different climate models and marine sediment proxies (Born et al., 2010, 2011).

The EEM-PI$_{lowRes}$ change in SSTs and sea ice (Fig. 3, bottom row) deviates from EEM-PI$_{highRes}$. In fact, the North Atlantic
mostly cools and even the high levels of summer insolation during the Eemian only result in a moderate surface warming in shallow coastal waters. In the Nordic Seas, the summer SSTs even decrease and the sea ice cover expanded during the Eemian compared to the pre-industrial climate throughout the year. Hence, the EEM$_{lowRes}$ simulation seems to strongly respond to the decrease in winter insolation rather than to the increase in summer insolation. A possible reason for the diverging oceanic responses of the lowRes and highRes versions of the fully-coupled CCSM3 to the same Eemian external forcing is likely
connected to model biases. More precisely, even for pre-industrial conditions the two model versions show clear differences in the SST (not shown) and sea ice (solid contours in Fig. 3) climatology. Thereby, the overestimation of the NH sea ice in the lowRes CCSM3 (Yeager et al., 2006) likely generates North Atlantic conditions that prevent an Eemian strengthening of the subpolar gyre in contrast to EEM-PI$_{highRes}$. This is due to the non-linear character of the gyre dynamics and its strong dependence on the background salinity and thus freshwater fluxes linked to sea ice processes (Born and Stocker, 2013).

When using these CCSM3 sea ice and SSTs as prescribed lower boundary conditions for the CCSM4 atmosphere-land-only simulations, the distinct differences in the EEM-PI changes in terms of lower boundary conditions directly translate into similar responses in the CCSM4 atmospheric temperature (compare Fig. 3 and Fig. 4 top and middle row). As expected, the influence of sea ice and SSTs is particularly strong on SAT above oceanic grid cells, e.g., any EEM-PI cooling or warming in SSTs can be identified in the EEM-PI SAT response. For example, in Eemian winters the decreased solar insolation leads
to a widespread atmospheric cooling, but in EEM-PI$_{highRes}$ (Fig. 4, top left) the direct effect of the external forcing on SATs is superimposed in the North Atlantic domain by oceanic changes showing a warming (Fig. 3 top left). Consequently, we find clear differences between the EEM-PI$_{highRes}$ and EEM-PI$_{lowRes}$ warming in both annual mean (Fig. 2) and seasonal mean (Fig. 4) SATs. The strongest seasonal differences in SATs as a result of diverging lower boundary conditions is found for DJF and MAM (see Fig. 4, bottom row). In these two seasons, the EEM-PI$_{diff}$ warming is not restricted to oceanic areas but also
includes substantial changes in Greenland and European SATs. In contrast, the differences in lower boundary conditions hardly lead to a diverse warming outside of the North Atlantic domain during summer.

In summary, we have demonstrated that distinct differences in the simulated Eemian warming based on fully-coupled models are explained by their differences in sea ice and SSTs. The influence of the sea ice cover and the surface ocean on the EEM-PI atmospheric response is particularly strong in the North Atlantic and apparent in all four seasons but especially in winter. In
the following, we focus on winter and analyze the processes that are responsible to transmit changes in sea ice/SSTs to the atmosphere. Furthermore, we study atmospheric transport processes which decide whether and how the additionally available heat in the atmosphere is spatially distributed.



### 4.3 Oceanic heat sources

The distinct EEM-PI$_{diff}$ warming (Fig. 4, bottom row) needs to be understood as an additional Eemian warming caused by the highRes SSTs and sea ice with respect to the lowRes boundary conditions. This effect is unrelated to the direct atmospheric response to the Eemian external forcing (e.g., changes in the orbital parameters). Consequently, the EEM-PI$_{diff}$ warming re-
quires oceanic heat sources, i.e., an increased heat transfer from the surface ocean to the atmosphere. Two types of heat sources are possible: either a warmer surface ocean that warms the overlying atmosphere directly or a reduction in the sea ice cover, which exposes a relatively cold atmosphere to the underlying (warmer) surface ocean. In order to assess these two processes for winter, we compare the DJF EEM-PI$_{diff}$ SST and SIC anomalies with the response of the atmospheric surface energy fluxes (Fig. 5). All surface energy fluxes are defined positive in the upward direction, i.e., a positive flux is warming the overlying
atmosphere.

The comparison of the SST/SIC map (Fig. 5a) with the net surface energy flux response (Qnet, Fig. 5b) reveals that most of the warmer North Atlantic acts as a heat source. The strongest positive Qnet anomaly, however, is confined to the areas of sea ice retreat in the Labrador Sea, the East Greenland current south of Denmark Strait and the northern Nordic Seas. The dominant components of Qnet are the turbulent energy fluxes (sensible and latent heat, Fig. 5c,d) rather than the radiative fluxes. This
is in agreement with previous sea ice sensitivity experiments (e.g., Deser et al., 2010). In fact, the DJF net longwave radiation slightly increases over the warming North Atlantic (not shown) whereas shortwave radiation is mostly absent in the high latitude NH during winter. Note that we omit the shortwave component in the calculation of Qnet (shown in Fig. 5b) because increased downward shortwave radiation resulting from modifications in surface albedo (e.g., by changing an ocean grid cell from ice-covered to ice-free) does not warm the atmosphere directly but warms the ocean, an effect that is suppressed in our
experimental setup where SSTs are prescribed.

The turbulent energy fluxes (Fig. 5c,d) show negative responses in areas adjacent to sea ice loss and therefore adjacent to the regions with the strongest positive energy flux responses. The resulting dipole patterns can be understood by considering that the positive fluxes locally warm the low-level atmosphere and this heat can be transported to areas nearby. The warmer air masses then lose some of their excess heat to the underlying ocean resulting in negative heat fluxes. Hence, the
SSTs would rise in regions with negative flux responses and eventually this would dampen the negative fluxes by reducing the air-ocean temperature difference. However, as SSTs are prescribed in our CCSM4 simulations, this negative feedback is suppressed and consequently the dipoles in turbulent energy flux responses are rather pronounced. Nevertheless, similar dipole features were also identified in fully coupled model simulations (Deser et al., 2010) as well as in atmospheric reanalyses (Screen and Simmonds, 2010) and, thus, are only partly due to our experimental setup.
In summary, the DJF EEM-PI$_{diff}$ differences in terms of SSTs and SICs lead to several distinct oceanic heat source areas in the North Atlantic whereof the areas marked by a sea ice retreat are strongest as indicated by the maxima in (upward) surface energy flux anomalies (Fig. 5b-d). From the analysis so far, however, it is not possible to distinguish the impact of the heat source in the Labrador Sea from the ones in the Nordic Seas. To disentangle the effect of these two regions, we, consequently, use idealized sea ice sensitivity experiments, which simulate either a sea ice retreat in the Labrador Sea or in the Nordic Seas





(see Sect. 2.2.2 and Sect. 3 for a detailed description of the experimental setup). In particular, we are interested which sea ice retreat does not only cause a local atmospheric warming but a widespread temperature signal that extends to Greenland corresponding to the EEM-PI$_{diff}$ winter warming pattern in Fig. 4 (bottom left).

## 5 Atmospheric response to sea ice retreat in Labrador Sea vs. Nordic Seas

The idealized LabS-shift leads to a distinct winter sea ice reduction in the Labrador Sea accompanied by a SST increase of up to 5°C (Fig. 6a). Equivalent to the processes explained in Sect. 4.3, changes in lower boundary conditions act as local heat sources with anomalous surface heat fluxes transporting heat out of the ocean into the overlying atmosphere (Fig. 6b-d). Thereby, the key contribution to the net surface energy flux change (Fig. 6b) is again made by the turbulent energy fluxes (Fig. 6c,d). The positive (upward) net surface energy flux anomaly is strongest directly above the sea ice retreat (Fig. 6a) but also spreads to the Baffin Bay area. The latter is explained by considering that in summer and autumn the sea ice edge area lies in this more northern region (see Fig. 3) and consequently the LabS-shift results in a distinct seasonal sea ice retreat in these more northern areas (not shown). The summer/autumn sea ice reduction also affects the winter heat fluxes as the simulated snow cover accumulated on the Baffin Bay sea ice is highly reduced in EEM$_{LabS}$ compared to the reference simulation where snow can accumulate all year (not shown). As the snow cover also acts as a thermal insulation layer between the warm ocean and the cold atmosphere, similar to sea ice, a thinner snow layer leads to an increase in the local sensible heat flux (Fig. 6c). Furthermore, both turbulent heat fluxes exhibit again the dipole structure with negative flux anomalies in the area west of the Labrador Sea.

Correspondingly, the NordS-shift experiment (Fig. 6e-h) exhibits distinct SIC, SST, and energy flux anomalies in the Nordic Seas. The perturbation results in a sea ice retreat along the East Greenland coast, around Iceland and in the Fram Strait (Fig. 6e). The areas of SIC reduction coherently show an increase in SSTs whereas other areas in the Nordic Seas experience a moderate cooling of the surface ocean as a result of the SST-shift included in EEM$_{NordS}$. In agreement with the previous results, strong positive net surface energy flux anomalies (Fig. 6f) are simulated for all regions with decreasing SIC with sensible and latent heat (Fig. 6g,h) together accounting for most of this energy flux increase. At the same time, a decrease in the energy fluxes is found in areas adjacent to the sea ice reductions building the dipole-structure already observed in EEM-PI$_{diff}$ (Fig. 5b-d) and in the LabS-shift experiment (Fig. 6b-d).

The net surface energy flux response of the LabS- and NordS-shift experiments (Figs. 6b and 6f) confirms that our idealized sea ice shift experiments lead to distinct winter heat sources located either west (LabS) or east (NordS) of Greenland. With regard to the predominantly westerly flow in the NH extra-tropical atmosphere, one intuitively expects that heat released upstream of Greenland (i.e., in the LabS) spreads to Greenland rather than heat released downstream of Greenland (i.e., in the NordS). The simulated SAT response to the two shift-experiments, however, reveals a different picture (Fig. 7): the LabS-shift leads to a surface warming above the Labrador Sea/Baffin Bay area but hardly any warming over the adjacent land masses. Over Greenland, significant warming is limited to the western coastal regions that have direct contact to the heat source in the Labrador Sea (Fig. 7a). In contrast, the SAT response to the NordS-shift (Fig. 7b) reveals an atmospheric surface warming that



substantially extends beyond the heat source area (i.e., the positive Qnet anomalies in Fig. 6b). The NordS-shift SAT response shows significant warming all over Greenland, the Baffin Bay and the northeastern North Atlantic. However, neither the heat released in the NordS area nor in the LabS area is able to spread to continental Europe.

## 5.1 Heat budget

To understand the SAT response of the two sea ice shift experiments, we consider the atmospheric heat budget. The heat budget is based on the thermodynamic energy equation (TEE) in which the conservation of energy is applied to a moving fluid (Holton, 2004):

$$\frac{\delta T}{\delta t} = -\boldsymbol{v} \cdot \nabla T - \frac{\delta T}{\delta p} \omega + \frac{\alpha}{c_p} \omega + \frac{J}{c_p}. \tag{1}$$

The terms of the TEE consist of the horizontal ($-\boldsymbol{v} \cdot \nabla T$) and vertical ($-\frac{\delta T}{\delta p} \omega$) heat flux convergence, the adiabatic compression ($\frac{\alpha}{c_p} \omega$) resulting from a vertical displacement of an air parcel, and diabatic processes ($\frac{J}{c_p}$) such as radiative or latent heating. Within the CAM4 model the heat budget is calculated considering modifications to the TEE as the physical principles are employed in a numerical modelling framework and certain processes need to be parameterized. For example, turbulence in the atmospheric boundary level is not resolved and consequently this transport is parameterized. Taking this into account, we use the simplified description of the CAM4 heat budget:

$$\frac{\delta T}{\delta t} = \text{HT}_{\text{dyn-core}} + \text{HT}_{\text{par}} + \frac{J}{c_p}. \tag{2}$$

In Eq. 2 the first three terms of the right hand side of the TEE (Eq. 1) are replaced with the heat transport resolved within the CAM4 dynamical core ($\text{HT}_{\text{dyn-core}}$) and the heat transport due to parameterized processes ($\text{HT}_{\text{par}}$). The latter mainly represents vertical heat transport due to sub-grid eddies. Note that those two heat transport terms refer to the CAM4 history fields named DTCORE and DTV, respectively. Note also that all simulations are run into equilibrium, so changes among the three terms of Eq. 2 compensate each other and the total temperature tendency ($\delta T/\delta t$) is almost zero.

The CAM4 heat budget response for both sea ice shift experiments is shown in Fig. 8 for the lowest terrain-following level. The winter mean temperature response at this level (not shown) strongly resembles the SAT response displayed in Fig. 7. The heat budget response to the LabS-shift experiment (Fig. 8a-c) indicates that over the Labrador Sea/Baffin Bay area $\text{HT}_{\text{par}}$ is the dominant process to vertically transport heat from the ocean surface to the overlying low-level atmosphere. In contrast, $\text{HT}_{\text{dyn-core}}$ is responsible to carry the excess heat away from the heat source area. This heat mainly accumulates in the North Atlantic area located south of Greenland where it is vertically mixed to the surface by sub-grid eddies (measured by $\text{HT}_{\text{par}}$) and, eventually, negative heat flux anomalies (Fig. 6b) that transfer the energy excess out of the atmosphere into the ocean. Furthermore, the warming in western Greenland (Fig. 7a) is related to enhanced $\text{HT}_{\text{par}}$ (Fig. 8b).

The response of the CAM4 heat budget to the NordS-shift is shown in Fig. 8d-f. Similarly to the LabS-shift experiment, the heat generated by the positive Qnet anomalies in the NordS sea ice retreat area (Fig. 6f) is vertically transported to the overlying





atmosphere by $HT_{par}$. Further, $HT_{dyn-core}$ is responsible for horizontally distributing the heat to the North Atlantic southwest of the sea ice retreat area. There, the excess heat is brought back down to the ocean surface by turbulent eddies (indicated as negative $HT_{par}$ anomaly, Fig. 8f) and is eventually lost to the ocean as revealed by negative Qnet anomalies (Fig. 6f). In contrast to the LabS-shift experiment, however, the sea ice retreat in the NordS also leads to distinct heat budget changes

over Greenland (Fig. 8). Depending on the Greenland region, the low-level warming is caused by either enhancement of the resolved ($HT_{dyn-core}$) or the parameterized ($HT_{par}$) heat transport (Fig. 8d,e). In contrast, diabatic processes are of secondary importance for explaining the spatial distribution of the heat released in the NordS source region (Fig. 8f). Above Greenland, the NordS-shift experiment mostly leads to a decrease in diabatic heating at low-levels (Fig. 8e) whereas the diabatic heating increases in the same areas at higher levels (not shown). This is explained by the fact that as atmospheric temperatures rise

above Greenland (see Fig. 7b) condensation of moisture is vertically shifted to higher atmospheric levels. In general, most of the diabatic heating response in both shift-experiments (Fig. 8c,f) can be attributed to changes in latent heating rather than radiative processes. Thus, the response of the cloud cover (that alters the radiation budget) to either sea ice perturbation is small and negligible (not shown).

Consequently, we find that moisture- and radiation-related processes are not of high relevance in explaining the presence

(absence) of a warming in Greenland in the NordS-shift (LabS-shift) experiment shown in Fig. 7. Instead, the warming in Greenland in the NordS-shift experiment is related to heat advection as suggested by the two heat transport terms (Fig. 8d,e). Theoretically, Greenland's warming can be caused by either direct advection of the heat from the heat source (i.e., the sea ice retreat area) or by changing the dynamics of the atmospheric flow above Greenland. Whereas the first process alters heat advection by changing temperature gradients, the latter has an impact on heat advection by changing the flow itself. In order to

analyze these processes in detail, we consider the low-level winds in and around Greenland (Fig. 9).

The atmospheric circulation in the NH during Eemian winters is similar to present-day winters (Merz et al., 2014a). The dominant circulation feature in Greenland is a stationary high-pressure system, known as the Greenland anticyclone (Hobbs, 1945). Accordingly, Greenland's wind field in the lower troposphere is characterized by strong winds that encircle Greenland clockwise whereas vertical winds indicate subsidence above the margins of the Greenland ice sheet (Fig. 9a). The Greenland

anticyclone, hence, can be regarded as an isolated wind system that hinders the exchange of heat and moisture between Greenland and adjacent areas. In the case of the LabS-shift experiment, the warming in the LabS area hardly leads to enhanced heat advection to Greenland as the winter mean winds do not point towards Greenland but rather to the North Atlantic areas located southeast (see vectors in Fig. 9a). There, enhanced heat advection is found based on the heat budget calculation (Fig. 8a) causing a local warming (Fig. 7a). The dynamic response of the winds in the LabS-shift experiment (Fig. 9b) even shows an

intensification of the northwestwards winds in the LabS area and implies an additional strengthening of the heat advection in southeasterly direction. In contrast, the low-level winds hardly change above Greenland and, thus, there is no indication for altered dynamics of the atmospheric flow above Greenland that would result in a respective temperature response in Greenland.

The Greenland anticyclone also acts as a barrier for heat approaching Greenland from the NordS area. In fact, the low-level winds east of Greenland indicate distinct atmospheric flow along Greenland's east coast (Fig. 9a) which further relates

to the Iceland low pressure system. Consequently, heat released by a sea ice retreat in the NordS domain is expected to be



advected southwards along Greenland's coast before being redirected by the cyclonic winds of the Icelandic low. Thus, there is no direct heat transport from the NordS domain towards central Greenland. However, the NordS-shift experiment shows distinct modifications to the low-level winds in and around Greenland (Fig. 9c): there is strong anomalous flow towards central Greenland from the North Atlantic area located to the southeast. Hence, the barrier effect of the Greenland anticyclone is

locally broken and warm air masses can enter Greenland. The vertical winds show anomalous upward motion in southeastern Greenland as the onshore winds are lifted over the steep margins of the ice sheet. Consequently, the sea ice perturbation of the NordS-shift experiment is able to substantially alter the atmospheric flow above Greenland leading to a change in heat transport (as indicated by the $HT_{\text{dyn-core}}$ and $HT_{\text{dyn-core}}$ anomalies in Fig. 8d,e). This, eventually, is responsible for the large-scale warming seen in Fig. 7b.

## 5.2 Moisture budget

Despite the result that moisture-related processes are not of high importance to explain the warming in either sea ice experiment (as explained in Sect. 5.1), the response of the hydrological cycle to the sea ice perturbations is substantial (Fig. 10). Changes in the hydrological cycle are described in terms of the atmospheric moisture budget that states that any change in moisture accumulation, defined as precipitation minus evaporation ($P - E$), must be compensated by moisture advection. The latter is

calculated as the convergence of the vertically-integrated zonal and meridional moisture fluxes. This calculation is based on daily model output using finite differences.

The LabS-shift response in $P - E$ shows that in the LabS area evaporation dominates over a concurrent precipitation increase (Fig. 10a). Hence, the sea ice retreat area acts as an atmospheric moisture source in addition to its role as heat source. The excess moisture is mainly transported eastwards (Fig. 10b) and deposited either in the North Atlantic located to the southeast or

in western Greenland. While the southeastward transport corresponds to the winter mean circulation indicated by the horizontal winds in Fig. 9a, the moisture advection to Greenland is due to synoptic systems (i.e., cyclones) that occasionally transport substantial amounts of moisture northwards along Greenland's west coast (Hutterli et al., 2005; Tsukernik et al., 2007) and, consequently, opposite to the winter mean circulation.

The response of the hydrological cycle to the sea ice shift in the NordS exhibits similar changes: in the areas of sea ice

reduction, increased evaporation (as also apparent in the latent heat flux, Fig. 6h) dominates over precipitation changes leading to distinctively negative $P - E$ anomalies (Fig. 10c). On the other hand, positive $P - E$ anomalies and hence increased moisture deposition are simulated for adjacent areas in the North Atlantic and in Greenland related to corresponding changes in moisture advection (Fig. 10d). For Greenland, most of the additionally available moisture precipitates above the steep margins of the ice sheet in the southeast where the moist air masses are lifted and, consequently, cause orographic precipitation. The resulting

maximum in winter precipitation in southeastern Greenland is a prominent feature in the North Atlantic winter climate (e.g., Tsukernik et al., 2007; Merz et al., 2014b) related to a local maximum in cyclone frequency in the area of the Icelandic low. Enhanced moisture availability in the NordS domain, thus, results in a precipitation increase in this specific Greenland region with cyclones being the carrier. Moreover, increased precipitation in southeastern Greenland relates to the previous result of





an enhancement of the onshore winds in response to the NordS-shift (Fig. 9c). Hence, the dynamic response itself fosters the advection of both heat and moisture from the Nordic Seas towards eastern Greenland.

### 5.3 Seasonality

The results presented so far show a distinct impact of regional sea ice reductions on the winter climate in the North Atlantic
sector. To assess the importance of changes in sea ice cover for the interpretation of Eemian climate proxy records, which mostly reflect annual mean changes, the temporal scope shall be broadened to the other seasons. In the following, we analyze the relationship between the seasonality in sea ice reduction and the seasonality of the atmospheric response. For this purpose, we compute the annual cycles of the area-averaged SIC, Qnet, and SAT anomalies for the LabS domain (Fig. 11a,c) and the NordS domain (Fig. 11b,d), respectively. Thereby, the responses to the two sea ice shift experiments (Fig. 11a,b) and to
EEM-PI$_{diff}$ (Fig. 11c,d) are compared.

The average monthly SIC reduction in the LabS domain as a result of the LabS-shift varies between 10-20% reduction (Fig. 11a). As previously discussed, a certain retreat in the sea ice cover reflects a change in lower boundary conditions to the atmosphere influencing the exchange of heat and moisture at the ocean–atmosphere interface. Hence in terms of energy, the sea ice retreat is transferred to the overlying atmosphere by anomalous net surface energy fluxes (Qnet in Fig. 11a). The
LabS-shift results in a distinct annual cycle of the Qnet response with the maximum increase during winter in contrast to almost no change in summer. Hence, the magnitude of the Qnet response is not tied to the concurrent SIC reduction but rather to seasonally diverse climate conditions. More precisely, we find a winter maximum in the turbulent (i.e., sensible and latent) heat flux response arising from the fact that this is the time of year when the low-level air temperatures are coolest relative to the underlying surface (sea ice or open water). Consequently, a sea ice retreat that exposes SSTs to the overlying
atmosphere has a distinct "heat source effect" in the winter half-year but hardly the same effect in summer when atmospheric and surface ocean temperatures are comparable. This seasonally diverse behavior of the heat flux response to changes in sea ice is well-known and has previously been identified in model and reanalysis studies investigating recent and future Arctic sea ice changes (Deser et al., 2010; Screen and Simmonds, 2010; Screen et al., 2013). As expected, an increase in the net energy flux directly translates in a local SAT signal and, thus, the annual cycles of Qnet and SAT strongly resemble each other (Fig. 11a).
Accordingly, the maximum SAT response in the LabS domain emerges in winter (>5°C) coinciding with the Qnet maximum. Vice-versa, the summer warming is of smaller magnitude (∼1°C).

Equivalently to the LabS-shift experiment, the NordS-shift results in a SIC reduction in the range of 10-15% throughout the year (Fig. 11b). However, the Qnet response to the NordS-shift lacks the winter maximum previously found for the LabS-shift (compare Fig. 11a and b). This is explained by the dipole effect in turbulent heat fluxes (see Fig. 6g,h): the strongly positive
heat flux anomalies in the sea ice retreat areas are partly offset by negative anomalies in adjacent areas when averaging across the NordS domain (as done in Fig. 11b). In contrast, in the LabS-shift experiment the negative part of the heat flux dipole is located outside of the LabS domain (see Fig. 6c,d) and hence not considered in the calculation of the Qnet values shown in Fig. 11a. Nevertheless, the seasonality of the SAT response to the NordS-shift (Fig. 11b) is similar to the LabS-shift experiment with a winter maximum and a summer minimum, respectively.



In summary, we find that a sea ice retreat substantially influences the local winter climate in both regions whereas the response of the summer climate is of smaller amplitude. The same result is true for the effect of the differing lower boundary conditions denoted by EEM-PI$_{diff}$ as shown by the annual cycles in Fig. 11c,d. Hence, although EEM-PI$_{diff}$ exhibits a sea ice reduction in any season and not mostly distinctively in winter, the Qnet and SAT response is largest in the cold season. In

the LabS domain, the winter sea ice reduction corresponding to EEM-PI$_{diff}$ is considerably smaller than for the LabS-shift experiment (compare Fig. 11a and 11c) and, accordingly, the Qnet and SAT maxima in winter are less distinct. On the other hand, the EEM-PI$_{diff}$ SIC reduction in the NordS domain during winter is in the same range as in the NordS-shift experiment (compare Fig. 11b and 11d) so the respective SAT responses are of similar magnitude as well. The comparability of the results of EEM-PI$_{diff}$ and the two shift-experiments further illustrates the utility of the idealized sea ice sensitivity experiments for

identifying the impact of regional sea ice changes on the Eemian climate.

Additionally to the seasonality of sea ice changes and its response on the overlying atmosphere, we assess the annual cycle in Greenland's SAT response (Fig. 11e). In contrast to the SAT response in the area of sea ice perturbation (i.e., the LabS or NordS), which is the direct result of altered surface energy fluxes, a change in Greenland temperatures additionally requires anomalous heat transport (as discussed in Sect. 5.1). The EEM-PI$_{diff}$ Greenland SAT response shows a distinct warming in

winter/spring but only a moderate warming during the warm season. Furthermore, the NordS-shift results in a very similar warming response as EEM-PI$_{diff}$ consolidating the previous result that a sea ice retreat in the NordS is crucial to explain the widespread warming seen in EEM-PI$_{diff}$ (Fig. 4 bottom row). In contrast, the sea ice perturbation caused by the LabS-shift hardly leads to higher temperatures in Greenland in any season. Hence, despite the distinct local warming (particularly in winter) caused by the LabS-shift (Fig. 11a) non-existing heat transport towards Greenland prevents a Greenland warming in

any season (Fig. 11e).

## 5.4   Impact of experimental design

As introduced in Sect. 3, we perform additional sea ice sensitivity experiments in which we test modifications to the sea ice shift approach. The results of these simulations with respect to the SAT response in the area of sea ice perturbation (LabS or NordS) as well as in Greenland are listed in Table 3.

In EEM2$_{LabS}$ and EEM2$_{NordS}$ we use the EEM$_{highRes}$ lower boundary conditions as a baseline to apply the shift instead of those of EEM$_{lowRes}$ used so far for EEM$_{LabS}$ and EEM$_{NordS}$. Fig. 3 shows that the position of the Eemian sea ice edge in EEM$_{lowRes}$ differs from EEM$_{highRes}$. Applying the shift to the latter, thus, results in a change of the location of the sea ice anomalies and hence in the location of the strongest heat flux anomalies (i.e., the heat source). In EEM2$_{LabS}$ and EEM2$_{NordS}$ the resulting heat source regions are shifted northwards with respect to EEM$_{LabS}$ and EEM$_{NordS}$. Comparing the temperature

response of EEM2$_{LabS}$/EEM$_{LabS}$ and EEM2$_{NordS}$/EEM$_{NordS}$, respectively (see Table 3), we find that shifting the position of the heat source area only has a moderate effect on the local warming as well as on the response in Greenland. Still, a northward shift of the heat source area seems to reduce the magnitude of warming.

Moreover, we generate four sensitivity experiments for which we shift the SICs but not the SSTs in order to exclude the response to a (possibly overestimated) surface ocean warming that comes along with the SST-shift (previously discussed in





Sect. 3). Accordingly, in these simulations (denoted with an $_{ICE}$-suffix in Table 3) the heat source is restricted to the area of sea ice retreat as the SST anomalies shown in Fig. 6a and Fig. 6e are omitted. In the LabS-region this model setup appears to be of minor importance as the warming response in both $_{ICE}$-simulations does not deviate from the response in the experiments including the SST-shift. In contrast, the effect is much larger for the NordS-shift experiment where ignoring the widespread SST

increase (shown in Fig. 6e) substantially reduces the strength of the heat source. Consequently, the $_{ICE}$-simulations generate a smaller temperature response compared to the simulations including the SST-shift.

Our whole set of sensitivity experiments covers a reasonable range of possible sea ice (and related SST) changes in the two target regions. The following results are robust among all simulations performed (based on Table 3): (i) a sea ice reduction in the LabS domain leads to a strong local warming (DJF: 5.3-6.0°C; annual: 2.3-3.6°C). (ii) The response in Greenland temperature

to a perturbation in the LabS is limited due to the lack of heat transport towards Greenland. The annual mean Greenland SAT increase of 0.4-0.5°C still is a significant warming but mostly reflects the warming in western Greenland shown in Fig. 7a. (iii) In the NordS region the strength of the heat source depends on the specific experimental setting (i.e., inclusion/exclusion of SST changes, location of the perturbation). This results in a considerable spread in the NordS temperature response (DJF: 2.3-4.6°C; annual: 1.2-3.1°C). (iv) Correspondingly, there is a spread in terms of warming in Greenland (DJF: 1.1-3.8°C; annual:

0.6-2.1°C) depending on the strength of the heat source in the NordS. (v) The impact of the NordS-shift on the Greenland SAT outranges the influence of the LabS-shift in all cases considered here.

Comparing the temperature responses of the sensitivity experiments and EEM-PI$_{diff}$ (Table 3) gives further insights into how the idealized experiments relate to the effect of different lower boundary conditions on the Eemian warming analyzed in Sect. 4. The EEM-PI$_{diff}$ in the LabS region is below the range of the sensitivity simulations implying that the heat source

employed in the idealized LabS-shift experiments is rather overestimated. In contrast, the EEM-PI$_{diff}$ warming in the NordS area conforms with the idealized experiment featuring the strongest heat source in the NordS (i.e., EEM$_{NordS}$). Hence, the scenario of a distinct sea ice reduction and surface warming included in EEM$_{NordS}$ (Fig. 6e) is in correspondence with EEM-PI changes simulated by state-of-the-art climate models.

# 6   Discussion

The results show how the representation of the lower boundary conditions (i.e., sea ice and SSTs) is crucial for the simulated warming during the Eemian, particularly in the North Atlantic. Substantially warmer than present annual mean SATs during the Eemian, as observed in proxy records (e.g., Turney and Jones, 2010), require warmer than present SSTs and a reduced sea ice cover. In fact, the external forcing of the Eemian used for respective climate model simulations consists of the orbital forcing leading to seasonally diverse insolation anomalies and lower than present GHG concentrations (Lunt et al., 2013, and

references therein). The direct effect of the climate system to this external forcing alone does not explain a year-round Eemian warming. Instead positive feedbacks associated with changes in sea ice, land ice, snow cover, and vegetation changes are required, especially to explain the distinct warming observed in the NH high latitudes resulting in a polar amplification pattern (CAPE Last Interglacial Project Members, 2006).



In this study, we show for the CCSM3 model that differences in the simulation of the lower boundary conditions explain most of the spread with respect to the EEM-PI atmospheric warming in the NH (see Fig. 2 and text in Sect. 4). Hence, feedbacks and changes in the model's ocean and sea ice component clearly influence the magnitude of the Eemian warming in the atmosphere. We hypothesize that the same is true for the remarkable spread found among the wide range of models in Lunt et al. (2013).

Furthermore, a climate model, which simulates for the Eemian warmer SSTs and a reduced sea ice cover and, consequently, a stronger atmospheric warming (here EEM-PI$_{highRes}$), is more in line with NH proxy records (Turney and Jones, 2010). This is true with respect to both marine and terrestrial temperature proxies. The picture, however, gets complicated when comparing models and proxy data on a regional scale as the proxies exhibit a wide range of Eemian minus pre-industrial temperature anomalies at similar latitudes (Otto-Bliesner et al., 2013; Lunt et al., 2013). Besides the spatial variability, a further degree of

complexity arises when considering the temporal evolution of the temperature proxy records throughout the Eemian which are rarely represented correctly in the models (Capron et al., 2014).

Another specific goal of this study is to assess the impact of sea ice changes on the climate in Greenland and its implications for temperature records derived from Greenland ice cores. Long-term records are available from deep ice cores, which are mostly drilled on top of the Greenland ice sheet. Currently, the NEEM core (NEEM community members, 2013) is the only

ice core covering the entire Eemian period. The Eemian ice in NEEM was originally deposited at pNEEM (Merz et al., 2014a, b), a location relatively close to the summit of the ice sheet. Consequently, we are interested in the simulated Eemian climate at pNEEM located approximately at 76°N/44°W. The temperature response at pNEEM to the shift-experiments as well as to the different EEM-PI$_{diff}$ lower boundary conditions is shown in Fig. 12. This figure confirms that sea ice and SST changes in the LabS area are hardly recorded on top of the Greenland ice sheet in contrast to the NordS-shift experiment and EEM-PI$_{diff}$.

The latter two are both characterized by distinct sea ice reductions in the NordS area (Fig. 6e and 5a) leading to a notable atmospheric warming above the oceans east of Greenland (Fig. 12b,c). Furthermore, the dynamical response of the atmosphere to the sea ice perturbation in the NordS area results in a widespread temperature response as the additionally available heat spreads over the lower troposphere of the North Atlantic and, thus, also to the Greenland ice core sites including pNEEM. Consequently, temperature records based on Greenland ice cores are sensitive to sea ice changes in the NordS area but rather

insensitive to sea ice changes in the LabS area. This is consistent with results by Li et al. (2010) who reported similar findings for glacial climate conditions, i.e., a substantial warming throughout Greenland for a sea ice reduction in the Nordic Seas but little impact of sea ice changes in the western North Atlantic. Hence, the demonstrated relationship between Greenland temperature and sea ice in the adjacent oceanic areas is not limited to the Eemian but very likely valid for any interglacial and glacial climate period. In the recent past, however, the strongest sea ice retreats have been detected in other NH regions, i.e.,

the Chukchi, East Siberian and Barents Sea leading to strong temperature responses in these regions (Screen and Simmonds, 2010; Vaughan et al., 2013). Nevertheless, projections for the twenty-first century also suggest a reduction of the remaining sea ice in the NordS and a related winter warming in Greenland (Deser et al., 2010).

Quantitative estimates of sea ice induced annual mean SAT changes in central Greenland including the pNEEM site are further shown in Table 4. All LabS-shift experiments results in statistically insignificant warming of at most 0.3°C. In contrast,

the NordS-shift experiments all result in significant annual mean warming in the range of 0.6-2.3°C. The magnitude of the





warming in central Greenland relates to the strength of the heat source in the NordS depending on whether a warming in SSTs accompanies the sea ice reduction (see details in Sect. 5). The $EEM_{NordS}$ and $EEM2_{NordS}$ experiments that include both sea ice and SST changes further show a significant increase in snow accumulation in central Greenland. This manifests the role of the NordS area as a moisture source for Greenland besides its role as heat source. Further, this implies that oceanic changes in the

NordS affect ice core based accumulation records.

     Measurements of the Eemian $\delta^{15}$N in the NEEM core suggest that annual mean Eemian firn temperatures were on average 5°C warmer than at present-day (NEEM community members, 2013). Based on our CCSM4 simulations we find an Eemian minus pre-industrial annual mean warming in central Greenland of 0.5°C ($EEM-PI_{lowRes}$) and 2.1°C ($EEM-PI_{highRes}$), respectively. Thus, the difference of 1.6°C for the Eemian warming relates to the different changes in the lower boundary conditions

(see $EEM-PI_{diff}$ in Table 4). Nevertheless, additional warming mechanisms not accounted for in this model framework are needed to explain the full magnitude of the determined $\delta^{15}$N signal. One possibility is an even stronger reduction in the NordS sea ice than considered in $EEM-PI_{highRes}$ resulting in an additional warming equivalent to the NordS-shift experiments. Another possible candidate are surface climate changes that relate to modifications in the Greenland ice sheet topography because in order to conform with observed sea level high stands (Church et al., 2013), Greenland must have been smaller during the

Eemian. Depending on the actual ice sheet topography this results in an additional annual mean warming of up to 3.1°C at pNEEM (Merz et al., 2014a). Hence, if a strong reduction in NordS sea ice coincided with a distinct retreat of the Greenland ice sheet, the full magnitude of the NEEM $\delta^{15}$N signal can be explained. Furthermore, the sea ice- and topography-related warming mechanisms may interact with each other as both modify Greenland's low-level winds. In order to assess possible feedbacks, it might be worth to generate respective model experiments that combine perturbations in sea ice with changes in

the Greenland ice sheet topography. Still, it is important to note that both the sea ice-related as well as the ice sheet topography-related warming mechanisms are rather of local nature and do not result in a respective warming in more distant regions, e.g., Europe. This implies that the distinct Eemian warming retrieved from the NEEM core shall be interpreted as a local rather than a hemispheric-scale climate signal.

     Sea ice changes further influence the stable water isotopes measured in the NEEM core, which show a reduced depletion

tion of at least 3‰ for the Eemian $\delta^{18}$O with respect to present-day (NEEM community members, 2013). Applying the temperature–$\delta^{18}$O relationship determined for the current interglacial, this translates in an Eemian temperature increase of 8±4°C (NEEM community members, 2013). Correspondingly, the NEEM $\delta^{18}$O record suggests an even stronger Eemian warming than measured in $\delta^{15}$N. Sime et al. (2013) showed within isotopic simulations that a reduction in the winter sea ice cover around the northern half of Greenland, together with an increase in SSTs in the same region, is sufficient to cause a

>3‰ interglacial enrichment of $\delta^{18}$O in central Greenland snow. The changes in SST and sea ice further lead to higher $\delta^{18}$O-temperature gradients, so a >3‰ enrichment in $\delta^{18}$O might rather correspond to a 5°C warming, which would be more in line with $\delta^{15}$N. Thereby, the underlying mechanism is that a reduction in sea ice increases the fraction of water vapor deposited in central Greenland originating from more local (isotopically enriched) at the expense of more distant (isotopically depleted) sources (Sime et al., 2013). However, the fact that the Eemian climate change in Greenland mainly occurs during summer when



the orbital forcing is strongest whereas $\delta^{18}$O is less tied to temperature in summer than in winter (Sjolte et al., 2014), further complicates a meaningful interpretation of the NEEM $\delta^{18}$O record.

## 7 Summary

We analyzed the response of the atmospheric component of the CCSM4 climate model to pre-industrial and Eemian lower boundary conditions (i.e., sea ice and SSTs) as well as to a set of idealized sea ice retreat scenarios. The overarching goal of the study was to demonstrate the role of sea ice for the warm climate of the Eemian, particularly for Greenland. The main findings are:

– The magnitude of the simulated Eemian warming in the NH strongly depends on concurrent changes in sea ice and SSTs. Fully-coupled models which simulate higher SSTs and a retreating sea ice cover for the Eemian with respect to present-day also show a stronger atmospheric warming. These simulations are in better agreement with Eemian SST and SAT proxy records.

– The effect of sea ice and SSTs on the NH climate is strongest in winter due to the maximum response of the surface energy fluxes during the colder season.

– Greenland temperatures are strongly influenced by the sea ice cover and SSTs in the Nordic Seas. In contrast, the impact of the Labrador Sea sea ice on the Greenland climate is marginal.

– Anomalous heat advection is the primary process to explain the large-scale warming found in response to a sea ice retreat in the Nordic Seas. Despite the fact that a sea ice retreat also has a significant impact on the North Atlantic moisture budget, anomalous diabatic heating associated with condensation processes is small and of lower order importance for the simulated temperature response.

– The Greenland anticyclone acts as a barrier for heat and moisture approaching Greenland and hinders a sea ice-induced warming in the Labrador Sea from spreading towards central Greenland. In contrast, the sea ice retreat in the Nordic Seas has a greater effect on the atmospheric dynamics in Greenland resulting in anomalous winds that break up the anticyclone and allow a wide-spread Greenland warming.

– The Eemian annual mean warming of 5°C above present-day derived from the NEEM $\delta^{15}$N record is consistent with CCSM4 model simulations for the scenario that a retreat in the Nordic Sea sea ice coincided with a reduction in the Greenland ice sheet. The model emphasizes that this distinct Greenland warming is mostly a local signal.

Note that our experiments only address the direct impact of North Atlantic sea ice loss on the surface climate and atmospheric circulation and hence neglect potential oceanic feedbacks. We are, however, confident that our results are robust as the dominant mechanism, which thermally transfers sea ice anomalies to the atmosphere (i.e., anomalous turbulent heat fluxes), is similar in fully-coupled and atmosphere-only simulations (Deser et al., 2010). Nevertheless, it would be interesting to repeat the sea ice





sensitivity experiments presented here in a fully-coupled model framework, e.g., analogue to Lehner et al. (2013), in order to assess the consequences for the ocean circulation and respective feedbacks to the atmosphere.

*Acknowledgements.* We kindly thank Vidya Varma, Matthias Prange, Ute Merkel and the NCAR for providing model output used as boundary conditions for our experiments. All simulations were produced at the Swiss National Supercomputing Centre (CSCS). We acknowledge

5    financial support by the Swiss National Science Foundation.



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



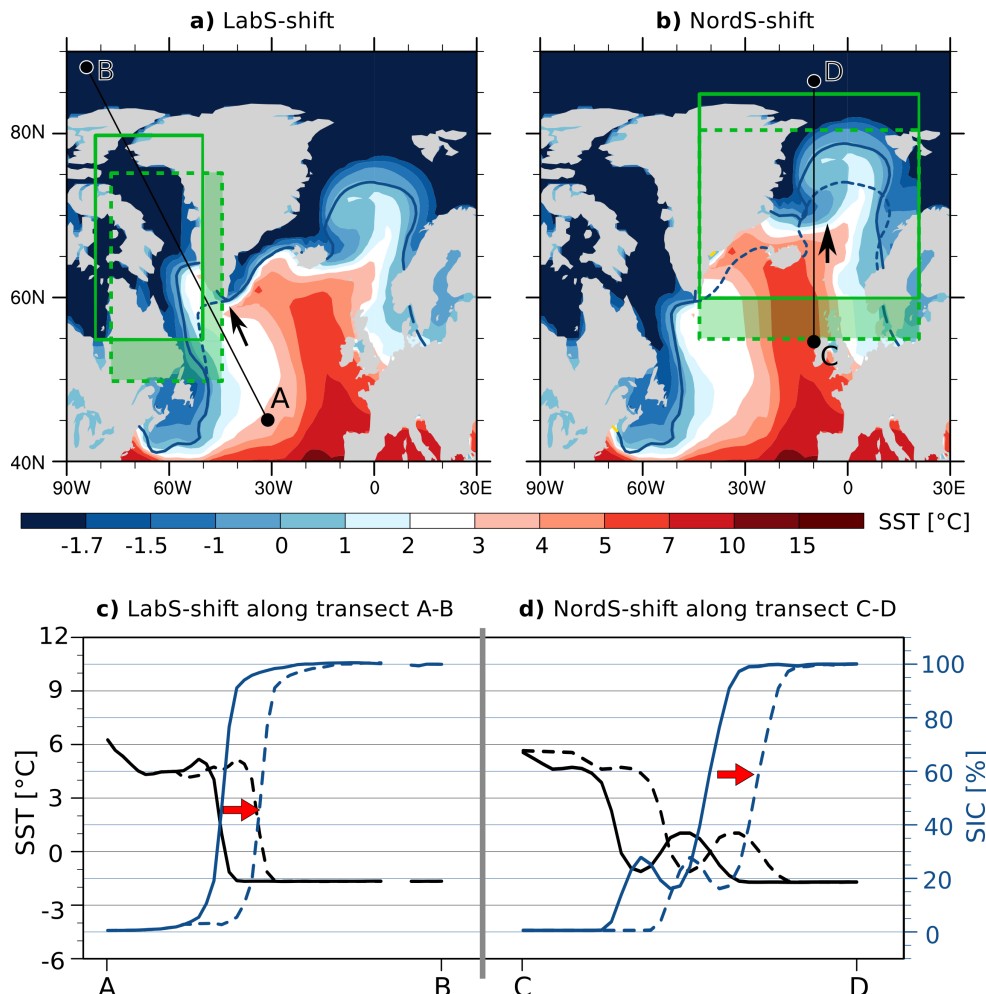

**Figure 1.** Illustration of sea ice shift experiments (shown for mean winter (DJF) conditions) in two areas around Greenland enclosed in the green rectangles: a) Labrador Sea shift in sea surface temperature (SST, shaded) and sea ice concentration (SIC, 50% solid contour) in EEM$_{LabS}$. b) Nordic Seas shift in SST (shaded) and SIC (50% solid contour) in EEM$_{NordS}$. The dashed contours in a) and b) denote the 50% SIC isoline before the shift (i.e., in EEM$_{lowRes}$). c) EEM$_{LabS}$ (dashed) vs. EEM$_{lowRes}$ (solid) SST and SIC along the Labrador Sea transect (A→B). d) EEM$_{NordS}$ (dashed) vs. EEM$_{lowRes}$ (solid) SST and SIC along the Nordic Seas transect (C→D).





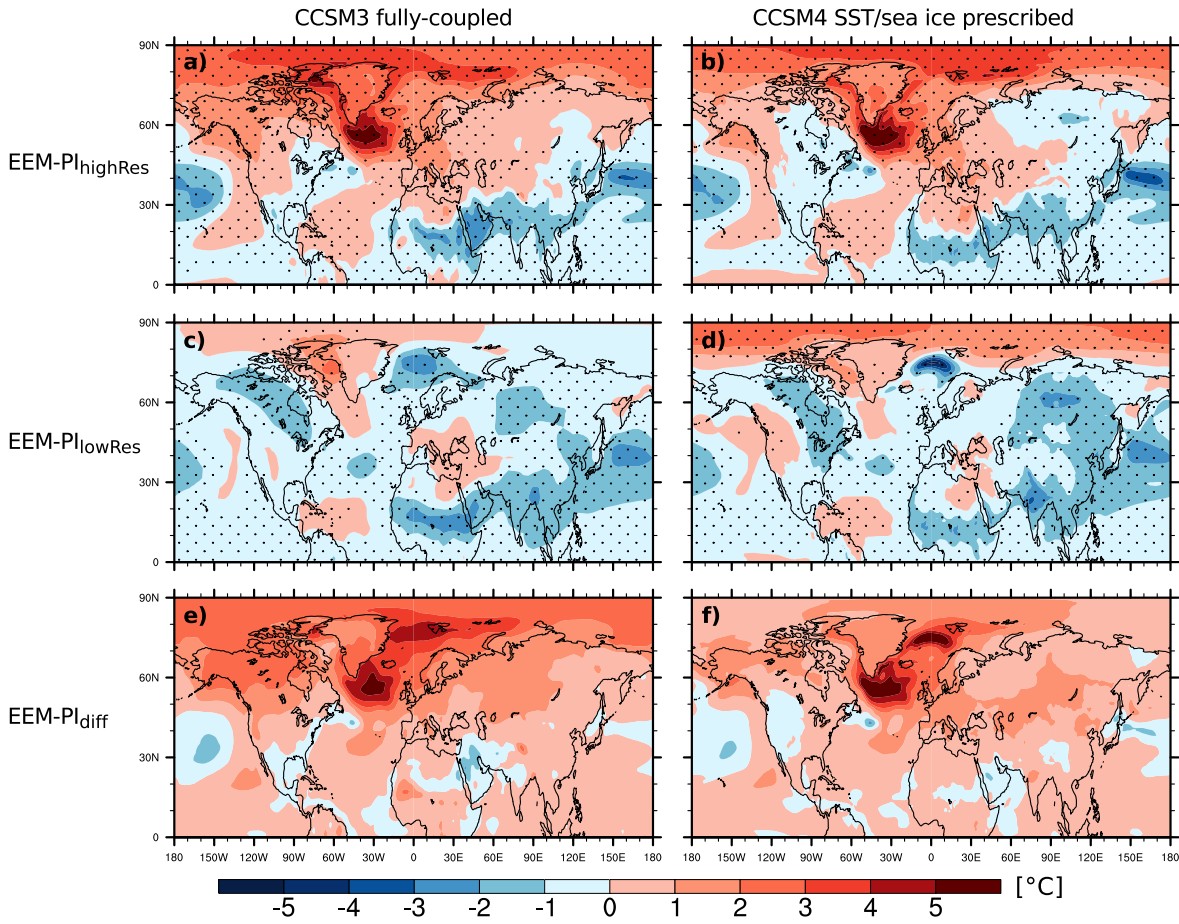

**Figure 2.** Eemian minus pre-industrial (EEM-PI) annual mean surface air temperature (SAT) change in (a,c,e) the fully-coupled CCSM3 and (b,d,f) the atmosphere-land-only CCSM4 with prescribed sea surface temperature (SST) and sea ice from the corresponding CCSM3 simulations. The top row shows the results from the highRes experiments, the middle row those from the lowRes experiments and the bottom row their differences, respectively. Stippling in the top and middle row denotes EEM-PI changes significant at the 5% level based on t-test statistics.




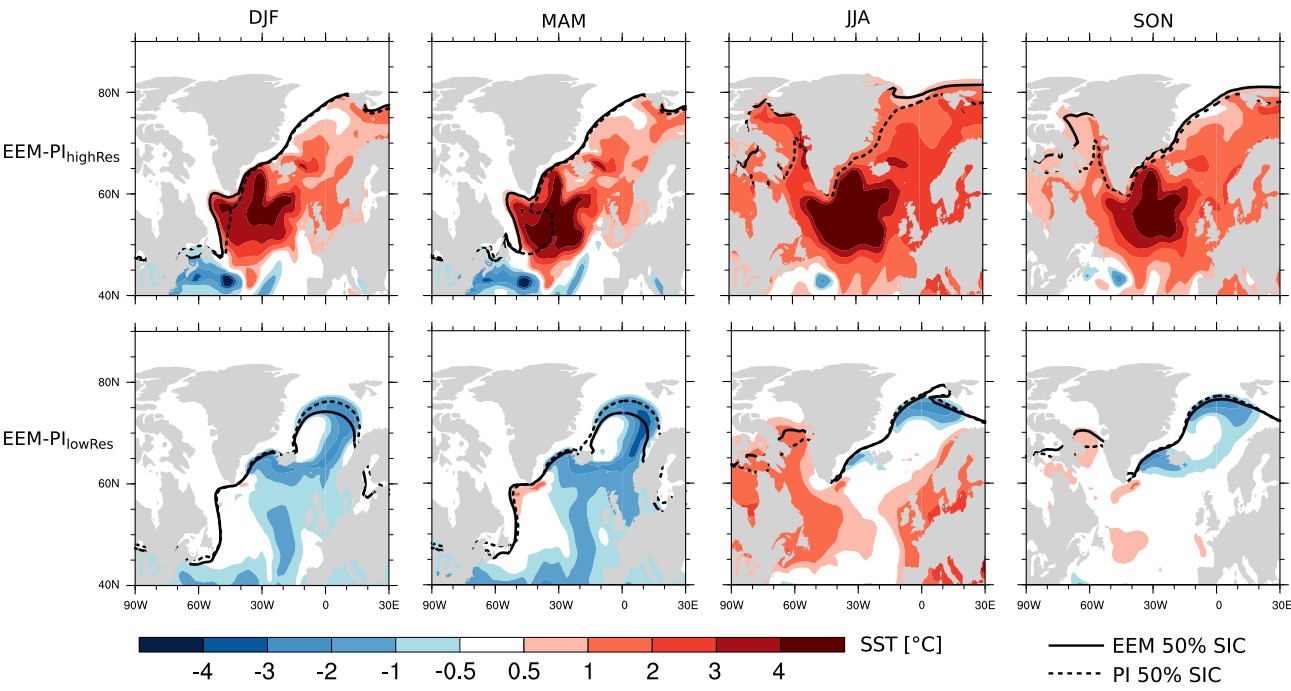

**Figure 3.** CCSM3 Eemian minus pre-industrial (EEM-PI) seasonal mean sea surface temperature change (SST, shaded) and EEM (solid) vs. PI (dashed) 50% sea ice concentration (SIC) contours. The top row is based on the highRes simulations and the bottom row on the lowRes simulations, respectively. Note that these SST/SIC fields are used as lower boundary conditions for the respective atmosphere-land-only CCSM4 simulations.





**Figure 4.** CCSM4 Eemian minus pre-industrial (EEM-PI) seasonal mean surface air temperature (SAT) change. The top row shows the result from the highRes experiments, the middle row the lowRes experiments and the bottom row their differences, respectively. Stippling in the top and middle row denotes EEM-PI changes significant at the 5% level based on t-test statistics.





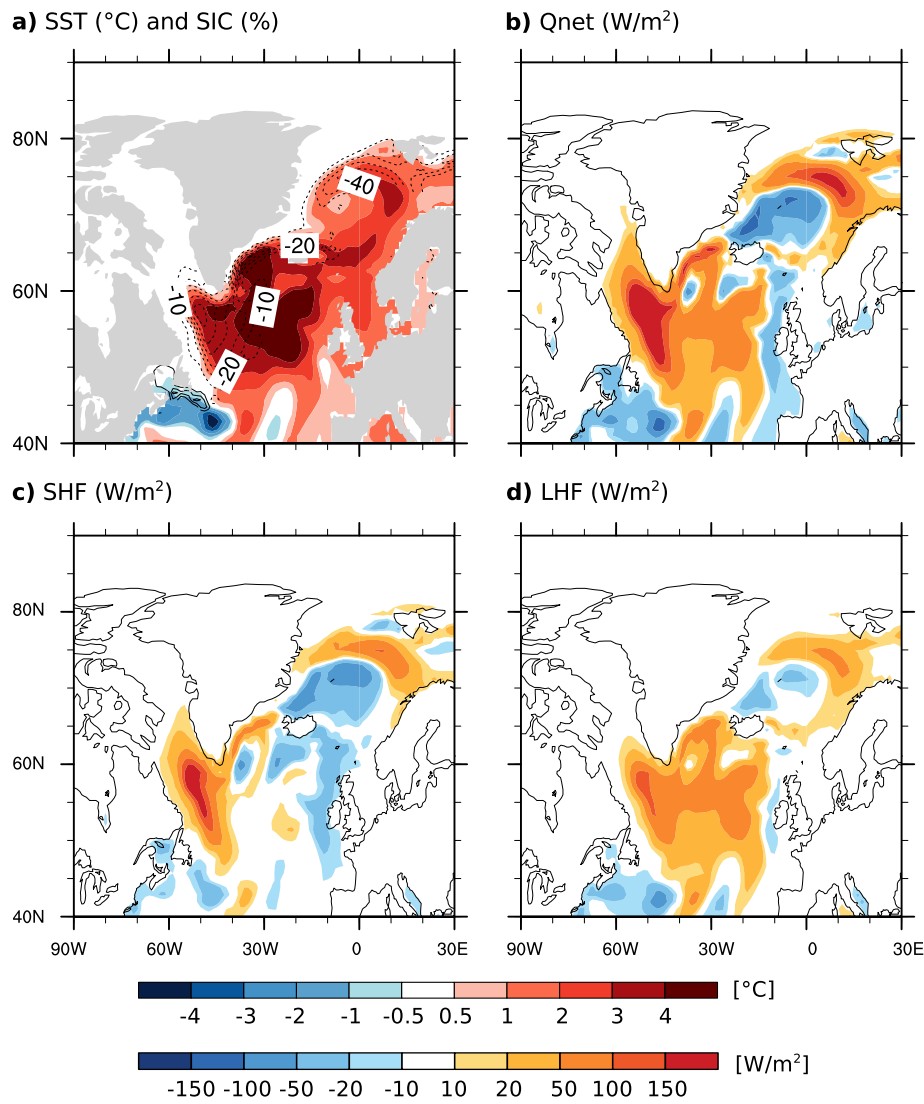

**Figure 5.** CCSM4 EEM-PI$_{diff}$ response in winter (DJF) mean a) sea surface temperature (SST, shaded) and sea ice concentration (SIC, contours), b) net surface energy flux (Qnet), c) sensible heat flux (SHF) and d) latent heat flux (LHF). Negative sea ice anomalies in a) are dashed and the contour interval is 10%. Energy fluxes are positive upward.





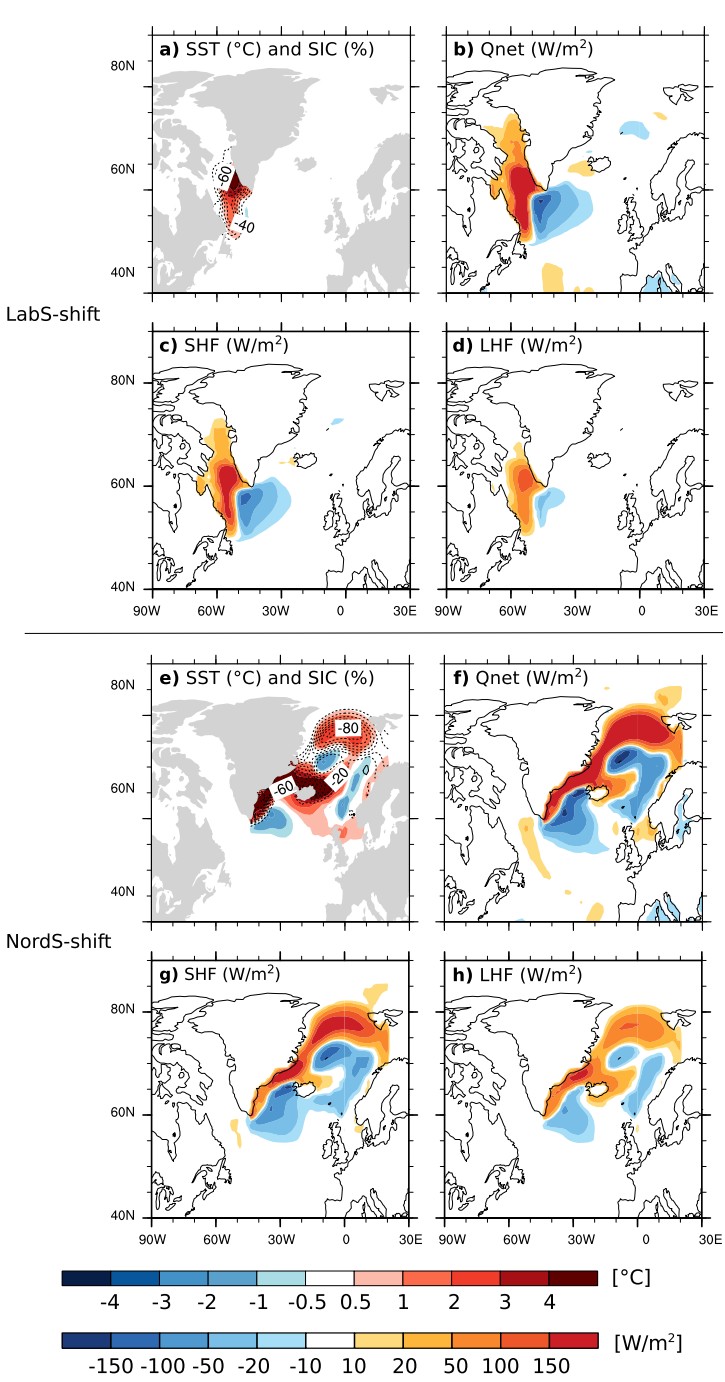

**Figure 6.** Same as Fig. 5 but for the LabS-shift response (a-d) and the NordS-shift response (e-h), respectively.



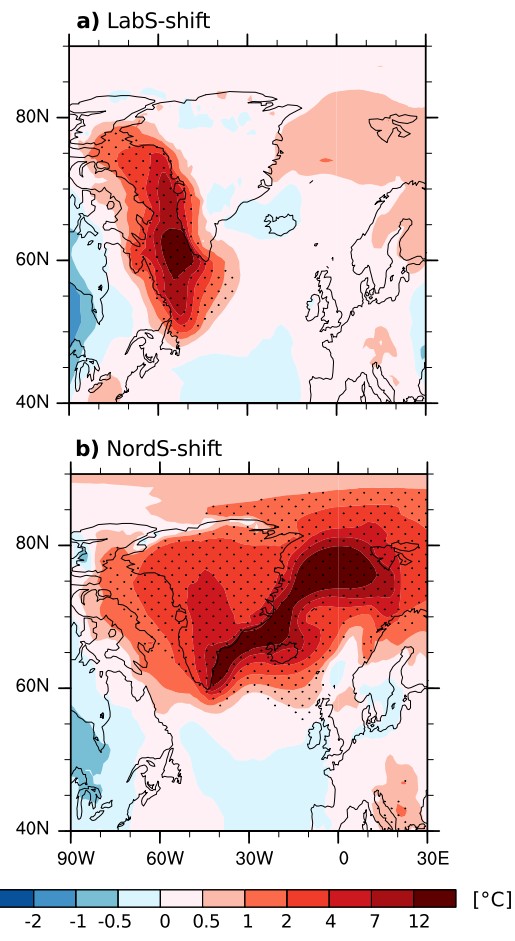

**Figure 7.** a) LabS-shift and b) NordS-shift response in winter (DJF) mean surface air temperature (SAT). Stippling denotes values significant at the 5% level based on t-test statistics.





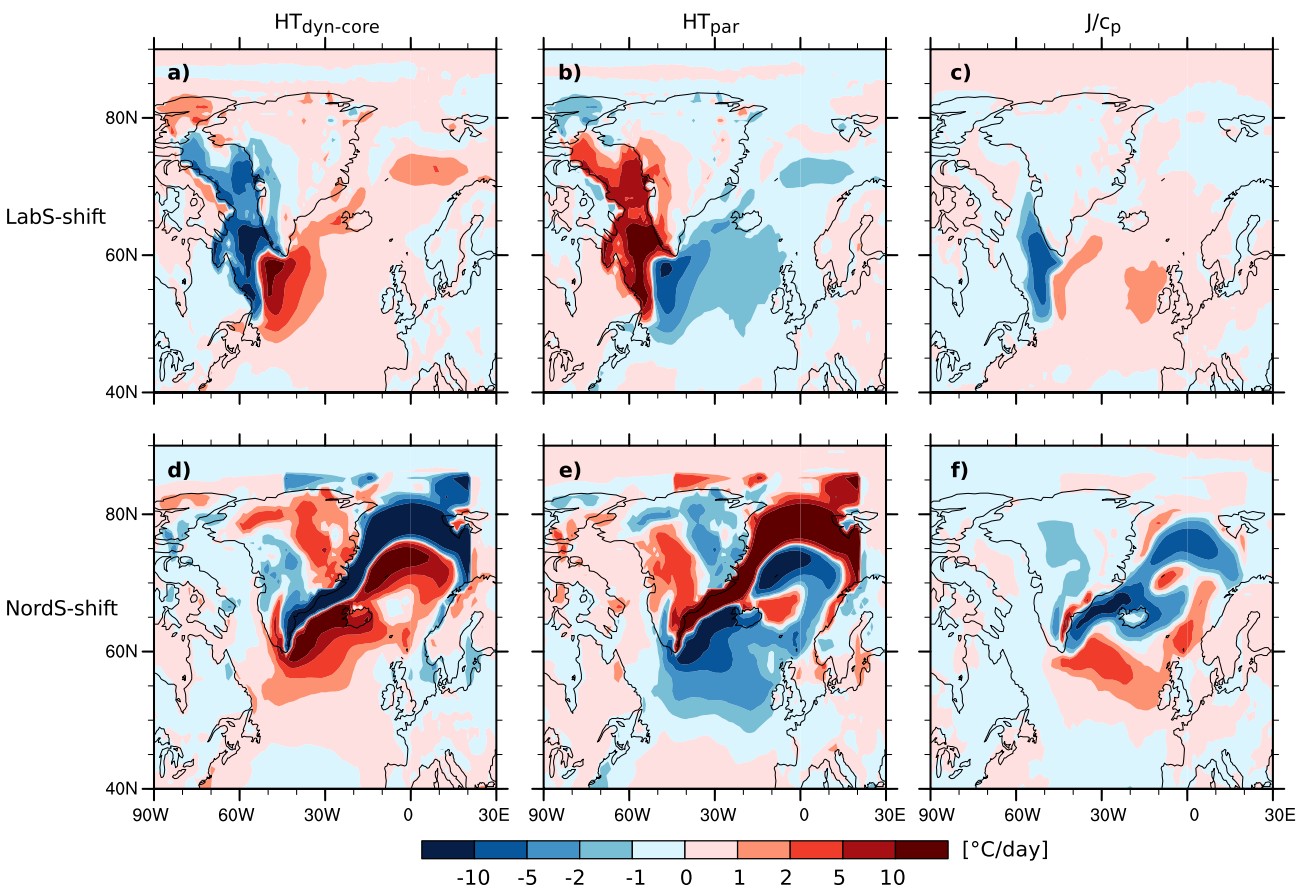

**Figure 8.** LabS-shift and NordS-shift response in winter (DJF) mean CAM4 heat budget components as given in Eq. 2 at the lowest terrain-following model level: temperature tendencies associated with a) and d) heat transport resolved within the CAM4 dynamical core ($HT_{dyn\text{-}core}$); b) and e) heat transport due to CAM4 parameterizations ($HT_{par}$); c) and f) diabatic processes ($\frac{J}{c_p}$).





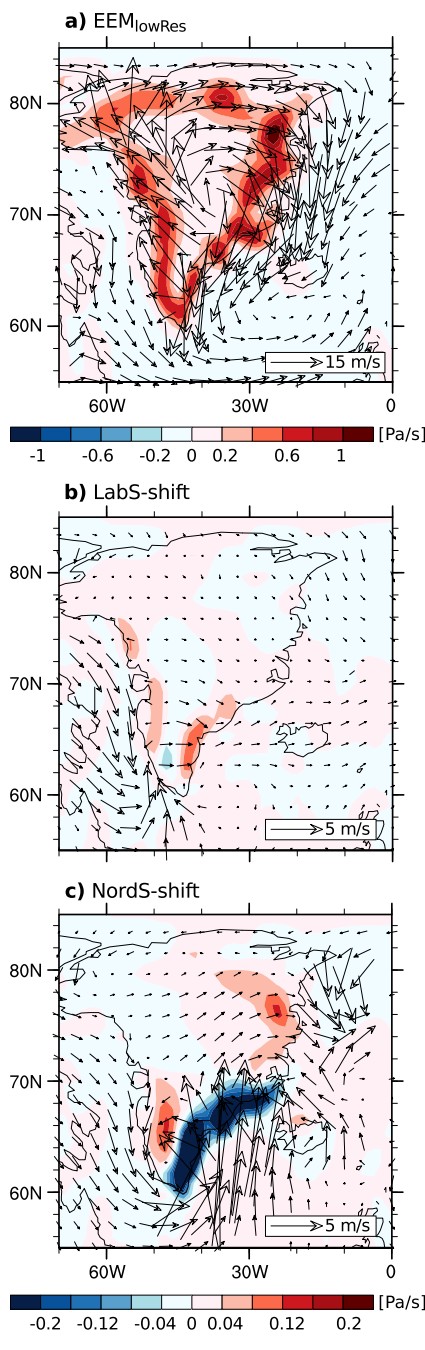

**Figure 9.** Winter (DJF) mean vertical (shaded) and horizontal (vectors) wind velocities at lowest terrain-following model level for a) EEM$_{lowRes}$, b) LabS-shift response, and c) NordS-shift response. Positive (negative) vertical wind velocities denote downward (upward) motion.







**Figure 10.** LabS-shift and NordS-shift response in winter (DJF) mean moisture budget: a) and c) denotes precipitation minus evaporation ($P - E$); b) and d) shows the vertically-integrated moisture fluxes (vectors) and their convergence (-div($\mathbf{Q}$), shaded), respectively. Stippling in a) and c) indicates $P - E$ changes significant at the 5% level based on t-test statistics.





**Figure 11.** Annual cycle of sea ice concentration (SIC, blue shading), net surface energy flux (Qnet, green lines) and surface air temperature (SAT, red lines) anomalies: a) response to LabS-shift for the LabS domain, b) response to NordS-shift for the NordS domain, c) EEM-PI$_{diff}$ response for the LabS domain, d) EEM-PI$_{diff}$ response for the NordS domain, and e) Greenland mean SAT response to LabS-shift (dotted), NordS-shift (dashed), and EEM-PI$_{diff}$ (solid). The LabS domain is designated as all oceanic grid points within the solid box in Fig. 1a and the NordS domain is the equivalent in Fig. 1b. Note that all annual cycles are calculated as spatial averages including area weighting.





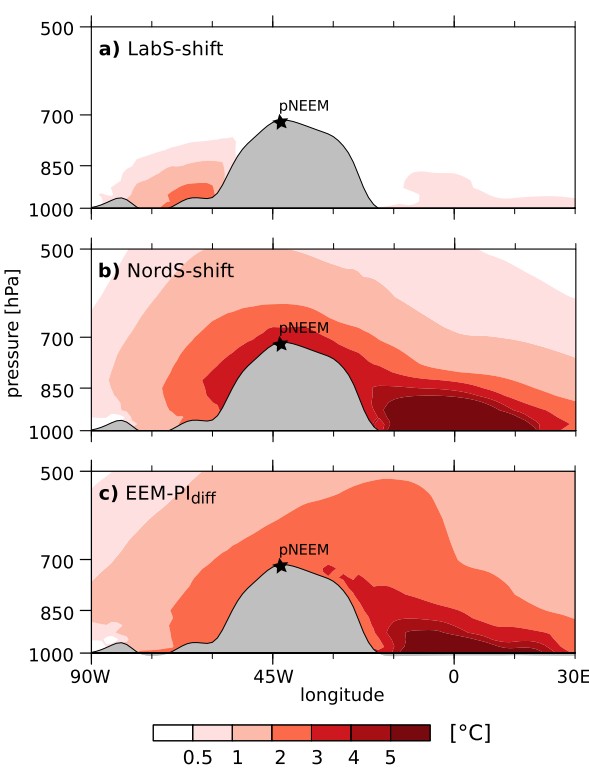

**Figure 12.** a) LabS-shift, b) NordS-shift, and c) EEM-PI$_{diff}$ response in winter (DJF) mean temperature shown as longitude–pressure cross section along the $76°$N latitude (i.e., the latitude of pNEEM).





**Table 1.** List of CCSM4 model simulations and the forcing used in the experiments. Present-day levels are denoted as pd, pre-industrial as pi, and Eemian (125 ka) as eem, respectively. The orbital parameters are calculated according to Berger (1978). SST and sea ice fields are output of fully-coupled CCSM3 simulations described in Section 2.1. GHG concentrations are fixed at the attributed level. For all simulations, solar forcing, vegetation and ice sheets are held constant at the pre-industrial level.

| Simulation | Orbital parameters | SST/ sea ice | $CO_2$ [ppm] | $CH_4$ [ppb] | $N_2O$ [ppb] |
|---|---|---|---|---|---|
| **Pre-industrial** | | | | | |
| $PI_{lowRes}$ | pd | pi 3° | 280 | 760 | 270 |
| $PI_{highRes}$ | pd | pi 1° | 280 | 760 | 270 |
| **Eemian** | | | | | |
| $EEM_{lowRes}$ | eem | eem 3° | 272 | 622 | 259 |
| $EEM_{highRes}$ | eem | eem 1° | 272 | 622 | 259 |
| $EEM_{LabS}$ | eem | LabS-shift | 272 | 622 | 259 |
| $EEM_{NordS}$ | eem | Nord-shift | 272 | 622 | 259 |

**Table 2.** Definitions of climate anomalies calculated as differences of the simulations presented in Table 1.

| Abbreviation | Calculation | Description |
|---|---|---|
| $EEM\text{-}PI_{lowRes}$ | $EEM_{lowRes}-PI_{lowRes}$ | Eemian minus pre-industrial climate anomaly based on lowRes simulations |
| $EEM\text{-}PI_{highRes}$ | $EEM_{highRes}-PI_{highRes}$ | Eemian minus pre-industrial climate anomaly based on highRes simulations |
| $EEM\text{-}PI_{diff}$ | $EEM\text{-}PI_{highRes}-EEM\text{-}PI_{lowRes}$ $= (EEM_{highRes}-PI_{highRes})-(EEM_{lowRes}-PI_{lowRes})$ | Difference in Eemian minus pre-industrial climate anomaly due to different (highRes vs. lowRes) SSTs and sea ice |
| LabS-shift | $EEM_{LabS}-EEM_{lowRes}$ | Climate anomaly due to idealized Labrador Sea shift |
| NordS-shift | $EEM_{NordS}-EEM_{lowRes}$ | Climate anomaly due to idealized Nordic Seas shift |





**Table 3.** Surface air temperature (SAT) anomalies averaged above the Labrador Sea (LabS), Greenland and the Nordic Seas (NordS) among the various sensitivity experiments compared to the respective control experiment. Bold values indicate anomalies significant at the 5% level based on t-test statistics.

| Simulation | LabS $\Delta$SAT [°C] | | Greenland $\Delta$SAT [°C] | | NordS $\Delta$SAT [°C] | |
| --- | --- | --- | --- | --- | --- | --- |
| | DJF | annual | DJF | annual | DJF | annual |
| EEM$_{LabS}$ | **6.0** | **3.6** | 0.7 | **0.4** | | |
| EEM2$_{LabS}$ | **5.4** | **2.8** | **0.9** | **0.5** | | |
| EEM$_{LabS\ ICE}$ | **5.3** | **2.9** | 0.7 | **0.5** | | |
| EEM2$_{LabS\ ICE}$ | **5.7** | **2.3** | 0.5 | **0.4** | | |
| EEM$_{NordS}$ | | | **3.8** | **2.1** | **4.6** | **3.1** |
| EEM2$_{NordS}$ | | | **3.0** | **2.0** | **3.2** | **2.3** |
| EEM$_{NordS\ ICE}$ | | | **2.2** | **0.9** | **3.8** | **2.0** |
| EEM2$_{NordS\ ICE}$ | | | **1.1** | **0.6** | **2.3** | **1.2** |
| EEM-PI$_{diff}$ | **2.9** | **1.8** | **2.8** | **1.5** | **4.4** | **3.3** |

**Table 4.** Surface air temperature (SAT) and accumulation (P-E) anomalies averaged above central Greenland among the various sensitivity experiments compared to the respective control experiment. Central Greenland is defined as 70–77°N, 35–45°W covering the summit area that includes the pNEEM, NGRIP and GRIP ice core sites. Bold values indicate anomalies significant at the 5% level based on t-test statistics.

| Simulation | Central Greenland annual $\Delta$SAT [°C] | Central Greenland annual $\Delta$(P-E) [%] |
| --- | --- | --- |
| EEM$_{LabS}$ | 0.1 | 3 |
| EEM2$_{LabS}$ | 0.2 | 2 |
| EEM$_{LabS\ ICE}$ | 0.2 | 3 |
| EEM2$_{LabS\ ICE}$ | 0.3 | 5 |
| EEM$_{NordS}$ | **2.3** | **12** |
| EEM2$_{NordS}$ | **2.3** | **10** |
| EEM$_{NordS\ ICE}$ | **0.8** | 2 |
| EEM2$_{NordS\ ICE}$ | **0.6** | 1 |
| EEM-PI$_{diff}$ | **1.6** | 5 |