# Peer review of "Warm Greenland during the last interglacial: the role of regional changes in sea ice cover"

_Climate of the Past, 2016_

## Referee Comment (RC1) · P. Bakker (Referee) · 24 Feb 2016

Merz et al. present an interesting study that for the first time quantifies the possibly important role of North Atlantic sea-ice changes, and there with the sea-ice sensitivity, in the last interglacial. This sea-ice sensitivity could to a large extend explain the model-data mismatch in terms of last interglacial Greenland temperatures, as well as explain large inter-model differences in simulated last interglacial climate changes at the high latitudes of the Northern Hemisphere. The methodology and analysis are well thought through and the manuscript well written. I suggest publishing this manuscript in climate of the past after minor revisions.

Main comments:

Main comment 1:

The manuscript shows that differences in simulated North Atlantic SST and sea-ice cover patterns are important to explain reconstructed Greenland temperature anomalies as well as inter-model differences in terms of simulated last interglacial temperatures. It does not attempt to explain the origin of these SST and sea-ice cover differences, which would likely be a whole study on its own. However, in my view this topic cannot be fully ignored and should at least be introduced and its potential implications discussed. Questions that arise are for instance:

What are the causes of the large SST and sea ice differences between the two versions of CCSM3? Yeager et al. show that under pre-industrial boundary conditions there are important differences in the simulated northward oceanic heat transport between the low and high resolution versions of CCSM3. These findings could be shortly summarized here. Can it be deduced which model version is closer to observations in terms of the simulated pre-industrial North Atlantic ocean circulation?

Are the inter-model differences also visible in figure 4 of Lunt et al.? And is the cold bias described here for the low resolution version also the cause of the comparatively low CCSM3 temperatures (winter and annual mean) in the transient last interglacial results (see Bakker et al. 2013, 2014) for the Northern Hemisphere? If so, both could be pointed out in the manuscript.

One could think that a bias in the climate can be accounted for by looking at the anomaly of last interglacial temperatures with respect to a pre-industrial simulation. How does the bias impact the last interglacial climate? Is also the sensitivity of the overturning more sensitivity to global warming, thus leading to cooling in the North Atlantic under last interglacial forcings?

Main comment 2:

The experiments successfully show the role of sea ice and SSTs in explaining the differences between two versions of the CCSM3 model, and provide a potential mechanism that can yield additional warming over Greenland. However, it does not give more

warming over Europe, something that is mentioned a couple of times in the manuscript. Please come back to this point at the end of the manuscript. Questions that come to mind are for instance:

What does it imply that the model-data temperature mismatch over Europe is not improved when using a model with a more sensitive sea-ice cover? Is there another mechanism or feedback missing? Maybe even a mechanism that can explain both the warming over Greenland and Europe without the need for a larger sea-ice retreat? Please shortly discuss this in the manuscript.

Main comment 3:

An important difficulty of last interglacial climate research is the relatively small number of well resolved temperatures and, especially, sea-ice reconstructions. Does the Holocene thermal maximum possibly provide an analogue that can inform us about what happened during the last interglacial because of higher data availability and the existence of sea-ice reconstructions?

Minor comments:

General 1: It is a rather long manuscript, so perhaps the reader can be helped a little more to keep track of the aims and line of the manuscript by shortly repeating those aims and or by providing sort summaries at different points in the manuscript.

General 2: The potentially important role of sea-ice changes in the North Atlantic in the last interglacial climate have been suggested previously, in relation with observations from Greenland ice cores (Sime et al., 2013) and with large inter-model differences in simulated annual mean and winter temperatures (Bakker et al., 2013). It would be good to mention this in the introduction.

Line 7 page 1: 'thus', not everyone is familiar with this model-data mismatch, shortly introduce it in the abstract.

Line 12 page 1: 'accumulation', this is not mentioned before in the abstract and thus

appears a little disconnected from the previously discussed issues.

Page 2: More work on the last interglacial and simulated temperatures over Greenland has been done previously, consider discussing that work, for instance by Loutre et al., Goelzer et al., Bakker et al. and Sanches-Goni et al. and Govin et al.

Line 19 page 1: As you are probably aware, the term Eemian is used to describe a pollen-based warm period in Europe, the regional continental equivalent of the general last interglacial period. Consider using last interglacial instead of Eemian throughout the manuscript.

Lines 3-6 page 2: These lines seem to suggest that proxies can resolve, annual, summer and winter temperature changes for the last interglacial. Please clarify.

Line 7 page 2: What 'Eemian proxies' is referred to here? From what region? Please provide references.

Line 31 page 2: Consider referring to Capron et al. and Govin et al.

Lines 29-33 page 2: What season is discussed here? Is it possible that the winters where warmer, but still the winters were not and neither was the accompanying sea-ice cover decreased?

Line 3 page 6: Why is a 2m thick sea-ice cover used? What are the potential implications of this assumption, please discuss.

Line 3 page 9: It would be helpful for the reader if the 125ka external forcings (GHG and orbital) and their impacts are shortly described (perhaps in the method section), in terms of their annual mean and also seasonal impact.

Line 13-14 page 9: Perhaps an order of magnitude difference can be given to illustrate the dominant role of the turbulent fluxes over the radiative fluxes.

Line 4 page 10: Perhaps at this point come back to the large inter-model spread suggested by previous work (Lunt et al., Otto-Bliesner et al., Nikolova et al. and others) to

put the findings in a bigger picture as an introduction to the next section.

Line 21 page 11: So what are the SATs discussed before if not 'lowest terrain-following level?

Line 13 page 12: Is the feedback by clouds also small over the Nordic Seas?

Line 3 page 13: Earlier on, when winter changes are discussed, mention that seasonality will be covered later.

Line 12 page 15: Are these SATs for Greenland averages over the whole of Greenland (and also in Figure 11E)?

Line 21-23 page 16: Consider repeating what EEM-PIdiff stands for to make this point more clear.

Line 15 page 17: Consider giving the ages covered by the NEEM core.

Line 17 page 17: Give distance between NEEM and pNEEM to give the reader an idea of the difference.

Lines 29-32 page 17: It is not clear how this connects to the topic of this manuscript, please clarify.

Line 7 page 18: Give range of temperature estimate. Is this number altitude corrected? This seems relevant with the discussion later on.

Line 15 page 18: Is this 3.1K because of elevation changes, circulation changes? Please shortly summarize. What about other work on this topic by for instance Stone et al., Langebroeck et al. and Fyke et al.?

Line 34 page 18: Be more specific about what 'climate change' means here.

Line 2 page 19: What about changes in the seasonality of precipitation?

Line 11 page 19: Is this for specific regions? Please clarify.

[Figure]

Lines 24-26 page 19: Make clear that this combined experiment has in fact not been performed.

Figure 2: So does this indicate that the atmosphere is of little importance in determining the LIG climate response to the orbital forcing? What about the role of vegetation?

Figure 3: The patterns are very different for the high and low resolution model runs. Does this point to an important role of differences in ocean dynamics?

Figure 3: Why is there no EEM-PI-diff row in this figure?

Figure 3: Why are the patterns in SST so different from the SAT (Figure 4) patterns for, for instance, the Arctic region?

Figure 8d-e: There appears to be a dipole kind of structure over Greenland for HTdyn-core and HTpar. Why is that and how are they related to the large scale wind changes?

Technical comments:

Line 3 page 1: Perhaps 'Northern Hemisphere high latitudes'.

Line 11 page 1: Perhaps 'Nordic Seas sea ice retreat'.

Line 14 page 3: The line 'Thereby the authors...' seems a little redundant and could be removed.

Line 3 page 5: Remove 'it' and put comma after 'Sect. 4'.

Line 16 page 5: A new type of idealized.

Line 10 page 6: Consider replacing 'cutting through' with 'in'.

Line 22 page 7: Twice CCSM4, should one of them be CCSM3?

Line 17 page 11: HTdyn-core is not a very descriptive acronym. Consider using something else that makes it clearer that it deals with resolved heat transport.

Line 19 page 11: Giving the field names is perhaps not necessary.

Line 19-20 page 11: Consider rewording to "Note also that all simulations are run into equilibrium, so the total temperature tendency (dTdt) is almost zero.

Lines 13-12 page 17: words 'which are mostly drilled on top of the Greenland ice sheet', is not very relevant, consider removing.

Line 3 page 18: Refer to table 4.

Lines 12-14: Difficult to read, please reword.

Line 22 page 18: Should instead of shall.

Line 32 page 18: Remove 'Thereby'.

Line 34 page 18 to line 2 page 19: Difficult to read, please rephrase.

Figure 1: Explain meaning of solid versus stippled green boxes.

Figure 2: Mention ones, here or in main text, how significance level is determined. Using yearly averaged time series?

Figure 6: Continents are either white or grey in the different panels.

Figure 7: Perhaps a personal preference, but I like better the colour scales that have white around the zero value (for instance figure 5).

Figure 10: Perhaps in panels b and d remove the vectors if they are not significant.

Figure 12: Indicate on a map (perhaps in figure 1) where the NEEM or pNEEM site is located.

Figure 12: Indicate significance of simulated temperature changes.

Table 1: Why are the other sensitivity tests not included?

Table 3: Perhaps a printing issue on my side, but the bold letters are very difficult to distinguish.

Table 3 and 4: Using different regions for Greenland (whole island, central Greenland or pNEEM) is a little confusing and perhaps not necessary.

---

## Referee Comment (RC2) · Anonymous Referee #2 · 25 Feb 2016

The authors analyze the role of sea ice and SST anomalies in the Labrador and Nordic seas in controlling surface air temperature anomalies over the North Atlantic region (with a special focus on Greenland) during the Last Interglacial (LIG). Using the atmosphere component of CCSM4, a state-of-the-art climate model, a set of sensitivity experiments was performed to disentangle the influence of the Labrador Sea versus the Nordic Seas. The results were analyzed very carefully and in much detail considering heat and moisture budgets. It is found that sea ice retreat and warming in the Nordic Seas is crucial for the simulation of high Greenland temperatures during the LIG, which are evidenced by proxy records, whereas the role of the Labrador Sea is minor. The paper is well written and clearly structured. Although similar experiments and ideas have been published before by Li et al. (2010) with a focus on the last glacial, the results by Merz et al. are novel and show the importance of Nordic Seas ice cover

for the LIG. As such, the study by Merz et al. is certainly of interest for the paleomodelling community and suitable for Climate of the Past. However, the following points have to be taken into account before publication of the study.

1) p. 1, line 11: "Diabatic processes play a secondary role". This statement is confusing. The simulated SAT anomalies are ultimately caused by anomalous surface energy fluxes, e.g. sensible heating, which is a diabatic process. I think the authors refer to latent heating and radiative processes. Please be more precise.

2) p. 1, line 23: In both models and data the LIG warming is mostly restricted to the extratropics, whereas the tropics show cooling in many regions. Again, please be more precise.

3) p. 2, line 13: The transient CCSM3 simulation used in this study was not part of the paper by Lunt et al. (2013). The CCSM3_Bremen simulation in Lunt et al. (2013) is a time slice (125 kyr BP) run using the T31-version of CCSM3. It is different to the transient simulation by Varma et al. (2015). Please clarify.

4) p. 3, line 5: In addition to the papers by Li et al. (2005, 2010), cite the study by Zhang et al. (2014), which strongly supports the findings by Li et al., but in a fully-coupled setup.

5) p. 4, line 1: In addition to Varma et al. (2015), cite the studies by Bakker et al. (2013) and Govin et al. (2014), where the transient CCSM3 LIG simulation has been published first.

6) p. 4, line 5: The two realizations do not only differ in horizontal resolution. Note that different greenhouse gas concentrations have been used as well as a different solar constant. Moreover, the transient character of the low-resolution run as well as the short integration time of the high-resolution time slice simulation should be taken into account. Please rephrase.

7) p. 5, line 26: How was the decision made on how far the sea ice margin is shifted

to the north? Is it based on the high-resolution LIG simulation or is it arbitrary? Please explain.

8) p. 6, line 31: The authors have not used the CCSM3_Bremen simulation from Lunt et al. (see above).

9) p. 7, line 1: As mentioned above, the difference is not only due to horizontal resolution. Different GHG concentrations have been used. In particular, N2O concentration is much higher in the high-resolution CCSM3 experiment than in the low-resolution run. Moreover, a higher solar constant (1367 W/m2) has been used in the high-resolution experiment.

10) p. 7, line 2: Vegetation is fixed (modern) in the transient CCSM3 low-resolution run.

11) p. 7, line 11: As mentioned above, higher N2O and solar constant contribute to the warming in the high-resolution CCSM3. I agree that the ocean is also a likely candidate. In fact, as shown in Bakker et al. (2013) the AMOC in the transient low-resolution CCSM3 simulation is relatively weak. Reduced oceanic heat transport would contribute to the relatively cool conditions in the North Atlantic. In addition, it should be noted that the pre-industrial reference run by Merkel et al. (2010) has much higher GHG concentrations than the transient LIG simulation (in particular CH4).

12) p. 19, line 30: The study by Zhang et al. (2014) may be cited here, showing that processes are similar in coupled and uncoupled (Li et al., 2010) experiments.

13) p.36, Table 1: A reference is missing for the chosen GHG values.

---

## Referee Comment (RC3) · Anonymous Referee #3 · 14 Mar 2016

**Review of:**
Merz et al. Warm Greenland during the last interglacial: the role of regional changes in sea ice cover

**Summary:**
This papers investigates the potential importance of SST and sea ice for the climate conditions in the North Atlantic and Greenland in the Eemian interglacial. Simulations are conducted with CAM3 and CAM4, comparing the pre-industrial (PI) and Eemian

climates using prescribed sea-surface conditions from fully coupled simulations (at different resolutions) with CCSM3. The main conclusion is that sea ice in the North Sea region can have a large impact on the Greenland climate and a reduction of its prevalence generates a substantial warming over the ice sheet. The sea ice in the Labrador Sea is important for the local climate conditions but has a little to no impact on the Greenland climate. The authors conclude that the climate impact is mostly mediated by near surface turbulent fluxes that influence the atmospheric circulation and thereby cause a warming over the ice sheet. The paper is generally well written and is suitable for Climate of the Past, though first after a substantial revision.

**Major comments:**

**1. Model validation and motivation**
(i) In all modeling studies it is mandatory to prove that the model is capable of producing a reasonable climate that conforms to observations or proxy data records (climate reconstructions) when studying past climates. This is a first sanity check that tells the reader that it might be worth while spending the time end energy reading the paper. This manuscript only contains difference fields and the reader is never shown the actual climatological states. I suggest adding a figure showing a comparison of the pre-industrial (PI) simulation with either a reanalysis product or a reliable climate reconstruction (show full fields and how they differ from observations). For the Eemian you can compare with proxy data where such are available. Though this type of comparison is mandatory, in this study it is extra important since the model seems to be sensitive to the horizontal resolution.

(ii) I would like to see a better motivation of the study. What is the goal (what do we wish to learn) and why are we interested in this particular problem? The current motivation seems to be that fully coupled models simulate different sea-surface temperature (SST) and sea-ice cover (SIC) in the Eemian. This is perhaps not too surprising given the large model spread in simulations of both present and future climates. It would be better to motivate the study from available proxy data records from ice cores as well as

terrestrial and marine records. Given the large model spread, what makes this model better than any other model and can we trust the results presented here (connected to the model validation)? You can also extend the motivation by looking at AMOC in different models and connect that to differences in the sea-surface conditions.

**2. Modeling approach**

(i) Initially you show that the low and high resolution models yield different results in terms of SST and sea ice in the North Atlantic. It is further mentioned that the low resolution model has known problems and does not simulate a reasonable PI climate in the North Atlantic sector (is this also true for the Eemian?). Despite this claim, the majority of the experiments and figures (according to Table 2) are based on results from the low resolution model. This seems like a very odd choice to me. If the model is biased and has known problems, why base almost all figures and analysis on data from this model? Are there even worse problems associated with the high resolution model? If not, can we expect different conclusions if the same analysis is performed on the high resolution data?

(ii) I am generally skeptical to the approach taken in sections 4.1 and 4.2 and I am afraid that we are not learning very much from this exercise. CCSM3 and CCSM4 are highly dependent models (e.g. Knutti et al., 2013) that are part of the same model family, meaning that the atmospheric components (CAM3 and CAM4) share the majority of the same code base. The biggest difference between the models is the deep convection scheme, which plays virtually no role in the latitude range of your focus. Consequently, the comparison of the two atmospheric models is largely redundant as you basically compare results from two simulations with almost the same model using identical forcing protocols. I argue that you can omit this whole comparison and just state that you use SST/SIC from CCSM3 in CAM4 and then prove that the simulated climates are reasonable with respect to reliable data. Also, the near surface temperature is not the best field to use to evaluate differences between AMIP simulations. If the model is capable of producing a realistic climate with realistic turbulent fluxes (e.g. near

surface gradients), the near surface temperature is by definition largely similar to skin temperature and you basically prescribe the phenomena that you are investigating.

(iii) A large part of the analysis is based on differences between difference fields (EEM-$PI_{diff}$). These results are almost impossible to wrap ones head around and I wonder what we can learn from such a comparison, especially since the low resolution model has known biases. Also, it would help the interpretation of the results if you used the same color scale in all figures showing the same/similar quantities.

(iv) My main concern has to do with the sea-ice retreat experiments. First of all, the amount by which you shift the sea ice seems to be arbitrarily chosen and should be motivated. Second, I am not convinced that these perturbation experiments are designed in a way that they will teach us anything useful about the last interglacial climate. In steady state (no drift due to external forcing) the circulation in atmosphere and ocean is by definition what determines the sea-surface conditions; the SST/SIC is essentially determined by the internal heat flux (Qflux) in the ocean mixed layer and the balance between radiative and turbulent surface fluxes in the atmosphere (SST $\sim$ SWnet – LWnet – LHflux – SHflux – Qflux). When you prescribe the sea-surface conditions and introduce local changes in the SST/SIC, you also introduce a local climate forcing that could never happen in the real world as it is not supported by the rest of the climate system (the open water that is introduced is not consistent with the general circulation).

If we assume that the sea-ice cover in the Labrador sea collapsed (for whatever reason), the climate system would do everything it can to rebuild the sea ice over the next few seasons (as is evident from the almost $100\,\mathrm{W\,m^{-2}}$ imbalance in sensible and latent heat fluxes that are reported in the analysis). If we instead assume that we could collapse the Labrador sea ice and keep the region ice free, the rest of the ocean circulation (and atmospheric circulation for that matter) would have to be different to sustain the reduced sea ice; i.e. there would be changes in the SST field elsewhere and the turbulent fluxes would almost certainly be lower as sea-ice otherwise would form. I know that the chosen modeling approach is not new and that other people have done

similar experiments before you (e.g. Deser et al., 2010), but I am concerned that this modeling approach does more damage than good in this particular study. I don't have a patented solution to the problem but I argue that it would be better to run a slab ocean model and alter the internal heat flux convergence in the mixed layer (in a conservative way so it doesn't introduce a global climate forcing) so that the sea ice retreats from the desired regions. This is arguably a better solution as the surface temperature and sea-ice margin are determined by the surface energy balance, which means that it is theoretically possible to construct a climate where there is no sea ice in the desired regions but you have sea-surface conditions that are in balance with the circulation and external forcing. Whether or not this climate state is realistic is of course another question.

**3. Interpretation of results**

(i) Following the previous comment, it is not at all surprising that you get very strong turbulent fluxes in the sea-ice sensitivity experiments. The prescribed SST/SIC implies that the climatological atmospheric circulation is more or less determined by the prevailing sea-surface conditions. When making local changes to the SIC and prescribe SSTs that are not consistent with the circulation, you introduce regions where the climate "wants" to have sea ice, as cold air is advected over open water, but the prescribed sea-surface conditions prevents it from forming. This gives rise to artificial vertical gradients and turbulent fluxes that would never happen in nature as the SST/SIC would respond and go back to an ice covered state. This in turn induces and anomalous atmospheric circulation that has no real world analogue, at least not in a climatological state which is what is investigated here.

(ii) In my mind, one of the most interesting results in the whole paper is the changes in the lower tropospheric wind field (Fig. 9) that results from manipulating the local SST/SIC in the North Sea. However, no explanation is provided as to why the wind field changes the way it does. I want to see a dynamic argument made for the somewhat counterintuitive response where the lower tropospheric winds impinge on Greenland

from seemingly the wrong direction; SE instead of NE where the forcing is located.

**Line-by-line comments:**

Page 1, line 1: I would be careful suggesting that the Eemian is a possible analog to the climate in the near future. The Eemian was warm primarily as a result of increased insolation whereas future climates are warm because of higher greenhouse gas concentrations. The former only plays a direct role during parts of the year (in high latitudes) whereas the latter influence the longwave radiation in all seasons.

Page 1, line 19: This time interval contains both warm and cold phases.

Page 4, lines 1 & 3: Write out the equivalent grid resolution for T31 and T85.

Page 4, lines 19-25: This is more of a curious comment than anything else but when you regrid the T31 SST/SIC to the T85 grid, you implicitly introduce an outline of the T31 grid but at the higher resolution. Do you have a feeling for if this will influence the results?

Page 5, line 24: How does the absence of inter-annual variability in the SST/SIC degrade the representation of the stormtrack? Add a sentence explaining that.

Page 6, line 7: The $-1.8°C$ temperature is only used for the SSTs underneath sea ice. The actual temperature of the sea ice is determined by the local surface energy balance, which is generally much lower. It is therefore a bit misleading to use the SST as a measure of the surface temperature and I suggest showing the actual surface temperature instead.

Page 7, and of section 4.1: Determine whether the difference in temperature signal is due to the PI, Eemian or both climate states when going to the lower resolution.

Page 7, line 12: with and an excessive... -> with an excessive...

Page 7, line 22: CCSM4 and CCSM4 -> CCSM3 and CCSM4

Page 8, line 1: What is the relationship between the SST and the sub-polar gyre?

Page 8, line 11: Show the PI SST, it is important for the story!

Page 8, line 18: particularly strong on SAT above oceanic grid cells... Don't you use identical SST/SIC in CAM3 and CAM4? If so, you expect to see very similar SAT as it represents the temperature just above the ocean surface.

Page 8, line 19: How much is the winter insolation decreased in winter?

Page 9: What can we possibly learn from $\Delta(\Delta_1 - \Delta_2)$ when at least one of the $\Delta_{\#}$s have known biases?

Page 9, line 6: surface ocean -> ocean surface

Page 9, line 11: Which terms does Qnet contain? Radiative fluxes? Turbulent fluxes? Internal heat sources in the ocean? A combination of all or a subset of the above?

Page 9, lines 21-29: You have prescribed SST, which means that you easily get artificial turbulent surface fluxes as the ocean temperature acts as an infinite source and sink of energy (sign depends on atmospheric conditions).

Page 10, line 5: Write out the resolution used in the "Shift" experiments.

Page 10: Why do you use the low resolution model when it has known biases?

Page 10: Fixed SST is almost certainly the source of the strong turbulent fluxes that are highly artificial as they would never happen in nature in the way described in the manuscript, at least not over a long period of time.

Page 11, lines 1-3: Why does the warming spread over Greenland? Comment on changes in atmospheric circulation.

Page 11, line 8-10: Eq. 1 is written in advective form, not flux form. The terms you refer to are therefore showing temperature advection and not heat flux convergence.

Page 11, lines 16-19: Are you talking about month to month variability or the climatology? The terms have to be identically equal to zero in the latter if the model is in balance.

Page 11, line 20: The temperature tendency has to be identically zero for the model to be in balance. You are looking at a climatology after all, or...?

Page 12, line 6: How much is actually resolved at T31?

Page 12, line 13: How does that hang together with the enormous increase in LH flux? I would expect to see a great moistening of the atmosphere when the LH flux increases that much, which in turn increases the cloudiness.

Page 12, line 33-Page 13, line 9: This paragraph is very confusing because you first talk about what you expect to see and then you show that the expected circulation is in fact not true.

Page 12: What happens to mid- and upper tropospheric winds in these experiments?

Page 13, line 20: I don't see a southeastward transport in the figure.

Page 13, lines 20-23: Is this also true in these experiments? Have you done the proper analysis or is it just a conjecture?

Page 14, lines 15-19: This is the heart of my concern. Everything in the climate system acts to build sea ice where it has been removed but the prescribed SST/SIC don't allow the sea ice to regrow. Since the summer temperature is higher, there will not be any regrowth in the summer season and you don't see equally outrageous turbulent fluxes.

Page 14, lines 27-34: This is not very surprising either. There is a prevailing southwesterly flow over the northeastern Atlantic, meaning that warm and moist air is advected over the region where you remove the sea ice. There is thus a smaller "urge" for the climate system to regrow sea ice there and you don't see equally large turbulent fluxes.

Page 17, line 34: "statistically insignificant warming" sounds strange. Rewrite the sentence in a way that allows you to use something like "not significantly significant".

Page 18, line 9-10: Have you adjusted the Greenland elevation in these simulations?

Page 18, line 15: A 3.1°C temperature difference could in principle be due to a lowering of the ice sheet. Since the sea level was quite a bit higher in the Eemian, this is not a bad first guess that could be explored in a greater detail in the manuscript.

Page 18, lines 29-34: This section is a bit speculative. Maybe you can extend the discussion to include the importance of precipitation seasonality and the temperature inversion relationship recently discussed by Pausata and Löfverström (2015).

Page 19, lines 20-23: You haven't really shown or discussed any proper atmospheric dynamics in this paper. The main focus is on the turbulent fluxes that no doubt will influence the atmospheric circulation. This has not been shown properly though so this statement is merely a conjecture.

Figures: Use the same colorscale in all figures showing the same/similar quantities.

Figure 1: Consider changing the transect to a different color. It is very hard to see black on top of dark blue.

Figure 2: Validate the model by showing full fields as well as a climate reconstruction.

Figure 3: The large sensitivity of SIC to the model resolution is curious. Is there any proxy data you can compare this with?

Figure 3: What is the purpose of this figure when Fig. 4 shows almost exactly the same thing, though extended to show the response over land as well?

Figure 5: Number labels have not been defined.

Figure 10: I am curious as to why there are such large differences in e.g. the Norwegian Sea and southwestern Greenland?

Table 3: Write out the abbreviations and resolutions in the caption.

**References**

Deser, C., Tomas, R., Alexander, M., and Lawrence, D.: The seasonal atmospheric response to projected Arctic sea ice loss in the late twenty-first century, Journal of Climate, 23, 333–351, 2010.

Knutti, R., Masson, D., and Gettelman, A.: Climate model genealogy: Generation CMIP5 and how we got there, Geophysical Research Letters, 40, 1194–1199, 2013.

Pausata, F. S. R. and Löfverström, M.: On the enigmatic similarity in Greenland $\delta$18O between the Oldest and Younger Dryas, Geophysical Research Letters, 42, doi:10.1002/2015GL066042, 2015.

---

## Author Comment (AC1) · 17 May 2016

**Response to reviewer #1**

General:

Merz et al. present an interesting study that for the first time quantifies the possibly important role of North Atlantic sea-ice changes, and there with the sea-ice sensitivity, in the last interglacial. This sea-ice sensitivity could to a large extend explain the model data mismatch in terms of last interglacial Greenland temperatures, as well as explain large inter-model differences in simulated last interglacial climate changes at the high latitudes of the Northern Hemisphere. The methodology and analysis are well thought through and the manuscript well written. I suggest publishing this manuscript in climate of the past after minor revisions.

We thank the referee for the careful review and the constructive comments. Please find the answers to all specific comments below. We have not responded yet to the technical corrections (wording etc.) but will do so when preparing a revised manuscript.

Main comment 1:

The manuscript shows that differences in simulated North Atlantic SST and sea-ice cover patterns are important to explain reconstructed Greenland temperature anomalies as well as inter-model differences in terms of simulated last interglacial temperatures. It does not attempt to explain the origin of these SST and sea-ice cover differences, which would likely be a whole study on its own. However, in my view this topic cannot be fully ignored and should at least be introduced and its potential implications discussed. Questions that arise are for instance:

What are the causes of the large SST and sea ice differences between the two versions of CCSM3? Yeager et al. show that under pre-industrial boundary conditions there are important differences in the simulated northward oceanic heat transport between the low and high resolution versions of CCSM3. These findings could be shortly summarized here. Can it be deduced which model version is closer to observations in terms of the simulated pre-industrial North Atlantic ocean circulation?

Both the low and the high resolution versions of CCSM3 have known deficiencies in its representation of Arctic sea ice and heat transport in the Atlantic Ocean (Collins et al, 2006 and Yeager et al., 2006). In particular, the low resolution CCSM3 has a too extensive sea ice cover and an underestimated ocean heat transport. The sea ice cover is smaller and thinner in the high resolution version, which is closer to observations. On the other hand, the high resolution CCSM3 still has a pronounced cold anomaly in the subpolar North Atlantic compared to observations (Collins et al., 2006).

Large and Danabasoglu (2006) devote a whole study to the attribution and impacts of upper-ocean biases in the high (and medium) resolution CCSM3. The study shows that too strong surface winds are likely one reason. Besides, the biases in upper-ocean temperature and salinity along ocean basin boundaries relate to problems in the representation of ocean upwelling.

We agree that these are important points and we will include this information in the revised manuscript.

Are the inter-model differences also visible in figure 4 of Lunt et al.? And is the cold bias described here for the low resolution version also the cause of the comparatively low CCSM3 temperatures (winter and annual mean) in the transient last interglacial results (see Bakker et al. 2013, 2014) for the Northern Hemisphere?

The "error" of the high and low resolution pre-industrial control simulations compared to NCEP (Fig. 4 in Lunt et al., 2013) shows a cold bias in the North Atlantic for both cases. Partly due to the chosen color scale in Fig.4 in Lunt et al., 2013 it is not apparent which of the cold bias is stronger but the high resolution bias seems more spatially extensive. Nevertheless, Fig. 4 in Lunt et al., 2013 nicely illustrates that the high and low resolution versions of CCSM3 produce quite different SAT patterns (globally) and thus should be regarded as different climate models even though they base on some common model physics.

We will make an effort to make this last point clearer in the manuscript

Furthermore, we don't' believe that the cold bias described for the low resolution version necessarily is the cause of the comparatively low CCSM3 temperatures (winter and annual mean) in the transient last interglacial results (see Bakker et al. 2013, 2014). Note that the latter are low CCSM3 temperatures for the last interglacial with respect to pre-industrial. Hence, this "relative" cooling of the last interglacial CCSM3 has to be clearly distinguished from the cold model biases found for absolute present-day/pre-industrial temperatures.

The CCSM3 EEM-PI cooling found in Bakker et al., 2013/2014 bases on the same simulation (conducted by people from University of Bremen) as the the low resolution CCSM3 simulations shown in our manuscript. Hence, our analysis of those simulations shows that the atmospheric cooling in the last interglacial is explained by concurrent sea ice growth and cooling SSTs (see Fig. 3) which likely bases on a reduced oceanic heat transport. The latter seems to be the model's response to the Eemian external forcing (we cannot think of a mechanism why it should link to the model bias, i.e., an already underestimated ocean heat transport simulated for present-day/pre-industrial). Reviewer # 2 correctly pointed out that the CCSM3 low resolution pre-industrial run by Merkel et al. (2010) has higher GHG concentrations than the transient Eemian simulation (in particular $CH_4$), which likely fosters the relatively cold Eemian temperatures. Further, one can speculate that the model might be more sensitive than other models (including the high resolution CCSM3) to the decrease in winter insolation resulting from the Eemian orbital forcing.

If so, both could be pointed out in the manuscript. One could think that a bias in the climate can be accounted for by looking at the anomaly of last interglacial temperatures with respect to a pre-industrial simulation. How does the bias impact the last interglacial climate? Is also the sensitivity of the overturning more sensitivity to global warming, thus leading to cooling in the North Atlantic under last interglacial forcings?

Please see the answers above.

We will provide more details on the biases of the two versions of CCSM3 in the revised manuscript. We will also clarify our motivation to investigate the impact of these biases on the uncertainty of last interglacial temperatures over Greenland. With regard to further interpretation of the origin of the biases and implications for the stability of the overturning circulation during the last interglacial or under global warming, we feel that this would probably be too speculative and that a comprehensive discussion exceeds the scope of this study.

Main comment 2:

The experiments successfully show the role of sea ice and SSTs in explaining the differences between two versions of the CCSM3 model, and provide a potential mechanism that can yield additional warming over Greenland. However, it does not give more warming over Europe, something that is mentioned a couple of times in the manuscript. Please come back to this point at the end of the manuscript. Questions that come to mind are for instance:

What does it imply that the model-data temperature mismatch over Europe is not improved when using a model with a more sensitive sea-ice cover? Is there another mechanism or feedback missing? Maybe even a mechanism that can explain both the warming over Greenland and Europe without the need for a larger sea-ice retreat? Please shortly discuss this in the manuscript.

We believe that a retreating sea ice cover is one important mechanism to explain the Eemian warmth but it does not exclude other influences. The finding that temperatures over Europe are largely insensitive to changes in the sea ice cover illustrates this point well. We will revise the manuscript to clarify that the large biases in the representation of sea ice in CCSM3 and other climate models complicate the quantification of the impact of other variations.

Main comment 3:

An important difficulty of last interglacial climate research is the relatively small number of well resolved temperatures and, especially, sea-ice reconstructions. Does the Holocene thermal maximum possibly provide an analogue that can inform us about what happened during the last interglacial because of higher data availability and the existence of sea-ice reconstructions?

Reconstructions of sea ice are generally rare for all paleoclimatic epochs including the mid Holocene. The intent of our study was not to propose the most likely sea ice simulation for the last interglacial, but to highlight how the uncertainty in the ice cover of periods in the past propagates into the estimates of Greenland temperatures.

Minor comments:

General 1: It is a rather long manuscript, so perhaps the reader can be helped a little more to keep track of the aims and line of the manuscript by shortly repeating those aims and or by providing sort summaries at different points in the manuscript.

We agree with the referee that we should help the reader not to lose track in the rather long manuscript. We thus intend to include short repetitions of the study aims at the beginning of Sections 4.1, 4.2 and 5.

General 2: The potentially important role of sea-ice changes in the North Atlantic in the last interglacial climate have been suggested previously, in relation with observations from Greenland ice cores (Sime et al., 2013) and with large inter-model differences in simulated annual mean and winter temperatures (Bakker et al., 2013). It would be good to mention this in the introduction.

Thank you for bringing these papers to our attention. We will investigate their findings in detail and include respective references in the revised manuscript if applicable.

Line 7 page 1: 'thus', not everyone is familiar with this model-data mismatch, shortly introduce it in the abstract.

The abstract will be revised to make this clearer.

Line 12 page 1: 'accumulation', this is not mentioned before in the abstract and thus appears a little disconnected from the previously discussed issues.

We consider revising the abstract to better introduce moisture and accumulation processes.

Page 2: More work on the last interglacial and simulated temperatures over Greenland has been done previously, consider discussing that work, for instance by Loutre et al., Goelzer et al., Bakker et al. and Sanches-Goni et al. and Govin et al.

We will revise the introduction to account for these studies.

Line 19 page 1: As you are probably aware, the term Eemian is used to describe a pollen-based warm period in Europe, the regional continental equivalent of the general last interglacial period. Consider using last interglacial instead of Eemian throughout the manuscript.

We consider replacing the term „Eemian" by last interglacial although we feel that „Eemian" is a widely accepted term in the paleoclimate scientific community. Using the term "Eemian" is more in line with our previous studies (Merz et al., 2014a,b) which can be regarded as companion papers also focusing on the climate in/around Greenland during this time period.

Lines 3-6 page 2: These lines seem to suggest that proxies can resolve, annual, summer and winter temperature changes for the last interglacial. Please clarify.

We will revise the aforegoing sentence to make this clear. The seasonality issue rather relates to the models than to the proxies which can provide information about the temperature seasonality of the Eemian.

Line 7 page 2: What 'Eemian proxies' is referred to here? From what region? Please provide references.

We will add references to Turney & Jones (2010) and Capron et al., 2014.

Line 31 page 2: Consider referring to Capron et al. and Govin et al.

We will add the according references.

Lines 29-33 page 2: What season is discussed here? Is it possible that the  summers? where warmer, but still the winters were not and neither was the accompanying sea-ice cover decreased?

Axford et al., 2011 refers to summer temperatures whereas Bauch et al. 2012 does not refer to a single season. We are further not aware of temperature reconstructions for the winter season for the last interglacial in this area. In the low resolution CCSM3 we see that Eemian winters were colder and sea ice was rather expanding (likely due to the negative orbital forcing in NH winters) but again this model seems in contrast with many other climate models which generally show a stronger warming for the last interglacial (e.g., see Bakker et al., 2013, Lunt et al., 2013). Hence, we can hardly do more than speculate on the last interglacial state of the NH sea ice, particularly for the winter season.

Line 3 page 6: Why is a 2m thick sea-ice cover used? What are the potential implications of this assumption, please discuss.

2 meter sea ice thickness is standard for all CCSM3/CCSM4 atmospheric simulations with prescribed sea ice cover and there is no choice on that in the state-of-the art configurations of the (atmospheric) CCSM simulations. We cannot really comment on this standard but it corresponds to the observed sea ice thickness in the NH although there is quite a range in sea ice thickness (0-5m), e.g., based on recent CryoSat-2 measurements.

Hence, we haven't tested the sensitivity of sea ice thickness on the Arctic climate. Please refer to Holland et al., 2006 for a respective study. We will add this reference and a statement that the sea ice thickness is not tested in this study.

Line 3 page 9: It would be helpful for the reader if the 125ka external forcings (GHG and orbital) and their impacts are shortly described (perhaps in the method section), in terms of their annual mean and also seasonal impact.

We agree that this information should be included in the manuscript and we will revise the text accordingly.

Line 13-14 page 9: Perhaps an order of magnitude difference can be given to illustrate the dominant role of the turbulent fluxes over the radiative fluxes.

We will add respective estimates which are of the order of 10-20 $W/m^2$ (LWnet) and up to 150 $W/m^2$ for SHF and LHF.

Line 4 page 10: Perhaps at this point come back to the large inter-model spread suggested by previous work (Lunt et al., Otto-Bliesner et al., Nikolova et al. and others) to put the findings in a bigger picture as an introduction to the next section.

As stated above we will add some sentences here (at the beginning of Section 5) to remind the reader of the goals of the study and the initial problem with the large inter-model spread.

Line 21 page 11: So what are the SATs discussed before if not 'lowest terrain-following level?

The SAT refers to the 2m temperature which is state of the art in most climate models. The 2m temperature is an interpolated diagnostic measure whereas temperature at the lowest terrain-following level conforms to the temperature in the lowest layer of the atmospheric grid. We consider taking out the sentence at (page 11, line 22) as it might confuse the reader.

Line 13 page 12: Is the feedback by clouds also small over the Nordic Seas?

We do find some moderate increase in cloud cover directly above the main SHFLX anomalies in the Nordic Seas. However, we find that all changes in cloud cover and do not lead to significant radiation anomalies and hence are not of crucial importance for the temperature response.

Line 3 page 14: Earlier on, when winter changes are discussed, mention that seasonality will be covered later.

We will add a respective statement at the end of section 4 to advertise the seasonality section:

Line 12 page 15: Are these SATs for Greenland averages over the whole of Greenland (and also in Figure 11E)?

Yes. We add the following statement to the caption of Fig. 11:

*[The Greenland mean SAT refers to the area-averaged SAT of whole Greenland.]*

Line 21-23 page 16: Consider repeating what EEM-PIdiff stands for to make this point more clear.

We revise this paragraph to clarify this issue:

Line 15 page 17: Consider giving the ages covered by the NEEM core.

We don't feel that this adds much clarification here as the full NEEM core actually extends beyond the Eemian but its information from the penultimate glacial is disturbed by folding effects etc. Moreover, our simulations are rather generally valid for an Eemian optimum but do not refer to a specific time period or a transient evolution of the Greenland temperature.

Line 17 page 17: Give distance between NEEM and pNEEM to give the reader an idea of the difference.

We will add the respective information, i.e. that pNEEM is located ca. 300km upstream of NEEM relatively close to the summit of the ice sheet.

Lines 29-32 page 17: It is not clear how this connects to the topic of this manuscript, please clarify.

This statement is included to provide some perspective on our results in the context of contemporary climate change. We also think that the previous sentence that our results are "not limited to the Eemian but very likely valid for any interglacial and glacial climate period" requires this specification to not mislead the reader.

Line 7 page 18: Give range of temperature estimate. Is this number altitude corrected? This seems relevant with the discussion later on.

We prefer to just mention the upper limit of the temperature estimate as we focused on the maximum temperature response in Merz et al., 2014a, i.e. for the simulated minimum in the Eemian Greenland ice sheet volume/extent. Further, the number (3.1K) is altitude corrected what will be clarified in the revised statement (see next point).

Line 15 page 18: Is this 3.1K because of elevation changes, circulation changes? Please shortly summarize. What about other work on this topic by for instance Stone et al., Langebroeck et al. and Fyke et al.?

The full warming effect to explain the 3.1K is due to a series of changes in the low-level winds and eventually the surface energy balance following a change in the Greenland ice sheet topography as discussed in full details in Merz et al., 2014a. We will extend the sentence at page 18, line 15 to make this clearer.

However, we prefer to guide the reader to the reference rather than giving a full summary of the topography-effects as this would further lengthen the already rather extensive discussion section. To our knowledge, the studies mentioned above investigate possible changes in the Greenland ice sheet topography during the Eemian but do not estimate/simulate the associated climate/temperature effect.

Line 34 page 18: Be more specific about what 'climate change' means here.

"Eemian climate change" is changed to "Eemian warming"

Line 2 page 19: What about changes in the seasonality of precipitation?

In Merz et al., 2014b we show that Greenland precipitation is more biased towards the summer season in the Eemian compared to PI. However, Sime et al., 2013 states that uncertainty about local interglacial sea surface conditions, rather than precipitation intermittency changes, may lead to the largest uncertainties in interpreting temperature from Greenland ice cores.

Line 11 page 19: Is this for specific regions? Please clarify.

We revise the statement as follows:

*[These simulations are in better agreement with Eemian SST and SAT proxy records from the NH extratropics.]*

Lines 24-26 page 19: Make clear that this combined experiment has in fact not been performed.

We will revise the statement to make this clear.

Figure 2: So does this indicate that the atmosphere is of little importance in determining the LIG climate response to the orbital forcing? What about the role of vegetation?

Fig. 2 does imply that the ocean and sea ice component are most likely responsible for the spread among different EEM-PI simulations. This does not mean that the atmosphere itself is not reacting to the anomalous orbital forcing but in both CCSM3 model simulations in a rather consistent way. However, as always the pure sensitivity of a single component of the climate system is only to guess from a fully-coupled setup. An experiment with an atmospheric model simulation forced by the anomalous Eemian orbital forcing but pre-industrial sea ice/SSTs might be a possible experiment to answer this question in detail. The vegetation is held to modern values in all CCSM3 experiments (our initial statement that the CCSM3 EEM$_{lowRes}$ simulation used a dynamic vegetation model was actually wrong as correctly pointed out by Reviewer #2 – we will revise it accordingly). Hence, in the CCSM3 simulations shown here vegetation processes are not taken into account and therefore cannot be responsible for the temperature spread seen in EEM-PI$_{diff}$ (Fig. 2).

Figure 3: The patterns are very different for the high and low resolution model runs. Does this point to an important role of differences in ocean dynamics?

Yes, very likely. Unfortunately, we didn't have the model output available to properly analyse this aspect. Furthermore, a comprehensive analysis of the different ocean dynamics might likely be beyond the scope of this paper. An indication for the cooling North Atlantic in the lowRes CCSM3 experiments stems from the comparison of the AMOC during the LIG (Bakker et al., 2013) compared to PI (Yeager et al., 2006), which we will acknowledge with a statement (Page 7, line 13): We are not aware of comparable AMOC diagnostics for the highRes CCSM3 model.

Figure 3: Why is there no EEM-PI-diff row in this figure?

We prefer to show the EEM-PI-diff of SST and sea ice in Fig. 5 (for DJF) together with the resulting heat flux anomalies and hence we have omitted a EEM-PI-diff row in Fig. 3.

Figure 3: Why are the patterns in SST so different from the SAT (Figure 4) patterns for, for instance, the Arctic region?

In all CCSM3 simulations the Arctic ocean is covered by sea ice throughout the year and hence the SSTs are constantly set to the freezing point temperature of -1.8°C which is the standard for ocean cells fully covered by sea ice. However, EEM-PI changes in the amount of snow falling on sea ice and the resulting changes in insulation of the cold winter atmosphere from the ocean below, explains the SAT pattern over the Arctic ocean in Fig. 4 (most distinctively in autumn).

Figure 8d-e: There appears to be a dipole kind of structure over Greenland for HTdyncore and HTpar. Why is that and how are they related to the large scale wind changes?

The change in surface winds in the NordS-shift experiment indicates anomalous flow above Greenland in the southwest to northeast direction. This likely relates to the observation that the advective transport (Fig. 8d) fosters warming in northeastern Greenland at the expense of a cooling southwestern Greenland building this dipole pattern. This dipole is compensated

by the heat transport associated with HTpar, which due to the fact that it represents parameterized (subgrid) processes is much harder to link with other changes in atmospheric circulation.

Figure 12: Indicate on a map (perhaps in figure 1) where the NEEM or pNEEM site is located.

Will be added to Fig.1

Figure 12: Indicate significance of simulated temperature changes.

Will be added.

Table 1: Why are the other sensitivity tests not included?

As mentioned on page 4, line 15 we only list the six (out of 12) CCSM4 simulations which build the core of the study. We prefer doing so, as the other 6 simulations use the same setup as EEMLabs and EEMNordS except for SST/sea ice, so only little additional info would be displayed by adding those 6 simulations to Table 1.

Table 3: Perhaps a printing issue on my side, but the bold letters are very difficult to distinguish.

We have checked this issue but it indeed seems to be a printing issue on your side.

Table 3 and 4: Using different regions for Greenland (whole island, central Greenland or pNEEM) is a little confusing and perhaps not necessary.

Table 4 has the purpose of displaying the results for the key region of the ice core community and hence can be regarded as an additional service. Table 3 focusing on Greenland as a whole is complementing Figure 11 and corresponds to the overall analysis with a general focus on Greenland as a whole. We, thus, prefer to keep both tables.

**Additional references used in response (and not yet included in manuscript)**

Bakker, P., Stone, E. J., Charbit, S., Gröger, M., Krebs-Kanzow, U., Ritz, S. P., Varma, V., Khon, V., Lunt, D. J., Mikolajewicz, U., Prange, M., Renssen, H., Schneider, B., and Schulz, M.: Last interglacial temperature evolution – a model inter-comparison, Clim. Past, 9, 605-619, doi:10.5194/cp-9-605-2013, 2013.

Bakker, P. and Renssen, H.: Last interglacial model–data mismatch of thermal maximum temperatures partially explained, Clim. Past, 10, 1633-1644, doi:10.5194/cp-10-1633-2014, 2014

Large, W. G., and Danabasoglu, G. Attribution and Impacts of Upper-Ocean Biases in CCSM3, Journal of Climate,19:11, 2325-2346, 2006

Holland, M., Bitz, C. M., Hunke, E.C., Lipscomb, W. H., and Schramm, J.L. Influence of the Sea Ice Thickness Distribution on Polar Climate in CCSM3, Journal of Climate,19:11, 2398-2414, 2006

---

## Author Comment (AC2) · 17 May 2016

**Response to reviewer#2**

The authors analyze the role of sea ice and SST anomalies in the Labrador and Nordic Seas in controlling surface air temperature anomalies over the North Atlantic region (with a special focus on Greenland) during the Last Interglacial (LIG). Using the atmosphere component of CCSM4, a state-of-the-art climate model, a set of sensitivity experiments was performed to disentangle the influence of the Labrador Sea versus the Nordic Seas. The results were analyzed very carefully and in much detail considering heat and moisture budgets. It is found that sea ice retreat and warming in the Nordic Seas is crucial for the simulation of high Greenland temperatures during the LIG, which are evidenced by proxy records, whereas the role of the Labrador Sea is minor. The paper is well written and clearly structured. Although similar experiments and ideas have been published before by Li et al. (2010) with a focus on the last glacial, the results by Merz et al. are novel and show the importance of Nordic Seas ice cover for the LIG. As such, the study by Merz et al. is certainly of interest for the paleo-modelling community and suitable for Climate of the Past. However, the following points have to be taken into account before publication of the study.

We thank the referee for the careful review and the constructive comments. Please find the answers to all specific comments below.

1) p. 1, line 11: "Diabatic processes play a secondary role". This statement is confusing. The simulated SAT anomalies are ultimately caused by anomalous surface energy fluxes, e.g. sensible heating, which is a diabatic process. I think the authors refer to latent heating and radiative processes. Please be more precise.

We agree with the referee that our statement is confusing. What we meant is that the large-scale spreading of the warming is related to heat advection (of sensible heat) rather than to changes in condensation or radiation processes. We will revise the statement to clarify this issue

2) p. 1, line 23: In both models and data the LIG warming is mostly restricted to the extratropics, whereas the tropics show cooling in many regions. Again, please be more precise.

We agree and we will revise the introduction accordingly.

3) p. 2, line 13: The transient CCSM3 simulation used in this study was not part of the paper by Lunt et al. (2013). The CCSM3_Bremen simulation in Lunt et al. (2013) is a time slice (125 kyr BP) run using the T31-version of CCSM3. It is different to the transient simulation by Varma et al. (2015). Please clarify.

Thank you for this accurate observation. We will clarify this issue, e.g. at the following passages: page 2, line 13 and page 6, lines 31-33.

4) p. 3, line 5: In addition to the papers by Li et al. (2005, 2010), cite the study by Zhang et al. (2014), which strongly supports the findings by Li et al., but in a fully-coupled setup.

We will add a respective reference.

5) p. 4, line 1: In addition to Varma et al. (2015), cite the studies by Bakker et al. (2013) and Govin et al. (2014), where the transient CCSM3 LIG simulation has been published first.

We will add the respective references.

6) p. 4, line 5: The two realizations do not only differ in horizontal resolution. Note that different greenhouse gas concentrations have been used as well as a different solar constant. Moreover, the transient character of the low-resolution run as well as the short integration time of the high-resolution time slice simulation should be taken into account. Please rephrase.

We add the following statement to clarify this issue:

*[Note that the two sets of EEM-PI realizations also use slightly different values for GHG concentrations and solar constant (Bakker et al., 2013 and Otto-Bliesner et al., 2013). Furthermore, the transient character of EEM lowRes is different from the time-slice approach of EEM$_{highRes}$.]*

7) p. 5, line 26: How was the decision made on how far the sea ice margin is shifted - to the north? Is it based on the high-resolution LIG simulation or is it arbitrary? Please explain.

The character (direction and magnitude) of the shift was chosen to resemble the EEM-PI$_{diff}$ sea ice anomaly in the respective region (compare Figs. 4 and 5). We add a respective statement (page 6, line 4) to make this clear:

8) p. 6, line 31: The authors have not used the CCSM3_Bremen simulation from Lunt et al. (see above).

As discussed in response to 3) we clarify this and revise the respective statements.

9) p. 7, line 1: As mentioned above, the difference is not only due to horizontal resolution. Different GHG concentrations have been used. In particular, N2O concentration is much higher in the high-resolution CCSM3 experiment than in the low-resolution run. Moreover, a higher solar constant (1367 W/m2) has been used in the high-resolution experiment.

Thank you for this correct observation. This will be accounted for as shown in response to 6)

10) p. 7, line 2: Vegetation is fixed (modern) in the transient CCSM3 low-resolution run.

True, this will be revised accordingly.

11) p. 7, line 11: As mentioned above, higher N2O and solar constant contribute to the warming in the high-resolution CCSM3. I agree that the ocean is also a likely candidate. In fact, as shown in Bakker et al. (2013) the AMOC in the transient low resolution CCSM3 simulation is relatively weak. Reduced oceanic heat transport would contribute to the relatively cool conditions in the North Atlantic. In addition, it should be noted that the pre-industrial reference run by Merkel et al. (2010) has much higher GHG concentrations than the transient LIG simulation (in particular CH4).

Thank you for this valuable comment. We will revise the following paragraph (p.7 line 9-16) to include the additional potential reasons for the diverse EEM-PI warming in the lowRes and highRes CCSM3 simulations.

12) p. 19, line 30: The study by Zhang et al. (2014) may be cited here, showing that processes are similar in coupled and uncoupled (Li et al., 2010) experiments.

We will add the respective reference and an additional statement to account for this valid observation. Your comment is very useful to further strengthen the credibility of our results.

13) p.36, Table 1: A reference is missing for the chosen GHG values.

The GHG values are chosen to correspond with Varma et al., 2015. We add a respective reference in Table 1.

---

## Author Comment (AC3) · 17 May 2016

**Response to reviewer #3**

Summary:

This paper investigates the potential importance of SST and sea ice for the climate conditions in the North Atlantic and Greenland in the Eemian interglacial. Simulations are conducted with CAM3 and CAM4, comparing the pre-industrial (PI) and Eemian C1 CPD Interactive comment Printer-friendly version Discussion paper climates using prescribed sea-surface conditions from fully coupled simulations (at different resolutions) with CCSM3. The main conclusion is that sea ice in the North Sea region can have a large impact on the Greenland climate and a reduction of its prevalence generates a substantial warming over the ice sheet. The sea ice in the Labrador Sea is important for the local climate conditions but has a little to no impact on the Greenland climate. The authors conclude that the climate impact is mostly mediated by near surface turbulent fluxes that influence the atmospheric circulation and thereby cause a warming over the ice sheet. The paper is generally well written and is suitable for Climate of the Past, though first after a substantial revision.

We thank the referee for the thorough review and the stimulating comments, which will help to further improve the manuscript. Please find the answers to all specific comments below. We have not responded yet to the fully technical corrections (e.g. wording, details in figures etc.), but will do so when preparing a revised manuscript.

Major comments:

1. Model validation and motivation

(i) In all modeling studies it is mandatory to prove that the model is capable of producing a reasonable climate that conforms to observations or proxy data records (climate reconstructions) when studying past climates. This is a first sanity check that tells the reader that it might be worthwhile spending the time end energy reading the paper. This manuscript only contains difference fields and the reader is never shown the actual climatological states. I suggest adding a figure showing a comparison of the pre-industrial (PI) simulation with either a reanalysis product or a reliable climate reconstruction (show full fields and how they differ from observations). For the Eemian you can compare with proxy data where such are available. Though this type of comparison is mandatory, in this study it is extra important since the model seems to be sensitive to the horizontal resolution.

We fully acknowledge that model validation is an important prerequisite. For the climate models used here (CCSM3 and CCSM4) this exercise has already been tackled in several previous studies. The most prominent examples of CCSM3 model evaluation for present-day conditions are Collins et al., 2006 and Yeager et al., 2006 (for the lowRes version). Similarly, the CCSM4 model is validated in Gent et al., 2011, Neale et al. 2010, 2013 (atmospheric component CAM4) and Evans et al., 2013, the latter looking at the atmospheric-land-only setup of CCSM4 specifically. Furthermore, Vizcaino et al. 2013 compare CCSM4 with observations with a focus on the climate in Greenland.

The set of CCSM3 experiments in this study build on simulations which are used in several studies (e.g. Otto-Bliesner et al. 2013, Lunt et al. 2013, Bakker et al. 2013, Varma et al. 201?). In these studies, the fully coupled simulations are assessed with respect to their ability simulating Eemian climate conditions including comparisons with Eemian proxy records.

The CCSM4 simulations generated for this paper are also similar to previous studies using the same model (PI/Eemian) setup focusing on the climate around Greenland (Merz et al., 2013, 2014a, 2014b)- These studies include comparisons with reanalysis data for several aspects of atmospheric circulation, precipitation and snow accumulation in Greenland.

In summary, we want to avoid too much overlap with existing studies and rather be concise in this topic. We acknowledge the importance of model validation and we will therefore add a summary of the above mentioned studies in section 2.

(ii) I would like to see a better motivation of the study. What is the goal (what do we wish to learn) and why are we interested in this particular problem? The current motivation seems to be that fully coupled models simulate different sea-surface temperature (SST) and sea-ice cover (SIC) in the Eemian. This is perhaps not too surprising given the large model spread in simulations of both present and future climates. It would be better to motivate the study from available proxy data records from ice cores as well as terrestrial and marine records. Given the large model spread, what makes this model better than any other model and can we trust the results presented here (connected to the model validation)? You can also extend the motivation by looking at AMOC in different models and connect that to differences in the sea-surface conditions.

Proxy data is one important source of information on past climates and as a consequence about the sensitivity of the climate system itself. Numerical modeling offers a second, complementary approach, which is what we aim to focus on in this study. Thus, one of our main motivations is that current simulations of the Eemian are not able to simulate a warming of 7-8C over north western Greenland (suggested by ice cores). In two previous studies (Merz et al. 2014a,b) we assessed the role of the ice sheet configuration and associated moisture changes. Another model deficiency is that coupled models tend to generate too much sea ice (already in the present day climate simulations). So, the question which is answered in this study is how much of the Greenland warming may be due to the uncertainty arising from SST-SIC distribution around Greenland.

We will make an effort to make this clearer in the abstract and the introduction.

2. Modeling approach

(i) Initially you show that the low and high resolution models yield different results in terms of SST and sea ice in the North Atlantic. It is further mentioned that the low resolution model has known problems and does not simulate a reasonable PI climate in the North Atlantic sector (is this also true for the Eemian?). Despite this claim, the majority of the experiments and figures (according to Table 2) are based on results from the low resolution model. This seems like a very odd choice to me. If the model is biased and has known problems, why base almost all figures and analysis on data from this model? Are there even worse problems associated with the high resolution model? If not, can we expect different conclusions if the same analysis is performed on the high resolution data?

The majority of the experiments and results are based on the CCSM4 simulations which all use the same nominal 1° horizontal resolution and showed good ability in simulating the climate in the North Atlantic and in Greenland (Evans et al., 2013, Merz et al., 2013, 2014a, 2014b, Vizcaino et al., 2013). We just use the SSTs and sea ice from $EEM_{lowRes}$ as the basis for our sensitivity experiments shown in the Sections 5.1-5.3. However, we have conducted the same sea ice shift experiments using $EEM_{highRes}$ as basis (see Section 5.4) and come up with very similar results (e.g., compare EEM1/EEM2 numbers in Table 3&4). We will revise the caption and contents of tables 3 and 4 to make this point clearer. We further consider to revise the lowRes/highRes terminology for the CCSM4 simulations (which all use the same resolution) to avoid confusion of the reader.

(ii) I am generally skeptical to the approach taken in sections 4.1 and 4.2 and I am afraid that we are not learning very much from this exercise. CCSM3 and CCSM4 are highly dependent models (e.g. Knutti et al., 2013) that are part of the same model family, meaning that the atmospheric components (CAM3 and CAM4) share the majority of the same code base. The

biggest difference between the models is the deep convection scheme, which plays virtually no role in the latitude range of your focus. Consequently, the comparison of the two atmospheric models is largely redundant as you basically compare results from two simulations with almost the same model using identical forcing protocols. I argue that you can omit this whole comparison and just state that you use SST/SIC from CCSM3 in CAM4 and then prove that the simulated climates are reasonable with respect to reliable data. Also, the near surface temperature is not the best field to use to evaluate differences between AMIP simulations. If the model is capable of producing a realistic climate with realistic turbulent fluxes (e.g. near surface gradients), the near surface temperature is by definition largely similar to skin temperature and you basically prescribe the phenomena that you are investigating.

We are fully aware of the fact that CCSM3 and CCSM4 are similar models as they stem from the same model family. The comparison in Fig. 2 of CCSM3 and CCSM4 simulations is to show the agreement between fully-coupled simulations and atmospheric (-land-only) simulations which use the SSTs/sea ice from the fully-coupled simulations. Hence we see the common/similar model physics in the atmospheric components of CCSM3/CCSM4 rather as an advantage for our study.

The comparison of Eemian proxy data with climate simulations (including the low resolution and high resolution CCSM3) has already been done in various studies (Lunt et al., 2013, Otto-Bliesner et al., 2013, Capron et al., 2014. Hence, we don't want to repeat this comparison as we feel that it will lengthen the study without adding too much novel information. We will make an effort to better summarize the results of the aforementioned references (specifically with respect of the CCSM3 simulations) in the revised version of the manuscript.

Lastly, we totally agree that over ocean/sea ice points the surface temperature is largely determined by the surface heat fluxes and thus a respective signal in SSTs/sea ice concentration directly translates in a corresponding surface temperature signal. This is clearly no surprise to the reasons you mentioned. However, we are also interested in temperatures over land (in particular in Greenland) e.g., as displayed in Fig. 2. For the land points the influence of SST/sea ice changes is not as straightforward as for ocean points and hence worth a closer look (what is done in this study). The key message from Fig. 2 is that the warming patterns in panels e) and f) (i.e. EEM-PI$_{diff}$ for CCSM3 fully-coupled vs. CCSM4 atmosphere only) do not only agree over ocean but also to some degree over land points.

(iii) A large part of the analysis is based on differences between difference fields (EEMPIdiff). These results are almost impossible to wrap ones head around and I wonder what we can learn from such a comparison, especially since the low resolution model has known biases. Also, it would help the interpretation of the results if you used the same color scale in all figures showing the same/similar quantities.

Since the analysis is based entirely on the nominal 1° CCSM4, where some simulations use boundary conditions of an earlier coarser resolution CCSM3 simulation, our results are minimally affected from model or resolution biases. All new simulations that we carried out for this study are classical sensitivity simulations where a single aspect of the model setup is changed at the time to isolate different influences and processes. So, the difference EEM-PI denotes the sensitivity of a single model to the two respective sets of boundary conditions, lowRes or highRes. This is the familiar concept of a climate sensitivity, albeit in this case not related to $CO_2$. The quantity EEM-PI$_{diff}$ is the difference between these two sensitivities. We will revise the text of the manuscript to clarify this term.

For a given variable the color scale is consistent except for Fig. 7 where the SAT range (which is from -5 to 5 C in the other Figures) is from -3 to 12 C. We feel that adapting the

range of Fig. 7 to -5 to 5 C would be unfavorable to display the effect of the warming induced by the shift experiments. We consider adding a respective note in the caption of Fig. 7 to make it clear to the reader that the color scale of Fig. 7 differs from the one in other figures.

(iv) My main concern has to do with the sea-ice retreat experiments. First of all, the amount by which you shift the sea ice seems to be arbitrarily chosen and should be motivated.

We will motivate the design of our sensitivity experiments better in the revised manuscript. The character (direction and magnitude) of the shift was chosen to resemble the EEM-PI$_{diff}$ sea ice anomaly in the respective region (compare Figs. 4 and 5).

We also tried different magnitudes of the shift (not shown in the manuscript) which give results in agreement with what is shown in the current manuscript. We consider mentioning these additional experiments in the revised manuscript.

Second, I am not convinced that these perturbation experiments are designed in a way that they will teach us anything useful about the last interglacial climate. In steady state (no drift due to external forcing) the circulation in atmosphere and ocean is by definition what determines the sea-surface conditions; the SST/SIC is essentially determined by the internal heat flux (Qflux) in the ocean mixed layer and the balance between radiative and turbulent surface fluxes in the atmosphere (SST ~ SWnet –LWnet – LHflux – SHflux – Qflux). When you prescribe the sea-surface conditions and introduce local changes in the SST/SIC, you also introduce a local climate forcing that could never happen in the real world as it is not supported by the rest of the climate system (the open water that is introduced is not consistent with the general circulation).

We agree that any change to the coupled system will result in an imbalance and thus potentially invalidate the new solution. However, in the sea ice shift experiments, this only applies to the ocean circulation. The atmospheric circulation will adjust to the prescribed SST/SIC. Several analyses in the manuscript ensure that (a) our atmosphere-only simulations are not in contradiction with the physically consistent coupled simulations that the boundary conditions were taken from, and (b) that the unavoidable inconsistencies in the ocean implied by the shift in sea ice do not invalidate the basic finding and the benefit of these idealized simulations. We will revise the text to make these points clearer.

If we assume that the sea-ice cover in the Labrador Sea collapsed (for whatever reason), the climate system would do everything it can to rebuild the sea ice over the next few seasons (as is evident from the almost 100W/m2 imbalance in sensible and latent heat fluxes that are reported in the analysis). If we instead assume that we could collapse the Labrador Sea ice and keep the region ice free, the rest of the ocean circulation (and atmospheric circulation for that matter) would have to be different to sustain the reduced sea ice; i.e. there would be changes in the SST field elsewhere and the turbulent fluxes would almost certainly be lower as sea-ice otherwise would form. I know that the chosen modeling approach is not new and that other people have done similar experiments before you (e.g. Deser et al., 2010), but I am concerned that this modeling approach does more damage than good in this particular study. I don't have a patented solution to the problem but I argue that it would be better to run a slab ocean model and alter the internal heat flux convergence in the mixed layer (in a conservative way so it doesn't introduce a global climate forcing) so that the sea ice retreats from the desired regions. This is arguably a better solution as the surface temperature and sea-ice margin are determined by the surface energy balance, which means that it is theoretically possible to construct a climate where there is no sea ice in the desired regions but you have sea-surface conditions that are in balance with the circulation and external forcing. Whether or not this climate state is realistic is of course another question.

We agree that the sea ice shift experiments partly break the physical consistency of the coupled system. This is a general and irremediable aspect of atmosphere-only simulations. However, it also represents an opportunity to investigate the impact of changes when applied within physically reasonable limits. In this context, we argue that the northwestward shifted sea ice edge does not surpass these limits, because it is not generally inconsistent with possible states of the ocean circulation as shown by the fully-coupled CCSM3 simulations. The LabS-shift and NordS-shift experiments are designed to resemble this coupled simulation regionally in order to disentangle the effects of one ocean basin versus the other. A slab ocean model is not suited here because it only includes meridional heat transport in the ocean. In the case of the Labrador Sea, the zonal ocean heat transport is very important.

To illustrate that distinct surface heat fluxes are not only an artifact of atmospheric-only simulations but also possible in CCSM4 fully-coupled simulations we show here sensible and latent heat flux anomalies for a LGM compared to a PI simulation (Fig. R1) (diagnostics provided by the NCAR: http://www.cesm.ucar.edu/experiments/cesm1.0/diagnostics/) Note that such diagnostics are not available for an Eemian simulation. Similar to our atmospheric-only CCSM4 simulations, the heat flux anomalies from the fully-coupled model shown below are the result of distinct changes in SSTs and sea ice (see also Fig. R2) originally caused by changes in external forcing (here LGM vs. PI, in our manuscript EEM vs. PI). More precisely in the LGM the sea ice strongly increased in the Labrador Sea and the Norwegian Sea (Fig. R2) leading to distinct negative heat fluxes in these regions. At the same time adjacent areas in the North Atlantic show distinct positive heat flux anomalies building the dipole structures alike the ones found in our atmospheric simulations.

[Figure]

*Fig. R1: Winter mean (DJF) LGM minus PI change in (left) sensible heat fluxes (W/m$^2$) and (right) latent heat fluxes (W/m$^2$) based on CCSM4 fully-coupled simulations.*

[Figure]

*Fig. R2: Winter mean (DJF) LGM minus PI change in (left) sea ice concentration (%) based on the same CCSM4 fully-coupled simulations as used for Fig. R1.*

3. Interpretation of results

(i) Following the previous comment, it is not at all surprising that you get very strong turbulent fluxes in the sea-ice sensitivity experiments. The prescribed SST/SIC implies that the climatological atmospheric circulation is more or less determined by the prevailing seasurface conditions. When making local changes to the SIC and prescribe SSTs that are not consistent with the circulation, you introduce regions where the climate "wants" to have sea ice, as cold air is advected over open water, but the prescribed sea-surface conditions prevents it from forming. This gives rise to artificial vertical gradients and turbulent fluxes that would never happen in nature as the SST/SIC would respond and go back to an ice covered state. This in turn induces and anomalous atmospheric circulation that has no real world analogue, at least not in a climatological state which is what is investigated here.

As detailed in our reply to the above comments, we understand this concern and we agree that physical inconsistencies due to the *uncoupling* of the formerly consistent ocean-atmosphere system requires great care and a critical discussion of what can be concluded from such idealized numerical experiments. We regret that the current manuscript falls short in this regard and we will revise the text to avoid future misunderstandings. However, we also feel that the comment above is too general in its criticism. Regions where very cold continental air meets open ocean surfaces do exist in the real world. They are not always a model artifact. Specifically, in the Labrador Sea, it is important to note that the coupled version of CCSM4 does simulate open waters here (e.g., Jahn et al., 2012) that produce a strong air-sea heat flux. CCSM4 uses the same atmosphere model as our atmosphere-only simulations, CAM4, which illustrates that a situation similar to our idealized LabS-shift experiment is not physically impossible in this model.

(ii) In my mind, one of the most interesting results in the whole paper is the changes in the lower tropospheric wind field (Fig. 9) that results from manipulating the local SST/SIC in the North Sea. However, no explanation is provided as to why the wind field changes the way it does. I want to see a dynamic argument made for the somewhat counterintuitive response where the lower tropospheric winds impinge on Greenland from seemingly the wrong direction; SE instead of NE where the forcing is located.

We agree with the referee that this is a interesting result worth a closer look. We will report on the results of an extended analysis of this issue when submitting the revised manuscript.

Line-by-line comments:

Page 1, line 1: I would be careful suggesting that the Eemian is a possible analog to the climate in the near future. The Eemian was warm primarily as a result of increased insolation whereas future climates are warm because of higher greenhouse gas concentrations. The former only plays a direct role during parts of the year (in high latitudes) whereas the latter influence the longwave radiation in all seasons.

We are fully aware of the diverse causes and impacts between the Eemian and the current/future warming. Still, the Eemian period remains a valuable test bed period for studying the dynamics of the high-latitude climate system for atmospheric/oceanic conditions warmer than present. We will revise the statement to make it clearer.

Page 1, line 19: This time interval contains both warm and cold phases.

We changed the definition of the last interglacial (Eemian) to 129-116 ka which corresponds to IPCC AR5. The last interglacial is clearly known as warm period although the time interval may include also parts of the transition phases with the preceding/following glacial period. Defining the exact interval of any glacial/interglacial period is a challenge on its own.

Page 4, lines 19-25: This is more of a curious comment than anything else but when you regrid the T31 SST/SIC to the T85 grid, you implicitly introduce an outline of the T31 grid but at the higher resolution. Do you have a feeling for if this will influence the results?

We are a bit uncertain what the referee exactly means here. We extrapolate the SSTs from both types of CCM3 simulations, i.e. 3deg grid (T31x3 simulation) and 1deg (T85x1deg), across all land points and regrid it to the 0.9x1.25 resolution, so there is no "outline" of the original 3deg/1deg land mask.

Page 5, line 24: How does the absence of inter-annual variability in the SST/SIC degrade the representation of the storm track? Add a sentence explaining that.

The study Raible and Blender (2004) shows that the Pacific storm track is shifted north in the absence of inter-annual SST/SIC variability (mainly due to the missing ENSO variability. In the Atlantic there are also changes, in particular more storms move zonally and less to the Northeast. Please note that using a mixed ocean model instead lead to similar behavior of the storm track as for simulations with no inter-annual SST/SIC variability.

Page 6, line 7: The -1.8_C temperature is only used for the SSTs underneath sea ice. The actual temperature of the sea ice is determined by the local surface energy balance, which is generally much lower. It is therefore a bit misleading to use the SST as a measure of the surface temperature and I suggest showing the actual surface temperature instead.

Your comment is completely valid for areas with partial sea ice coverage in terms of that the atmosphere is feeling the surface temperature of the ice according to the local surface energy balance (calculated by the thermodynamic module of the sea ice model CICE). Nevertheless, we prefer to show SSTs in Fig. 1,3,5,6 as it can be shown for both partially ice-free and fully ice-free regions rather than showing the ice temperature for the small areas with partial sea ice coverage what would complicate the illustrations. The SSTs further provide information about how much energy from the surface ocean is available for the atmosphere.

Page 7, end of section 4.1: Determine whether the difference in temperature signal is due to the PI, Eemian or both climate states when going to the lower resolution.

The temperature signal assigned to EEM-PI$_{diff}$ is by definition a combination of both climate states. The positive EEM-PI$_{diff}$ temperature signal tells us that the difference between the absolute temperatures in EEM$_{lowRes}$ and EEM$_{highRes}$ is larger than the difference between the absolute temperatures in PI$_{lowRes}$ and PI$_{highRes}$.

Page 8, line 1: What is the relationship between the SST and the sub-polar gyre?

The circulation of the subpolar gyre influences SSTs in several ways. Firstly, a stronger gyre results in a stronger Irminger Current that transports heat and salt south of Iceland in a westward direction. While this causes a weak direct warming, the salt transport is more important. The enhanced influx of saline waters into the relatively fresh Labrador Sea strengthens deep convection in this region. Since subsurface waters are warmer than the strongly cooled surface waters in this region, this second effect also results in a warming. Lastly, in broader terms, the gyre heat transport dominates over the overturning heat transport in the subpolar latitudes of the North Atlantic. Thus, a stronger subpolar gyre transports more heat northward across the entire width of the ocean basin.

We will revise the manuscript to clarify this statement without distracting from the main focus by adding too much information.

Page 8, line 11: Show the PI SST, it is important for the story!

We are considering including a Figure with the PI SSTs although we think that it is not of too crucial importance as we are mostly discussing EEM-PI changes rather than absolute EEM or PI values. The absolute PI SSTs currently are only mentioned in one single sentence.

Page 8, line 18: particularly strong on SAT above oceanic grid cells... Don't you use identical SST/SIC in CAM3 and CAM4? If so, you expect to see very similar SAT as it represents the temperature just above the ocean surface.

Yes, the SSTs in the CCSM3 simulations (fully-coupled) and the CCSM4 simulations (atmosphere-land-only) are identical and hence we expect very high similarity. Any difference is related to differences in the CAM3 and CAM4 physics/numerics and due to the fact the CCSM3 highRes/lowRes simulations slightly vary with respect to external forcing. This has been pointed out by Reviewer 2 and will be clarified in the revised manuscript.

In contrast, the CCSM4 highRes/lowRes simulations use identical setups (e.g., external forcing) apart from the SST and sea ice fields prescribed as lower boundary conditions and hence are more reasonable test beds to retrieve the role of SSTs and sea ice for the EEM-PI SAT warming pattern.

Page 8, line 19: How much is the winter insolation decreased in winter?

The decrease in insolation depends on the latitude. Please refer to Fig. 1 in Lunt et al., 2013 for specific values. We will add a respective statement to the manuscript.

Page 9: What can we possibly learn from ($\Delta 1 - \Delta 2$) when at least one of the $\Delta$#s have known biases?

Even though both models (1 and 2) might have biases in terms of absolute values, we can investigate the EEM-PI changes in both models and study the influence between different fields (SSTs, sea ice, temperature, precipitation etc.) as these changes base on the physical principles employed in the climate model. Assuming that the physics in the model are correct, we can learn about the importance of single processes and their interactions with other components in the complex climate system.

Page 9, line 11: Which terms does Qnet contain? Radiative fluxes? Turbulent fluxes? Internal heat sources in the ocean? A combination of all or a subset of the above?

Qnet refers to the atmospheric energy balance and is defined as follows:
Qnet =LH+SH+LWnet.

We omit SWnet in the definition due to reason stated in the manuscript (page 9, lines 17-20):

*[Note that we omit the shortwave component in the calculation of Qnet (shown in Fig. 5b) because increased downward shortwave radiation resulting from modifications in surface albedo (e.g., by changing an ocean grid cell from ice-covered to ice-free) does not warm the atmosphere directly but warms the ocean, an effect that is suppressed in our experimental setup where SSTs are prescribed.]*

We will further add the definition to the manuscript in the revised version.

Page 9, lines 21-29: You have prescribed SST, which means that you easily get artificial turbulent surface fluxes as the ocean temperature acts as an infinite source and sink of energy (sign depends on atmospheric conditions).

In all atmospheric simulations with prescribed SSTs, the SSTs are static and hence the atmosphere finds its own equilibrium given the regional heat input by the ocean surface. In agreement with your comment and as described in the manuscript (page 9, lines 21-29), the surface heat flux response to an initial SST change are therefore stronger than in a fully-coupled simulation run in its equilibrium. However, keep in mind that the purpose of the atmospheric CCSM4 simulation is to mimic the sea ice, SST changes (and consequently also the resulting surface energy flux changes) found in the fully-coupled CCSM3 simulations (Fig. 5). As the SST and sea ice anomalies stem from fully-coupled CCSM3 simulations run in their equilibriums, these anomalies are based on physical mechanisms.

We further feel that the physical inconsistency in the atmospheric-only simulations is a small price to pay for the flexibility to investigate a specific detail of the coupled system. Note also that our CCSM4 atmosphere-only simulations are run for only 30 years so that an energy imbalance from a small region does not integrate over very long distances as the ocean's response times (to adjust all ocean circulations) is much slower.

Please also refer to our responses to your major comments.

Page 10: Why do you use the low resolution model when it has known biases?

We use the atmosphere-land-only CCSM4 model which has a nominal 1° resolution for all sea ice experiments and prescribe SSTs/sea ice from both the low resolution and the high resolution model as input (see Chapter 5.4 for an overview of all simulations). However, as we are mostly interested in changes between two simulations, the absolute nature of the SSTs/sea ice input fields is not of importance. Note that the sea ice shift simulations have been repeated starting from the unperturbed conditions of the high resolution coupled CCSM3 simulation (described in Section 5.4). The results and conclusions from these additional simulations are virtually identical with the shift-experiments starting from the CCSM3 low resolution SSTs/sea ice.

Page 10: Fixed SST is almost certainly the source of the strong turbulent fluxes that are highly artificial as they would never happen in nature in the way described in the manuscript, at least not over a long period of time.

The idealized SST and sea ice fields are artificial but they do resemble the regional conditions in the EEM-PI CCSM3 highres coupled simulation and therefore are not fundamentally at odds with a physically consistent system. Note that distinct surface heat fluxes are also found in observations and fully-coupled simulations (Bates et al., 2012)

Please also note our reply to similar comments above.

Page 11, lines 1-3: Why does the warming spread over Greenland? Comment on changes in atmospheric circulation.

The role of changes in the atmospheric circulation is discussed in Chapter 5.1. It is shown that in the NordS-shift experiment the Greenland anti-cyclone is weakened allowing warm air from the Nordic Seas to be advected towards Greenland's interior. In contrast, for PI conditions as well as in the LabS-shift experiment the Greenland anticyclone is stronger and fosters the cold isolated climate in Greenland.

Page 11, line 8-10: Eq. 1 is written in advective form, not flux form. The terms you refer to are therefore showing temperature advection and not heat flux convergence.

We agree with the reviewer and will change to 'horizontal and vertical temperature advection.'

Page 11, lines 16-19: Are you talking about month to month variability or the climatology? The terms have to be identically equal to zero in the latter if the model is in balance.

See next answer

Page 11, line 20: The temperature tendency has to be identically zero for the model to be in balance. You are looking at a climatology after all, or...?

Yes we are looking at climatology. As the model is never to 100% in balance, the temperature tendency is only almost zero.

Page 12, line 6: How much is actually resolved at T31?

Note that all CCSM4 simulations (for which the heat budget calculation is applied) have 0.9x1.25 (not T31) resolution which corresponds to a grid space in Greenland of ca. 50km. Only the two lowRes CCSM3 simulations have T31 resolution.

Page 12, line 13: How does that hang together with the enormous increase in LH flux? I would expect to see a great moistening of the atmosphere when the LH flux increases that much, which in turn increases the cloudiness.

The moisture released by the positive latent heat flux anomaly is constantly transported away by enhanced moisture advection (see Fig. 10). Hence, the increase in atmospheric humidity above the moisture source region is limited as is the increase in cloudiness.

Page 12, line 33-Page 13, line 9: This paragraph is very confusing because you first talk about what you expect to see and then you show that the expected circulation is in fact not true.

We agree that this passage in confusing and it will be revised.

Page 12: What happens to mid- and upper tropospheric winds in these experiments?

The winds in the mid- and upper troposphere show no significant changes.

Page 13, line 20: I don't see a southeastward transport in the figure.

Will be changed to "eastward".

Page 13, lines 20-23: Is this also true in these experiments? Have you done the proper analysis or is it just a conjecture?

We haven't performed a cyclone analysis, which is beyond the scope here, but it is well-known from the literature.

Page 14, lines 15-19: This is the heart of my concern. Everything in the climate system acts to build sea ice where it has been removed but the prescribed SST/SIC don't allow the sea ice to regrow. Since the summer temperature is higher, there will not be any regrowth in the summer season and you don't see equally outrageous turbulent fluxes.

Please see our responses to your main comments on our thoughts why we feel that the sensitivity experiments are still valid.

Page 14, lines 27-34: This is not very surprising either. There is a prevailing southwesterly flow over the northeastern Atlantic, meaning that warm and moist air is advected over the region where you remove the sea ice. There is thus a smaller "urge" for the climate system to regrow sea ice there and you don't see equally large turbulent fluxes.

Figures 4 and 5 show that in both regions (LabS and NordS) removing sea ice leads to distinct winter heat flux anomalies as in both regions cold air is exposed to a relatively warm sea surface. We agree with the reviewer that the winter temperatures in the Labrador Sea are even colder than over large parts of the North Atlantic due to the local advection of cold air from the American continent in contrast to warmer air masses advecting eastward across the Atlantic. Nevertheless, the different magnitudes (in LabS vs. NordS) in winter heat flux anomalies shown in Fig. 11 mostly relate to the chosen boxes across we calculate the averages plotted in Fig. 11. As stated in the manuscript (page 14, lines 29pp) negative heat flux anomalies stemming from the dipole effect are included in the NordS box but not in the LabS box.

Page 18, line 9-10: Have you adjusted the Greenland elevation in these simulations?

Greenland is set to present-day conditions in all experiments presented here. Please refer to Merz et al. 2014a,b for results of simulations with a modified Eemian Greenland ice sheet.

Page 18, line 15: A 3.1_C temperature difference could in principle be due to a lowering of the ice sheet. Since the sea level was quite a bit higher in the Eemian, this is not a bad first guess that could be explored in a greater detail in the manuscript.

Please refer to the extensive analysis of Merz et al. 2014a referenced here.

Page 18, lines 29-34: This section is a bit speculative. Maybe you can extend the discussion to include the importance of precipitation seasonality and the temperature inversion relationship recently discussed by Pausata and Löfverström (2015).

We will consider including the paper in the discussion. Please note that the statements in our manuscript refer to the study by Sime et al. (2013).

Page 19, lines 20-23: You haven't really shown or discussed any proper atmospheric dynamics in this paper. The main focus is on the turbulent fluxes that no doubt will influence the atmospheric circulation. This has not been shown properly though so this statement is merely a conjecture.

We don't feel that is a valid comment as large parts of Section 5.1. are dedicated to changes in atmospheric dynamics.

Figure 2: Validate the model by showing full fields as well as a climate reconstruction.

As stated in our response to your main comment #1 we feel that a lengthy analysis of the full fields and a comparison with climate reconstructions is beyond the scope of this study and has already been done in earlier studies. We will make an effort to better discuss the results of these studies in our manuscript.

Figure 3: The large sensitivity of SIC to the model resolution is curious. Is there any proxy data you can compare this with?

To our knowledge, there is no sea ice proxy available for that period which could be used to judge about either Eemian sea ice mask produced by the two model versions.

Figure 3: What is the purpose of this figure when Fig. 4 shows almost exactly the same thing, though extended to show the response over land as well?

Figure 3 shows sea surface temperature (SST) and sea ice concentration (SIC) whereas Fig. 4 is showing surface air temperature (SAT), so they are not showing the same fields. It is worth showing both the SSTs/SICs (i.e. here used as a forcing as they are prescribed) and the SATs (i.e. the temperature response in the low-level atmosphere). The comparison shows how strongly the atmospheric temperature response is related to changes in SSTs and sea ice (not only over ocean points but also over land)

Figure 10: I am curious as to why there are such large differences in e.g. the Norwegian Sea and southwestern Greenland?

We are not sure what differences the referee refers to but if he/she thinks of the differences between Fig.10c,d it is likely that the our calculation of the moisture fluxes (through finite differences) is not able to fully close the moisture budget diagnosed by P-E.

**Additional references used in response (and not yet included in manuscript)**

Bates, S. C., Fox-Kemper, B., Jayne, S. R., Large, W. G., Stevenson, S., and Yeager, S. G., Mean Biases, Variability, and Trends in Air–Sea Fluxes and Sea Surface Temperature in the CCSM4, Journal of Climate, 25:22, 7781-7801, 2012

Jahn, A., and Coauthors. Late-twentieth-century simulation of Arctic sea ice and ocean properties in the CCSM4. Journal of Climate, 25, 1431–1452, 2012

Vizcaino, M., Lipscomb, W. H., Sacks, W. J., van Angelen, J. H., Wouters, B., and van den Broeke, M. R.: Greenland Surface Mass Balance as Simulated by the Community Earth System Model. Part I: Model Evaluation and 1850-2005 Results, Journal of Climate, 26, 7793–7812, 2013

---

## Author Response (AR1)

We kindly thank the three reviewers for their detailed and stimulating reports. The revised manuscript includes changes to various parts of the manuscript as highlighted in the marked-up manuscript version attached at the end of this document.

Please note that we have renamed the CCSM 4 highRes/lowRes simulations (e.g., $PI_{highRes}$ in $PI_1$ and $PI_{lowRes}$ in $PI_2$) to avoid possible confusion with fully-coupled CCSM3 simulations. The former suffixes for CCSM4 were describing the source of the lower boundary conditions rather than resolution differences in the CCSM4 model simulations themselves. Hence now the naming is clearly different for the CCSM3 and CCSM4 simulations.

In this author response, our answers are colored red and citations from the revised manuscript are in *blue color.*

**Response to reviewer #1**

General:

Merz et al. present an interesting study that for the first time quantifies the possibly important role of North Atlantic sea-ice changes, and there with the sea-ice sensitivity, in the last interglacial. This sea-ice sensitivity could to a large extend explain the model data mismatch in terms of last interglacial Greenland temperatures, as well as explain large inter-model differences in simulated last interglacial climate changes at the high latitudes of the Northern Hemisphere. The methodology and analysis are well thought through and the manuscript well written. I suggest publishing this manuscript in climate of the past after minor revisions.

We thank the referee for the careful review and the constructive comments. Please find the answers to all specific comments below.

Main comment 1:

The manuscript shows that differences in simulated North Atlantic SST and sea-ice cover patterns are important to explain reconstructed Greenland temperature anomalies as well as inter-model differences in terms of simulated last interglacial temperatures. It does not attempt to explain the origin of these SST and sea-ice cover differences, which would likely be a whole study on its own. However, in my view this topic cannot be fully ignored and should at least be introduced and its potential implications discussed. Questions that arise are for instance:

What are the causes of the large SST and sea ice differences between the two versions of CCSM3? Yeager et al. show that under pre-industrial boundary conditions there are important differences in the simulated northward oceanic heat transport between the low and high resolution versions of CCSM3. These findings could be shortly summarized here. Can it be deduced which model version is closer to observations in terms of the simulated pre-industrial North Atlantic ocean circulation?

Both the low and the high resolution versions of CCSM3 have known deficiencies in their representation of Arctic sea ice and heat transport in the Atlantic Ocean (Collins et al, 2006 and Yeager et al., 2006). In particular, the low resolution CCSM3 has a too extensive sea ice cover and an underestimated ocean heat transport. The sea ice cover is smaller and thinner in the high resolution version, which is closer to observations. On the other hand, the high resolution CCSM3 still has a pronounced cold anomaly in the subpolar North Atlantic compared to observations (Collins et al., 2006).

Large and Danabasoglu (2006) devote a whole study to the attribution and impacts of upper-ocean biases in the high (and medium) resolution CCSM3. The study shows that too strong surface winds are likely one reason. Besides, the biases in upper-ocean temperature and

salinity along ocean basin boundaries relate to problems in the representation of ocean upwelling.

We have added a respective description of these model biases including the reference to Large and Danabasoglu (2006) to a new section (2.3) that deals with model validation. We have also extended section 4.2 where we describe how different mean biases among the two model versions are a likely candidate for the very different EEM-PI responses in SSTs and sea ice.

Are the inter-model differences also visible in figure 4 of Lunt et al.? And is the cold bias described here for the low resolution version also the cause of the comparatively low CCSM3 temperatures (winter and annual mean) in the transient last interglacial results (see Bakker et al. 2013, 2014) for the Northern Hemisphere?

Yes, the "error" of the high and low resolution pre-industrial control simulations compared to NCEP (Fig. 4 in Lunt et al., 2013) shows different SAT patterns and manifests that both cases exhibit a cold bias in the North Atlantic. Partly due to the chosen color scale in Fig.4 in Lunt et al., 2013 it is not apparent which of the cold biases is stronger but the low resolution bias seems more spatially extensive. Nevertheless, Fig. 4 in Lunt et al., 2013 nicely illustrates that the high and low resolution versions of CCSM3 produce quite different SAT patterns (globally) and thus should be regarded as different climate models even though they are based on some common model physics.

Even though both models show a cold bias in the North Atlantic, their different model biases (and hence differences in the oceanic background state) are likely contributing to the diverging EEM-PI responses in terms of SST and sea ice. We have added a respective paragraph in Section 4.2 that addresses this issue and explains that in the high resolution CCSM3 we see a clear warming of the North Atlantic (and consequently a sea ice reduction) as the subpolar gyre gets stronger during the Eemian compared to pre-industrial. In contrast, in the low resolution CCSM3 we are missing this strengthening of the subpolar gyre for the Eemian due to non-linear gyre dynamics and the gyre's dependence on background salinity and hence sea ice processes. In summary, we indeed find indications that the cold bias in the low resolution CCSM3 (and particularly the too excessive sea ice cover) suppresses the mechanism that is responsible for the EEM-PI warming in the high resolution CCSM3.

If so, both could be pointed out in the manuscript. One could think that a bias in the climate can be accounted for by looking at the anomaly of last interglacial temperatures with respect to a pre-industrial simulation. How does the bias impact the last interglacial climate? Is also the sensitivity of the overturning more sensitivity to global warming, thus leading to cooling in the North Atlantic under last interglacial forcings?

Please see the answers above how we think that differences in mean climate biases (background state) might impact the EEM-PI climate response.

With regard to further interpretation of the origin of the biases and implications for the stability of the overturning circulation during the last interglacial or under global warming, we feel that this would probably be too speculative and that a comprehensive discussion exceeds the scope of this study.

Main comment 2:

The experiments successfully show the role of sea ice and SSTs in explaining the differences between two versions of the CCSM3 model, and provide a potential mechanism that can yield additional warming over Greenland. However, it does not give more warming

over Europe, something that is mentioned a couple of times in the manuscript. Please come back to this point at the end of the manuscript. Questions that come to mind are for instance:

What does it imply that the model-data temperature mismatch over Europe is not improved when using a model with a more sensitive sea-ice cover? Is there another mechanism or feedback missing? Maybe even a mechanism that can explain both the warming over Greenland and Europe without the need for a larger sea-ice retreat? Please shortly discuss this in the manuscript.

The focus of this manuscript clearly lies on the climate in Greenland (rather than Europe) and we have revised text in several occasions the text to make this clearer. We indeed find that temperatures over Europe are largely insensitive to changes in the sea ice cover. We believe that a retreating sea ice cover is one important mechanism to explain a local warming during the Eemian but it does not exclude other influences such as the response to vegetation changes. However, from our analysis we cannot determine the warming processes for Europe and in order to avoid speculative statements, we prefer not to include this in our discussion and hope to resolve the issue by being more specifically focused on Greenland and the North Atlantic sector throughout the manuscript.

Main comment 3:

An important difficulty of last interglacial climate research is the relatively small number of well resolved temperatures and, especially, sea-ice reconstructions. Does the Holocene thermal maximum possibly provide an analogue that can inform us about what happened during the last interglacial because of higher data availability and the existence of sea-ice reconstructions?

Reconstructions of sea ice are generally rare for all paleoclimatic epochs including the mid Holocene. The intent of our study was not to propose the most likely sea ice simulation for the last interglacial, but to highlight how the uncertainty in the ice cover of periods in the past propagates into the estimates of Greenland temperatures.

We have revised the introduction to define the scope of our study more clearly:

*In summary, the goals of the study are as follows: (1) quantifying the atmospheric warming in and around Greenland related to uncertainty in the Eemian sea ice cover (the uncertainty results from the spread in Eemian sea ice configurations among fully-coupled models), (2) determine whether a sea ice retreat in a particular region leads to a temperature and/or moisture signal recorded in Greenland ice cores such as NEEM, (3) understanding the key processes that link the climate in Greenland with the sea ice in adjacent areas. However, note that we do not aim to propose the most likely sea ice cover for the Eemian but rather like to show the consequences of one or the other scenario.*

Minor comments:

General 1: It is a rather long manuscript, so perhaps the reader can be helped a little more to keep track of the aims and line of the manuscript by shortly repeating those aims and or by providing sort summaries at different points in the manuscript.

We agree with the referee that we should help the reader not to lose track of the study's aims throughout the rather long manuscript. We thus have added an explicit statement of the scope of the manuscript (see comment above) and also included short repetitions of the study aims at the beginning of Sections 4.1, 4.2 and 5.

page 7, line 24:

*The first part of our analysis assesses the uncertainty of the Eemian warming as suggested by the spread among state-of-the-art climate models.*

*page 8, line 15:*

*In the next step, we aim to link the inter-model uncertainty in EEM-PI temperature response (discussed in Sect. 4.1) with the models' representation of SSTs and sea ice in the North Atlantic sector.*

page 11, line 5:

*Sect. 4.3 has demonstrated that the diverse Eemian warming (EEM-PI$_{diff}$, Fig. 4) links to uncertainty in the EEM-PI change in SSTs and sea ice. From the analysis so far, however, it is not possible to distinguish the impact of the heat source in the Labrador Sea from the ones in the Nordic Seas. To disentangle the effect of these two regions, we, consequently, use idealized sea ice sensitivity experiments, which simulate either a sea ice retreat in the Labrador Sea or in the Nordic Seas …*

General 2: The potentially important role of sea-ice changes in the North Atlantic in the last interglacial climate have been suggested previously, in relation with observations from Greenland ice cores (Sime et al., 2013) and with large inter-model differences in simulated annual mean and winter temperatures (Bakker et al., 2013). It would be good to mention this in the introduction.

Thank you for bringing these papers to our attention. We have included Bakker et al., 2013 in the revised introduction of our manuscript. The results by Sime et al., 2013 are discussed in Section 6.

Line 7 page 1: 'thus', not everyone is familiar with this model-data mismatch, shortly introduce it in the abstract.

The abstract has been revised to make this clear.

*The last interglacial, also known as the Eemian, is characterized by warmer than present conditions at high latitudes. This is implied by various Eemian proxy records as well as by climate model simulations, though the models mostly underestimate the warming with respect to proxies. Simulations of Eemian surface air temperatures (SAT) in the Northern Hemisphere further show large variations between different climate models and it has been hypothesized that this model spread relates to diverse representations of the Eemian sea ice cover.*

Line 12 page 1: 'accumulation', this is not mentioned before in the abstract and thus appears a little disconnected from the previously discussed issues.

We have changed "accumulation" to "snow accumulation" and hope that this is clearer. As the abstract should be as concise as possible there is unfortunately no space for a detailed introduction of any term and process.

Page 2: More work on the last interglacial and simulated temperatures over Greenland has been done previously, consider discussing that work, for instance by Loutre et al., Goelzer et al., Bakker et al. and Sanches-Goni et al. and Govin et al.

We have added the Bakker et al., 2013 to the references mentioned in the introduction. We are aware of the growing body of Eemian climate literature and have studied the other references which mostly are EMIC studies focusing on the interaction of freshwater fluxes with the transient evolution of the last interglacial. We feel that those studies only partly fit the scope of our study, i.e. the influence of regional sea ice/SST patterns on Greenland's surface climate.

Line 19 page 1: As you are probably aware, the term Eemian is used to describe a pollen-based warm period in Europe, the regional continental equivalent of the general last interglacial period. Consider using last interglacial instead of Eemian throughout the manuscript.

We are aware of the original meaning of "Eemian". Still, we feel that „Eemian" is a widely accepted term in the paleoclimate scientific community for the last interglacial period. Using the term "Eemian" is more in line with our previous studies (Merz et al., 2014a,b) which can be regarded as companion papers also focusing on the climate in/around Greenland during this time period.

Lines 3-6 page 2: These lines seem to suggest that proxies can resolve, annual, summer and winter temperature changes for the last interglacial. Please clarify.

The respective paragraph has been revised to avoid this confusion. The seasonality issue rather relates to the models than to the proxies as models can provide information about the temperature seasonality of the Eemian.

*In the Northern Hemisphere (NH), the models indeed show a distinct warming in summer which is a direct result of increased summer insolation. In contrast, the models mostly fail to simulate a warming for winter, but rather generate lower temperatures due to the decrease in winter insolation (Lunt et al., 2013). This leads to a disagreement between models and proxies in annual mean temperatures that either originates from missing feedbacks in the model simulations and/or misconceptions in the interpretation of the proxy records.*

Line 7 page 2: What 'Eemian proxies' is referred to here? From what region? Please provide references.

We have added according references (i.e. Turney and Jones (2010) and Capron et al., 2014).

Line 31 page 2: Consider referring to Capron et al. and Govin et al.

We have revised this sentence and added Capron et al., 2014 which seems to be the most appropriate reference here.

Lines 29-33 page 2: What season is discussed here? Is it possible that the  summers (?) were warmer, but still the winters were not and neither was the accompanying sea-ice cover decreased?

Axford et al., 2011 refers to summer temperatures whereas Bauch et al. 2012 does not refer to a single season. We are not aware of additional temperature reconstructions for the winter season for the last interglacial in this area. In the low resolution CCSM3 we see that Eemian winters were colder and sea ice was rather expanding but again this model seems in contrast with many other climate models which generally show a stronger warming for the last interglacial (e.g., see Bakker et al., 2013, Lunt et al., 2013). Hence, we can hardly do more than speculate on the last interglacial state of the NH sea ice, particularly for winter.

Line 3 page 6: Why is a 2m thick sea-ice cover used? What are the potential implications of this assumption, please discuss.

2-m sea ice thickness is standard for all CCSM4 atmospheric simulations with prescribed sea ice cover and there is no choice on that in the state-of-the art configurations of the

(atmospheric) CCSM simulations. We cannot really comment on this standard but it corresponds to the observed sea ice thickness in the NH although there is quite a range in sea ice thickness (0-5m), e.g., based on recent CryoSat-2 measurements.

We think that the thermodynamic module of the sea ice model has likely been developed to generate reasonable surface heat fluxes for a 2 meter thick sea ice layer. Moreover, in the marginal sea ice areas (where sea ice concentration is between >0% and <100%) most of the atmosphere-ocean heat exchange anyway happens through the gaps and fractures in the ice, i.e. the ocean surface NOT covered by sea ice. Thus, the thickness of the sea ice where it is present is of rather low importance.

Note also that we haven't tested the sensitivity of sea ice thickness on the Arctic climate as this is beyond the scope of this paper.

Line 3 page 9: It would be helpful for the reader if the 125ka external forcings (GHG and orbital) and their impacts are shortly described (perhaps in the method section), in terms of their annual mean and also seasonal impact.

We agree that this information should be included in the manuscript and have added a respective statement in the method section. Note that the impact of the Eemian vs. pre-industrial external forcing in the same atmosphere-land-only CCSM4 simulations has already been discussed in a chapter in Merz et al. (2014a) so we include a respective reference.

*The Eemian external forcing differs from pre-industrial conditions by lower GHG concentrations (Table 1) and anomalous solar insolation due to differences in the orbital parameters. The climate effect simulated by CCSM4 associated with these changes in external forcing is described in Merz et al. (2014a).*

Line 13-14 page 9: Perhaps an order of magnitude difference can be given to illustrate the dominant role of the turbulent fluxes over the radiative fluxes.

We have added respective estimates which are of the order of 10-20 W/m$^2$ (LWnet) and up to 150 W/m$^2$ for SHF and LHF.

Line 4 page 10: Perhaps at this point come back to the large inter-model spread suggested by previous work (Lunt et al., Otto-Bliesner et al., Nikolova et al. and others) to put the findings in a bigger picture as an introduction to the next section.

As stated above we have made an effort to better remind the reader of the goals of the study (here at the beginning of Section 5). Specifically, we have added the following sentences here relating our results to the inter-model spread found in the literature.

*Sect. 4.3 has demonstrated that the diverse Eemian warming links to uncertainty in the EEM-PI change in SSTs and sea ice. Consequently, our results support the hypothesis by Lunt et al. (2013),Otto-Bliesner et al. (2013),Nikolova et al. (2013) that sea ice is crucial in explaining the inter-model spread in simulated Eemian warming.*

Line 21 page 11: So what are the SATs discussed before if not 'lowest terrain-following level?

The SAT refers to the 2-m temperature which is common in most climate models. The 2-m temperature is an interpolated diagnostic and also a terrain-following measure. Therefore, SAT is virtually identical to the temperature at the lowest terrain-following model level (which is ca. at 20m height). We have taken out the sentence (former version: page 11, line 22) to avoid confusion.

Line 13 page 12: Is the feedback by clouds also small over the Nordic Seas?

We do find some moderate increase in cloud cover directly above the main LHFLX anomalies in the Nordic Seas. However, we find that all changes in cloud cover do not lead to significant radiation anomalies and hence are not of crucial importance for the temperature response.

Line 3 page 13: Earlier on, when winter changes are discussed, mention that seasonality will be covered later.

We have added a statement at the end of section 4.2 to advertise the seasonality section:

*Eventually, the seasonality of selected key processes is presented in Section 5.3.*

Line 12 page 15: Are these SATs for Greenland averages over the whole of Greenland (and also in Figure 11E)?

Yes. We these are area-averaged SATs for the whole of Greenland. We have added a respective clarification to the caption of Fig. 11.

Line 21-23 page 16: Consider repeating what EEM-PIdiff stands for to make this point more clear.

Thank you for this suggestion. We have revised this last paragraph of Section 5.4and included a repetition of the definition of EEM-PI$_{diff}$.

Line 15 page 17: Consider giving the ages covered by the NEEM core.

We do not feel that this adds much clarification here as the full NEEM core actually extends beyond the Eemian but its information from the penultimate glacial is disturbed by folding effects etc. Moreover, our simulations are rather generally valid for an Eemian optimum but do not refer to a specific time period or a transient evolution of the Greenland temperature.

Line 17 page 17: Give distance between NEEM and pNEEM to give the reader an idea of the difference.

We have added the following information:

*The Eemian ice in NEEM was originally deposited at pNEEM (Merz et al., 2014a,b), a location ca. 300km upstream of NEEM relatively close to the summit of the ice sheet.*

Lines 29-32 page 17: It is not clear how this connects to the topic of this manuscript, please clarify.

This statement was included to provide some perspective on our results in the context of the current climate change. However, we have revised this part of the manuscript and omitted the comparison with the sea ice changes related to global warming as it seemed to rather confuse the reader.

Line 7 page 18: Give range of temperature estimate. Is this number altitude corrected? This seems relevant with the discussion later on.

We prefer to just mention the upper limit of the temperature estimate as we focused on the maximum temperature response in Merz et al., 2014a, i.e. for the simulated minimum in the Eemian Greenland ice sheet volume/extent. Further, the number (3.1K) is altitude corrected as will be clarified in the revised statement (see next point).

Line 15 page 18: Is this 3.1K because of elevation changes, circulation changes? Please shortly summarize. What about other work on this topic by for instance Stone et al., Langebroeck et al. and Fyke et al.?

Yes the 3.1K estimate is altitude corrected. The full warming effect to explain the 3.1K is due to a series of changes in the low-level winds and eventually the surface energy balance following a change in the Greenland ice sheet topography as discussed in full details in Merz et al., 2014a. We have extended the sentence to make this clearer.

*Depending on the actual ice sheet topography this results in an additional annual mean warming of up to 3.1° C at pNEEM (altitude-corrected) resulting from changes in Greenland's surface energy balance (Merz et al., 2014a).*

However, we prefer to guide the reader to the reference rather than giving a full summary of the topography-effects as this would further lengthen the already rather extensive discussion section. To our knowledge, the studies led by Langebroeck, Fyke and Stone investigate possible changes in the Greenland ice sheet topography during the Eemian but do not estimate/simulate the associated climate/temperature effect.

Line 34 page 18: Be more specific about what 'climate change' means here.

"Eemian climate change" is changed to "Eemian warming"

Line 2 page 19: What about changes in the seasonality of precipitation?

In Merz et al., 2014b we show that Greenland precipitation is more biased towards the summer season in the Eemian compared to PI. However, Sime et al., 2013 states that uncertainty about local interglacial sea surface conditions, rather than precipitation intermittency changes, may lead to the largest uncertainties in interpreting d18O-related temperature estimates from Greenland ice cores. Anyway, the d18O-temperature relationship is complicated by a number of processes that can impact the assumptions taken in NEEM community members (2013) resulting in considerable uncertainty around the NEEM temperature estimate of 8 +/- 4 ° C. Hence, in this study we want to focus on d15N as this is more appropriate proxy to compare with annual mean SATs from model simulations. Nevertheless, we have included a statement that precipitation seasonality can also be an issue for a meaningful interpretation of the d18O record:

*However, a meaningful interpretation of the NEEM d18O record is further complicated by the fact that the Eemian warming in Greenland mainly occurs in summer (due to orbital forcing) but d18O is rather tied to winter temperatures (Sjolte et al., 2014). Further, there are possible interferences with changes in precipitation seasonality or the inversion temperature relationship (Pausata and Löfverström, 2015).*

Line 11 page 19: Is this for specific regions? Please clarify.

We have revised this statement as follows:

*These simulations are in better agreement with Eemian SST and SAT proxy records from the NH extra-tropics.*

Lines 24-26 page 19: Make clear that this combined experiment has in fact not been performed.

We have revised the statement as follows:

*The Eemian annual mean warming of 5°C above present-day derived from the NEEM d15N record is consistent with CCSM4 model simulations for the scenario that a retreat in the Nordic Seas sea ice (shown here) coincided with the warming associated with a substantial reduction of the Greenland ice sheet(shown in Merz et al. (2014)).*

Figure 2: So does this indicate that the atmosphere is of little importance in determining the LIG climate response to the orbital forcing? What about the role of vegetation?

Fig. 2 does imply that the ocean and sea ice components are most likely responsible for the spread among the two different EEM-PI simulations. This does not mean that the atmosphere itself is not reacting to the anomalous (Eemian) orbital forcing, but it does so in a rather consistent way in both CCSM3 model simulations though the different resolution also in the two atmospheric components. However, as always the pure sensitivity of a single component of the climate system (i.e. the atmospheric model component alone) is only to guess from a fully-coupled setup. An experiment with an atmospheric model simulation forced by the anomalous Eemian orbital forcing but pre-industrial sea ice/SSTs might be a possible experiment to answer this question in detail. For this study here, however, this simulation is rather beyond the scope.

Note that the vegetation is held to modern values in all CCSM3 experiments (our initial statement that the CCSM3 EEM$_{lowRes}$ simulation used a dynamic vegetation model was actually wrong as correctly pointed out by Reviewer #2 – we have revised it accordingly). Hence, in the CCSM3 simulations shown here vegetation processes cannot be responsible for the temperature spread seen in EEM-PI$_{diff}$ (Fig. 2).

Figure 3: The patterns are very different for the high and low resolution model runs. Does this point to an important role of differences in ocean dynamics?

Yes, very likely. Unfortunately, we did not have the model output available to properly analyse this aspect and a comprehensive analysis of the different ocean dynamics is beyond the scope of this paper. Nevertheless, as discussed in our response to your major point 1, we have added a discussion of possible ocean processes that likely explain the diverse EEM-PI climate response among the two CCSM3 versions.

Figure 3: Why is there no EEM-PI-diff row in this figure?

We prefer to show the EEM-PI-diff of SST and sea ice in Fig. 5 (for DJF) together with the resulting heat flux anomalies and hence we have omitted the EEM-PI-diff row in Fig. 3.

Figure 3: Why are the patterns in SST so different from the SAT (Figure 4) patterns for, for instance, the Arctic region?

In all CCSM3 simulations the Arctic Ocean is covered by sea ice throughout the year and hence the SSTs are constantly set to the freezing point temperature of -1.8°C which is the standard for ocean cells fully covered by sea ice. However, EEM-PI changes in the amount of snow falling on sea ice and the resulting changes in insulation of the cold winter atmosphere from the ocean below, explains the SAT pattern over the Arctic ocean in Fig. 4 (most distinctively in autumn). However, we feel that this process hardly relates to the rest of our study and hence we have omitted a respective discussion in the manuscript.

Figure 8d-e: There appears to be a dipole kind of structure over Greenland for HTdyncore and HTpar. Why is that and how are they related to the large scale wind changes?

The change in surface winds in the NordS-shift experiment indicates anomalous flow above Greenland in the southwest to northeast direction. This likely relates to the observation that the advective transport (Fig. 8d) fosters warming in northeastern Greenland at the expense of a cooling southwestern Greenland building this dipole pattern. This dipole is compensated by heat transport associated with HTpar, which due to the fact that it represents parameterized (subgrid) processes is much harder to link with other changes in atmospheric circulation.

Technical corrections:

Line 3 page 1: Perhaps 'Northern Hemisphere high latitudes'.

The beginning of the abstract has been revised to state more clearly that we focus on the Northern Hemisphere extra-tropics.

Line 11 page 1: Perhaps 'Nordic Seas sea ice retreat'.

Done

Line 14 page 3: The line 'Thereby the authors. . .' seems a little redundant and could be removed.

Done

Line 3 page 5: Remove 'it' and put comma after 'Sect. 4'.

Done

Line 16 page 5: A new type of idealized.

Done

Line 10 page 6: Consider replacing 'cutting through' with 'in'.

Done

Line 22 page 7: Twice CCSM4, should one of them be CCSM3?

Yes, we have revised it accordingly.

Line 17 page 11: HTdyn-core is not a very descriptive acronym. Consider using something else that makes it clearer that it deals with resolved heat transport.

We have changed HTdyn-core to HTres (representing the resolved heat transport).

Line 19 page 11: Giving the field names is perhaps not necessary.

Has been removed.

Line 19-20 page 11: Consider rewording to "Note also that all simulations are run into equilibrium, so the total temperature tendency (dT/dt) is almost zero.

Has been revised.

Lines 13-12 page 17: words 'which are mostly drilled on top of the Greenland ice sheet', is not very relevant, consider removing.

Done

Line 3 page 18: Refer to table 4.

A respective reference has been added.

Lines 12-14: Difficult to read, please reword.

This sentence has been revised as follows:

*Another possibility are surface climate changes related to modifications in the Greenland ice sheet topography as Greenland must have been smaller during the Eemian to conform with observed sea level high stands (Church et al., 2013).*

Line 22 page 18: Should instead of shall.

Done

Line 32 page 18: Remove 'Thereby'.

Done

Line 34 page 18 to line 2 page 19: Difficult to read, please rephrase.

This sentence has been revised as follows:

*However, a meaningful interpretation of the NEEM d18O record is further complicated by the fact that the Eemian warming in Greenland mainly occurs in summer (due to orbital forcing) but d18O is rather tied to winter temperatures (Sjolte et al., 2014).*

Figure 1: Explain meaning of solid versus stippled green boxes.

Done

Figure 2: Mention ones, here or in main text, how significance level is determined. Using yearly averaged time series?

We have clarified in the caption of Fig. 2 that we use annual mean SAT time series.

Figure 6: Continents are either white or grey in the different panels.

The continents in the different panels of Fig. 6 have different colors on purpose: continents are marked grey in Fig. 6a and 6e because SSTs and sea ice are not valid for land points.

White colors in Fig. 6b-d, 6f-h denote surface heat flux values between -10 W/m$^2$ and +10W/m$^2$ as indicated by the colorbar. Hence the white continents in these panels are not representing invalid values but indicate that the surface heat flux anomalies are very small over land. Consequently, we prefer to keep these map plots as is.

Figure 7: Perhaps a personal preference, but I like better the colour scales that have white around the zero value (for instance figure 5).

Thank you for spotting this. We have adapted the color scales in Figs. 2 and 4 accordingly. In the other cases we prefer to keep to the chosen color scales to clearly distinguish between positive and negative values (e.g., in Fig. 8 for the terms of the energy equation).

Figure 10: Perhaps in panels b and d remove the vectors if they are not significant.

We prefer to keep all vectors as determining the significance for vectors is not straight-forward as it combines the information of 2 components (zonal wind **u**, meridional wind **v**). Hence it could be that **u** changes significantly but **v** does not.

Figure 12: Indicate on a map (perhaps in figure 1) where the NEEM or pNEEM site is located.

Added to Fig.7

Figure 12: Indicate significance of simulated temperature changes.

Done

Table 1: Why are the other sensitivity tests not included?

As mentioned on page 4, line 15 we only list the six (out of 12) CCSM4 simulations which build the core of the study. We prefer doing so, as the other 6 simulations use the same setup as EEM$_{LabS}$ and EEM$_{NordS}$ except for SST/sea ice, so very little additional info would be displayed by adding those 6 simulations to Table 1.

Table 3: Perhaps a printing issue on my side, but the bold letters are very difficult to distinguish.

We have checked this issue but it indeed seems to be a printing issue on your side.

Table 3 and 4: Using different regions for Greenland (whole island, central Greenland or pNEEM) is a little confusing and perhaps not necessary.

Table 4 has the purpose of displaying the results for the key region of the ice core community (i.e. central Greenland) and hence can be regarded as an additional service. Table 3 focuses on Greenland as a whole, complements Figure 11 and corresponds to the overall analysis with a general focus on Greenland as a whole. We, thus, prefer to keep both tables.

Further, we prefer not to compute the simulated values for pNEEM itself (e.g., on a nearest-grid-point basis) as single grid point values can be problematic (e.g., neighbor grid points can differ quite distinctively, e.g. due to parameterized processes). Consequently, we rather compare the NEEM/pNEEM ice core estimates with the simulated central Greenland average (in Table 4).

**Response to reviewer#2**

General:

The authors analyze the role of sea ice and SST anomalies in the Labrador and Nordic Seas in controlling surface air temperature anomalies over the North Atlantic region (with a special focus on Greenland) during the Last Interglacial (LIG). Using the atmosphere component of CCSM4, a state-of-the-art climate model, a set of sensitivity experiments was performed to disentangle the influence of the Labrador Sea versus the Nordic Seas. The results were analyzed very carefully and in much detail considering heat and moisture budgets. It is found that sea ice retreat and warming in the Nordic Seas is crucial for the simulation of high Greenland temperatures during the LIG, which are evidenced by proxy records, whereas the role of the Labrador Sea is minor. The paper is well written and clearly structured. Although similar experiments and ideas have been published before by Li et al. (2010) with a focus on the last glacial, the results by Merz et al. are novel and show the importance of Nordic Seas ice cover for the LIG. As such, the study by Merz et al. is certainly of interest for the paleo-modelling community and suitable for Climate of the Past. However, the following points have to be taken into account before publication of the study.

We thank the referee for the careful review and the constructive comments. Please find detailed answers to all specific comments below.

Specific comments:

1) p. 1, line 11: "Diabatic processes play a secondary role". This statement is confusing.

The simulated SAT anomalies are ultimately caused by anomalous surface energy fluxes, e.g. sensible heating, which is a diabatic process. I think the authors refer to latent heating and radiative processes. Please be more precise.

We agree with the referee that our statement is somewhat confusing. What we meant is that the large-scale spreading of the warming is related to advection of sensible heat rather than to changes in condensation or radiation processes. We revised the statement as follows:

*The large-scale spread of the warming simulated for the sea ice retreat in the Nordic Seas is mostly explained by anomalous heat advection rather than by radiation or condensation processes.*

2) p. 1, line 23: In both models and data the LIG warming is mostly restricted to the extra-tropics, whereas the tropics show cooling in many regions. Again, please be more precise.

We have revised the introduction as follows to account for your valid comment:

*The last interglacial (ca. 129–116 ka), also known as the Eemian, is often regarded as a possible analogue for future climate as it stands for the most recent period in the past characterized by a warmer than present climate. In contrast to the future year-round warming induced by rising greenhouse gas (GHG) concentrations, the Eemian warming, driven by anomalous orbital forcing, was mostly confined to the summer season and the extra-tropics.*

3) p. 2, line 13: The transient CCSM3 simulation used in this study was not part of the paper by Lunt et al. (2013). The CCSM3_Bremen simulation in Lunt et al. (2013) is a time slice (125 kyr BP) run using the T31-version of CCSM3. It is different to the transient simulation by Varma et al. (2015). Please clarify.

Thank you for spotting this. We have adapted the text at the following passages to correct for this issue.

Page 2, line 13: remove Lunt et al., 2013 reference

Page 6, line 31: remove "reproduced here in Fig.2a,c"

Page 6, line 31-33: change to:

*Here we show a analogous comparison of the EEM-PI temperature response of a set of high-resolution (EEM-PI$_{highRes}$, Fig.2a) and a low-resolution (EEM-PI$_{lowRes}$, Fig. 2c) fully-coupled CCSM3 simulations, previously introduced in Section 2.1*

We also have clarified the correct source and references of the low resolution CCSM3 simulation in Section 2.1.

4) p. 3, line 5: In addition to the papers by Li et al. (2005, 2010), cite the study by Zhang et al. (2014), which strongly supports the findings by Li et al., but in a fully-coupled setup.

We added a respective reference.

5) p. 4, line 1: In addition to Varma et al. (2015), cite the studies by Bakker et al.(2013) and Govin et al. (2014), where the transient CCSM3 LIG simulation has been published first.

We added the respective references.

6) p. 4, line 5: The two realizations do not only differ in horizontal resolution. Note that different greenhouse gas concentrations have been used as well as a different solar constant. Moreover, the transient character of the low-resolution run as well as the short integration time of the high-resolution time slice simulation should be taken into account. Please rephrase.

We added the following statement to clarify this issue:

*Note that the two sets of EEM-PI realizations also use slightly different values for GHG concentrations and solar constant (Bakker et al., 2013 and Otto-Bliesner et al., 2013). Furthermore, the transient character of EEM$_{lowRes}$ is different from the time-slice approach of EEM$_{highRes}$.*

7) p. 5, line 26: How was the decision made on how far the sea ice margin is shifted - to the north? Is it based on the high-resolution LIG simulation or is it arbitrary? Please explain.

The character (direction and magnitude) of the shift was chosen to resemble the EEM-PI$_{diff}$ sea ice anomaly in the respective region (compare Figs. 4 and 5). We have made an effort to more clearly state these considerations in the revised manuscript:

*In summary, our sea ice shift experiments are of idealized nature but the SIC and SST boundary conditions locally resemble fields from the fully-coupled CCSM3 simulations. The direction and magnitude of the shift are chosen to locally, i.e., either in the Labrador Sea or the Nordic Seas, mimic the difference between CCSM3$_{lowRes}$ and CCSM3$_{highRes}$ in order to disentangle their combined effect in EEM-PI$_{diff}$.*

8) p. 6, line 31: The authors have not used the CCSM3_Bremen simulation from Lunt et al. (see above).

As discussed in response to 3) we clarify this and revised the respective statements.

9) p. 7, line 1: As mentioned above, the difference is not only due to horizontal resolution. Different GHG concentrations have been used. In particular, N2O concentration is much higher in the high-resolution CCSM3 experiment than in the low-resolution run. Moreover, a higher solar constant (1367 W/m2) has been used in the high-resolution experiment.

Thank you for this correct observation. This is now clarified in the model description of the CCSM3 simulations.

10) p. 7, line 2: Vegetation is fixed (modern) in the transient CCSM3 low-resolution run.

True, has been corrected.

11) p. 7, line 11: As mentioned above, higher N2O and solar constant contribute to the warming in the high-resolution CCSM3. I agree that the ocean is also a likely candidate. In fact, as shown in Bakker et al. (2013) the AMOC in the transient low resolution CCSM3 simulation is relatively weak. Reduced oceanic heat transport would contribute to the relatively cool conditions in the North Atlantic. In addition, it should be noted that the pre-industrial reference run by Merkel et al. (2010) has much higher GHG concentrations than the transient LIG simulation (in particular CH4).

Thank you for this valuable comment. We have added new paragraphs to Section 4.1 and 4.2 to describe the potential reasons for the diverse EEM-PI warming in the lowRes and highRes CCSM3 simulations.

12) p. 19, line 30: The study by Zhang et al. (2014) may be cited here, showing that processes are similar in coupled and uncoupled (Li et al., 2010) experiments.

We have added a respective reference and an additional statement to account for this valid observation. Your comment is very useful to further strengthen the credibility of our results.

*Further evidence for the validity of the used sea ice sensitivity experiment approach stems from the fact that the relationship between Nordic Seas sea ice and Greenland temperatures in a glacial climate is consistent among atmospheric (Li et al., 2010) and fully-coupled simulations (Zhang et al., 2014).*

13) p.36, Table 1: A reference is missing for the chosen GHG values.

The GHG values are chosen to correspond with Varma et al., 2015. We added a respective reference in Table 1.

**Response to reviewer #3**

General:

This paper investigates the potential importance of SST and sea ice for the climate conditions in the North Atlantic and Greenland in the Eemian interglacial. Simulations are conducted with CAM3 and CAM4, comparing the pre-industrial (PI) and Eemian C1 CPD Interactive comment Printer-friendly version Discussion paper climates using prescribed sea-surface conditions from fully coupled simulations (at different resolutions) with CCSM3. The main conclusion is that sea ice in the North Sea region can have a large impact on the Greenland climate and a reduction of its prevalence generates a substantial warming over the ice sheet. The sea ice in the Labrador Sea is important for the local climate conditions but has a little to no impact on the Greenland climate. The authors conclude that the climate impact is mostly mediated by near surface turbulent fluxes that influence the atmospheric circulation and thereby cause a warming over the ice sheet. The paper is generally well written and is suitable for Climate of the Past, though first after a substantial revision.

We thank the referee for the thorough review and the stimulating comments, which helped to further improve the manuscript. Please find the answers to all specific comments below.

Major comments:

1. Model validation and motivation

(i) In all modeling studies it is mandatory to prove that the model is capable of producing a reasonable climate that conforms to observations or proxy data records (climate reconstructions) when studying past climates. This is a first sanity check that tells the reader that it might be worthwhile spending the time end energy reading the paper. This manuscript only contains difference fields and the reader is never shown the actual climatological states. I suggest adding a figure showing a comparison of the pre-industrial (PI) simulation with either a reanalysis product or a reliable climate reconstruction (show full fields and how they differ from observations). For the Eemian you can compare with proxy data where such are available. Though this type of comparison is mandatory, in this study it is extra important since the model seems to be sensitive to the horizontal resolution.

We fully acknowledge that model validation is an important prerequisite. For the climate models used here (CCSM3 and CCSM4) this exercise has already been tackled in many existing studies as both models are extensively used in the climate science community. The most prominent examples of CCSM3 model evaluation for present-day conditions are Collins et al., 2006 and Yeager et al., 2006 (for the lowRes version). Similarly, the CCSM4 model is validated in Gent et al., 2011, Neale et al. 2010, 2013 (atmospheric component CAM4) and Evans et al., 2013, the latter looking at the atmospheric-land-only setup of CCSM4 specifically. Furthermore, Vizcaino et al. 2013 thoroughly validates CCSM4 with a focus on the climate in Greenland.

The set of CCSM3 experiments in this study build on simulations which are already published and described in several studies (e.g. Otto-Bliesner et al. 2013, Lunt et al. 2013, Bakker et al. 2013, Varma et al. 2015. In these studies, the fully coupled simulations are assessed with respect to their ability simulating Eemian climate conditions including comparisons with Eemian proxy records.

The CCSM4 simulations generated for this paper are also similar to previous studies using the same model (PI/Eemian) setup focusing on the climate around Greenland (Merz et al., 2013, 2014a, 2014b). These studies include comparisons with reanalysis data for several aspects of atmospheric circulation, precipitation and snow accumulation in Greenland.

In summary, we want to avoid too much overlap with existing studies and rather be concise in this topic as the focus of this study clearly is on the processes explaining simulated EEM-PI changes irrespective of the absolute PI/Eemian climate. Nevertheless, we acknowledge that the model needs to be able to reasonably simulate the present-day climate to have confidence in the respective results. Therefore, we have added a new subsection (2.3) accounting for "model validation" based on the wealth of existing study.

(ii) I would like to see a better motivation of the study. What is the goal (what do we wish to learn) and why are we interested in this particular problem? The current motivation seems to be that fully coupled models simulate different sea-surface temperature (SST) and sea-ice cover (SIC) in the Eemian. This is perhaps not too surprising given the large model spread in simulations of both present and future climates. It would be better to motivate the study from available proxy data records from ice cores as well as terrestrial and marine records. Given the large model spread, what makes this model better than any other model and can we trust the results presented here (connected to the model validation)? You can also extend the motivation by looking at AMOC in different models and connect that to differences in the sea-surface conditions.

Proxy data is one important source of information on past climates and as a consequence about the sensitivity of the climate system itself. Numerical modeling offers a second, complementary approach, which is what we aim to focus on in this study.

One of our main motivations is that current simulations of the Eemian are not able to simulate a warming of 7-8°C over north western Greenland that is suggested by ice core proxy data. However, based on the fully-coupled simulations it is nearly impossible to identify the physical reasons why the models underestimate the Eemian-PI warming.

Consequently, sensitivity studies altering a certain component of the climate system are a very useful tool to determine physical processes which may have contributed to the Eemian warming observed in the proxies but missed by the model's response to the Eemian external forcing. In two previous studies (Merz et al. 2014a, 2014b) we have assessed the role of the ice sheet configuration for the Greenland temperature and associated moisture changes. Complementary, we investigate here whether local sea ice reductions also have the potential to cause a significant warming recorded in Greenland ice cores.

The hypothesis that sea ice-related processes are a likely candidate for the underestimation of an Eemian warming connects to the fact that there are clear model deficiencies in that coupled models tend to generate too much sea ice (already in the present day climate simulations) but to various degrees and in various regions. So, the question which is answered in this study is how much of the Eemian Greenland warming (in the fully-coupled simulations) may be due to the uncertainty arising from SST/sea ice distribution around Greenland.

In summary, we clearly focus in this study on process understanding rather than trying to simulate the "most accurate" Eemian climate in terms of sea ice, SSTs, SAT etc. We have made an effort to describe our goals and motivation more clearly in the introduction. For example, we have included an explicit list of the goals of the study.

*In summary, the goals of the study are as follows: (i) quantifying the atmospheric warming in and around Greenland related to uncertainty in the Eemian sea ice cover (the uncertainty results from the spread in sea ice configurations among fully-coupled models), (ii) determine whether a sea ice retreat in a particular region leads to a temperature signal recorded in Greenland ice cores such as NEEM, (iii) understanding the key processes that link the climate in Greenland with the sea ice in adjacent areas. Note, however, that we do not aim to propose the most likely sea ice cover for the Eemian based on these model simulations but rather like to show the consequences of one or the other scenario of sea ice coverage around Greenland.*

2. Modeling approach

(i) Initially you show that the low and high resolution models yield different results in terms of SST and sea ice in the North Atlantic. It is further mentioned that the low resolution model has known problems and does not simulate a reasonable PI climate in the North Atlantic sector (is this also true for the Eemian?). Despite this claim, the majority of the experiments and figures (according to Table 2) are based on results from the low resolution model. This seems like a very odd choice to me. If the model is biased and has known problems, why base almost all figures and analysis on data from this model? Are there even worse problems associated with the high resolution model? If not, can we expect different conclusions if the same analysis is performed on the high resolution data?

The majority of the experiments and results are based on the CCSM4 simulations which all use the same nominal 1° horizontal resolution and showed good ability in simulating the climate in the North Atlantic and in Greenland (Evans et al., 2013, Merz et al., 2013, 2014a, 2014b, Vizcaino et al., 2013). We just use the SSTs and sea ice from $EEM_1$ (formerly $EEM_{lowRes}$) as the basis for the sea ice shift experiments shown in the Sections 5.1-5.3.

Our interest lies on the climate response simulated in the shift experiments which is the difference between two simulations (before and after the shift) so any absolute biases (e.g., a possible overestimation of sea ice in the $EEM_1$ simulation) is not of great importance as it is removed. The only relevant effect is the position of the Eemian sea ice edge, which determines the area where our sea ice shift causes the largest heat flux anomalies. However, as described in Section 5.4, we have also conducted the same sea ice shift experiments using $EEM_2$(formerly $EEM_{highRes}$) as basis (and hence the SSTs/sea ice from the highRes CCSM3 simulation). Thereby we obtain very similar results for the shift experiments (e.g., compare EEM1/EEM2 numbers in Table 3&4) and all conclusions remain the same for either the shift experiments starting from $EEM_1$ and from $EEM_2$.

Note that we have revised the lowRes/highRes terminology for the CCSM4 simulations ($EEM_{lowRes}$ -> $EEM_1$, $PI_{lowRes}$ -> $PI_1$, $EEM_{highRes}$ -> $EEM_2$, $PI_{highRes}$ -> $PI_2$) to avoid further confusion. All CCSM4 simulations were carried out with the same resolution. We have adapted the descriptions in Section 2 and Table 1, respectively.

(ii) I am generally skeptical to the approach taken in sections 4.1 and 4.2 and I am afraid that we are not learning very much from this exercise. CCSM3 and CCSM4 are highly dependent models (e.g. Knutti et al., 2013) that are part of the same model family, meaning that the atmospheric components (CAM3 and CAM4) share the majority of the same code base. The biggest difference between the models is the deep convection scheme, which plays virtually no role in the latitude range of your focus. Consequently, the comparison of the two atmospheric models is largely redundant as you basically compare results from two simulations with almost the same model using identical forcing protocols. I argue that you can omit this whole comparison and just state that you use SST/SIC from CCSM3 in CAM4 and then prove that the simulated climates are reasonable with respect to reliable data. Also, the near surface temperature is not the best field to use to evaluate differences between AMIP simulations. If the model is capable of producing a realistic climate with realistic turbulent fluxes (e.g. near surface gradients), the near surface temperature is by definition largely similar to skin temperature and you basically prescribe the phenomena that you are investigating.

We are fully aware of the fact that CCSM3 and CCSM4 are similar models as they stem from the same model family and that the similarity in responses can be partly anticipated. However, the comparison in Fig. 2 of CCSM3 and CCSM4 simulations is to show the agreement between fully-coupled simulations and atmosphere-land-only simulations which use the SSTs/sea ice from the fully-coupled simulations. This illustrates that the SAT

differences between CCSM3$_{lowRes}$ and CCSM3$_{highRes}$ are fully explained by their differences in lower boundary conditions, while resolution plays a minor role. Moreover, the reproduction of the results of the fully-coupled CCSM3 in CCSM4 represents the transition to a consistent model design for all subsequent simulations. As we want to use atmosphere-land-only simulations (main part of study) to learn about processes which are relevant for the fully-coupled simulations we feel that Fig. 2 and the results presented in Sect. 4.1 and 4.2 are necessary and valuable.

Note also that we want to explain the reasons for the model spread in terms of EEM-PI warming: generally found among different climate models (e.g., here for two versions of CCSM3 in Fig 2e). The analysis of the reasons (and particularly the role of SST and sea ice changes) for the SAT pattern in 2e is difficult based on the CCSM3 simulations due to their various differences in the simulation settings (e.g., horizontal resolution or some differences in external forcings, see Sect. 2.1). Hence, using one single model version of CCSM4 and creating two pairs of PI and EEM simulations with identical settings except the prescribed SSTs and sea ice is a much more consistent approach to assess the contribution of SST and sea ice changes in the EEM-PIdiff SAT pattern (Fig. 2f). This is also clearly stated in the manuscript:

*With the two pairs of PI and EEM atmosphere-land-only CCSM4 simulations we create equivalents to the existing fully-coupled CCSM3 simulations. Hence, we can compute two realizations of the EEM-PI climate anomaly based on the exact same CCSM4 model and external forcings but differing in terms of prescribed SSTs and sea ice. Consequently, this setup eliminates uncertainties arising from different model physics and parameterizations at different resolutions (as it is the case in the fully-coupled CCSM3 simulations). This enables a more robust analysis of the impact of sea ice and SSTs.*

Theoretically, we could also have used other fully-coupled climate models (e.g., IPSL and HadCM) as a starting point:1) to compare their differences in EEM-PI climate anomalies (equivalent to Fig.2a,c,e) and 2) to use their SSTs and sea ice to force the atmosphere-land-only CCSM4to retrieve the contribution of the spread in SSTs and sea ice changes (equivalent to Fig.2b,d,f). Hence the choice to start with two versions of CCSM3 was due to the availability of these simulations but not to show specific processes explicitly valid within the CCSM model family. However, the results of our study are not affected by this choice and do not lose their generality.

As mentioned above, the comparison of Eemian proxy data with climate simulations (including the low resolution and high resolution CCSM3) has already been done in various studies (e.g., Lunt et al., 2013, Otto-Bliesner et al., 2013, Capron et al., 2014) and is not the focal point of this study. Hence, we don't want to repeat this comparison as we feel that it will lengthen the study without adding much novel information. In addition to acknowledging these studies in the introduction we have added a respective statement in the new model validation section to guide the reader to the respective references that compare the Eemian CCSM3 simulations (also the ones used in this study) with Eemian proxy records.

Lastly, we totally agree that the surface air temperature is largely determined by the surface heat fluxes over ocean/sea ice points and thus a respective signal in SSTs/sea ice concentration directly translates in a corresponding surface temperature signal. This is clearly no surprise to the reasons you mentioned. However, we are also interested in temperatures over land (in particular in Greenland), e.g., as displayed in Fig. 2. For the land points the influence of SST/sea ice changes is not as straightforward as for ocean points and hence worth a closer look (what is done in this study with the heat budget analysis). One key message from Fig. 2 is that the warming patterns in panels e) and f) (in CCSM3 fully-coupled vs. CCSM4 atmosphere only) do not only agree over the ocean but also to some degree over land points.

(iii) A large part of the analysis is based on differences between difference fields (EEM-PIdiff).These results are almost impossible to wrap ones head around and I wonder what we can learn from such a comparison, especially since the low resolution model has known biases. Also, it would help the interpretation of the results if you used the same color scale in all figures showing the same/similar quantities.

Since the analysis is based entirely on the nominal 1° CCSM4, where some simulations use boundary conditions of an earlier coarser resolution CCSM3 simulation, our results are minimally affected from model or resolution biases. All new simulations that we carried out for this study are classical sensitivity simulations where one single aspect of the model setup is changed at the time to isolate single dependencies and processes (here: the influence of sea ice/SSTs on the atmosphere).

The EEM-PI$_{diff}$ denotes the sensitivity of a single model (CCSM4) to the two respective sets of boundary conditions, lowRes or highRes. This is the familiar concept of a climate sensitivity, albeit in this case not related to $CO_2$. The quantity EEM-PI$_{diff}$ is the difference between these two sensitivities. We have revised the text of the manuscript to more clearly state the meaning of EEM-PI$_{diff}$ in Sect. 2.3:

*The difference between these two last EEM-PI anomalies themselves is referred to as EEM-PI$_{diff}$ which stands for the climate response related to the spread/uncertainty in the EEM-PI sea ice and SST changes.*

For a given variable the color scale is consistent except for Fig. 7 where the SAT range (which is from -5 to 5 °C in the other Figures) is from -3 to 12 C. We feel that adapting the range of Fig. 7 to -5 to 5 °C would be unfavourable to display the effect of the warming induced by the shift experiments. Furthermore, it is on purpose that the color scale in Fig. 1 for absolute SSTs is different from the color scale in Figs. 3, 5 and 6, showing changes in SSTs.

(iv) My main concern has to do with the sea-ice retreat experiments. First of all, the amount by which you shift the sea ice seems to be arbitrarily chosen and should be motivated.

The character (direction and magnitude) of the shift was chosen to resemble the EEM-PI$_{diff}$ sea ice anomaly in the respective region (compare Figs. 4 and 5). Hence, the shift is not arbitrary but rather chosen to mimic sea ice anomalies that are output of fully-coupled climate simulations. Recall that the EEM-PI$_{diff}$ sea ice anomaly can be interpreted as uncertainty/spread in EEM-PI change in sea ice based on the two versions of CCSM3.

We have made an effort to motivate the design of our sensitivity experiments better in the revised manuscript.

*In summary, our sea ice shift experiments are of idealized nature but the SIC and SST boundary conditions locally resemble fields from the fully-coupled CCSM3 simulations. The direction and magnitude of the shift are chosen to locally, i.e., either in the Labrador Sea or the Nordic Seas, mimic the difference between CCSM3$_{lowRes}$ and CCSM3$_{highRes}$ in order to disentangle their combined effect in EEM-PI$_{diff}$.*

Note that we also tried different magnitudes of the shift (not shown in the manuscript). However, the results of those simulations are in agreement with what is shown in the manuscript (e.g., dominance of NordS-shift effect over LabS-shift effect for Greenland).

Second, I am not convinced that these perturbation experiments are designed in a way that they will teach us anything useful about the last interglacial climate. In steady state (no drift due to external forcing) the circulation in atmosphere and ocean is by definition what determines the sea-surface conditions; the SST/SIC is essentially determined by the internal

heat flux (Qflux) in the ocean mixed layer and the balance between radiative and turbulent surface fluxes in the atmosphere (SST ~ SWnet –LWnet – LHflux – SHflux – Qflux). When you prescribe the sea-surface conditions and introduce local changes in the SST/SIC, you also introduce a local climate forcing that could never happen in the real world as it is not supported by the rest of the climate system (the open water that is introduced is not consistent with the general circulation).

We agree that any change to the coupled system will result in an imbalance and thus potentially invalidate the new solution. However, in the sea ice shift experiments, this only applies to the ocean circulation that is not explicitly simulated. The atmospheric surface climate and circulation will adjust to the prescribed SST/SIC. As shown in Fig. 2, the atmospheric response to a given SST/sea ice anomaly is very similar in a fully-coupled and in an atmosphere-land only simulation. Thus, we ensure that our atmosphere-only simulations are not in contradiction with the physically consistent coupled simulations that the boundary conditions were taken from.

Consequently, assuming that the prescribed SSTs/sea ice anomalies are reasonable because they are taken from a fully-coupled simulation, we have great confidence in the results of our atmospheric simulations. As explained in our previous response, the shift experiments are designed to locally resemble the sea ice/SST anomalies in the fully-coupled CCSM3 and thus these are physically possible sea ice/SST changes that were in the fully-coupled model provoked by changes in the external forcing.

If we assume that the sea-ice cover in the Labrador Sea collapsed (for whatever reason), the climate system would do everything it can to rebuild the sea ice over the next few seasons (as is evident from the almost 100W/m2 imbalance in sensible and latent heat fluxes that are reported in the analysis). If we instead assume that we could collapse the Labrador Sea ice and keep the region ice free, the rest of the ocean circulation (and atmospheric circulation for that matter) would have to be different to sustain the reduced sea ice; i.e. there would be changes in the SST field elsewhere and the turbulent fluxes would almost certainly be lower as sea-ice otherwise would form. I know that the chosen modeling approach is not new and that other people have done similar experiments before you (e.g. Deser et al., 2010), but I am concerned that this modeling approach does more damage than good in this particular study. I don't have a patented solution to the problem but I argue that it would be better to run a slab ocean model and alter the internal heat flux convergence in the mixed layer (in a conservative way so it doesn't introduce a global climate forcing) so that the sea ice retreats from the desired regions. This is arguably a better solution as the surface temperature and sea-ice margin are determined by the surface energy balance, which means that it is theoretically possible to construct a climate where there is no sea ice in the desired regions but you have sea-surface conditions that are in balance with the circulation and external forcing. Whether or not this climate state is realistic is of course another question.

We agree that the sea ice shift experiments partly break the physical consistency of the coupled system. This is a general and irremediable aspect of atmosphere-only simulations. However, it also represents an opportunity to investigate the impact of changes when applied within physically reasonable limits. In this context, we argue that the shifted sea ice edge does not surpass these limits, because it is not generally inconsistent with possible states of the ocean circulation as shown by the fully-coupled CCSM3 simulations. The LabS-shift and NordS-shift experiments are designed to resemble this coupled simulation regionally in order to disentangle the effects of one ocean basin versus the other. A slab ocean model is not suited here because it only includes meridional heat transport in the ocean. In the case of the Labrador Sea, the zonal ocean heat transport is very important.

To illustrate that large surface heat fluxes are not only an artifact of atmospheric-only simulations but also possible in CCSM4 fully-coupled simulations we show here sensible and

latent heat flux anomalies for a LGM compared to a PI simulation (Fig. R1). The diagnostics are provided by the NCAR: http://www.cesm.ucar.edu/experiments/cesm1.0/)

Note that such diagnostics are not available for an Eemian simulation. Similar to our atmospheric-only CCSM4 simulations, the heat flux anomalies from the fully-coupled model shown below are the result of distinct changes in SSTs and sea ice (see also Fig. R2) originally caused by changes in external forcing (here LGM vs. PI, in our manuscript EEM vs. PI). More precisely in the LGM the sea ice strongly increased in the Labrador Sea and the Norwegian Sea (Fig. R2) leading to distinct negative heat fluxes in these regions. At the same time adjacent areas in the North Atlantic show distinct positive heat flux anomalies building the dipole structures alike the ones found in our atmospheric simulations.

[Figure]

*Fig. R1: Winter mean (DJF) LGM minus PI change in (top) sensible heat fluxes (W/m²) and (bottom) latent heat fluxes (W/m²) based on CCSM4 fully-coupled simulations.*

[Figure]

*Fig. R2: Winter mean (DJF) LGM minus PI change in (left) sea ice concentration (%) based on the same CCSM4 fully-coupled simulations as used for Fig. R1.*

Furthermore, a very recent study by Petrie et al., 2016 shows for another fully-coupled model that the climate response in a sea ice sensitivity experiment leads to considerable surface heat flux anomalies. This is another good indication that the climate response provoked by our shift experiments is not an artificial result caused by the atmospheric-land-only setup. We have included a reference for this paper in the revised version of the manuscript

3. Interpretation of results

(i) Following the previous comment, it is not at all surprising that you get very strong turbulent fluxes in the sea-ice sensitivity experiments. The prescribed SST/SIC implies that the climatological atmospheric circulation is more or less determined by the prevailing sea-surface conditions. When making local changes to the SIC and prescribe SSTs that are not consistent with the circulation, you introduce regions where the climate "wants" to have sea ice, as cold air is advected over open water, but the prescribed sea-surface conditions prevents it from forming. This gives rise to artificial vertical gradients and turbulent fluxes that would never happen in nature as the SST/SIC would respond and go back to an ice covered state. This in turn induces an anomalous atmospheric circulation that has no real world analogue, at least not in a climatological state which is what is investigated here.

Please see our responses to your comments above that address similar issues.

As detailed in our reply above, we understand this concern and we agree that physical inconsistencies due to the *uncoupling* of the formerly consistent ocean-atmosphere system requires great care and a discussion of what can be concluded from such idealized numerical experiments. The latter is accounted for by a respective paragraph in the final summary of the paper (page 21, line 7pp).

However, we feel that the comment is too general in its criticism. Regions where very cold continental air meets open ocean surfaces do exist in the real world. They are not always a model artifact. Specifically, in the Labrador Sea, it is important to note that the coupled version of CCSM4 does simulate open waters here (e.g., Jahn et al., 2012) that produce a strong air-sea heat flux. CCSM4 uses the same atmosphere model as our atmosphere-only simulations, CAM4, which illustrates that a situation similar to our idealized LabS-shift experiment is not physically impossible in this model.

(ii) In my mind, one of the most interesting results in the whole paper is the changes in the lower tropospheric wind field (Fig. 9) that results from manipulating the local SST/SIC in the North Sea. However, no explanation is provided as to why the wind field changes the way it does. I want to see a dynamic argument made for the somewhat counterintuitive response where the lower tropospheric winds impinge on Greenland from seemingly the wrong direction; SE instead of NE where the forcing is located.

Thank you for pointing at this interesting issue. The NordS shift leads to the warming pattern displayed in Fig. 7a that features the strongest warming east of southern Greenland extending towards Svalbard. The baroclinic response to this surface warming is a surface low pressure anomaly is strongest next to southern Greenland and has a secondary maximum further north in the middle of the Greenland Sea (see Fig. R3). Correspondingly, anomalous cyclonic winds are observed encircling these pressure minima (Fig. 9c). Since the pressure anomaly northeast of Greenland is relatively far away from the Greenland coast the anomalous winds in northeastern Greenland are northerly rather than causing a heat transport towards Greenland. In contrast, the cyclonic flow anomaly centered in the proximity of southeastern Greenland leads to anomalous onshore flow. As the onshore winds are overflowing the steep slopes of the topography in southeastern Greenland, the horizontal wind anomalies also cause considerable vertical wind anomalies (shaded in Fig. 9c). In summary, the wind anomalies in southeastern Greenland lead to a distinct weakening of the Greenland anti-cyclone (local background flow) whereas the baroclinic effect causing a

pressure anomaly in the Greenland Sea is too far off-shore to substantially alter the winds in northeastern Greenland.

[Figure]

*Fig. R3: Winter mean (DJF) sea-level pressure (SLP) response (shaded) to NordS-shift. The contour lines indicate the SLP pattern before the shift dominated in the North Atlantic by the Icelandic Low. Stippling denotes significant SLP changes at the 5%-level based on t-test statistics.*

We have revised the description of Fig. 9 in the manuscript to better explain these results.

Line-by-line comments:

Page 1, line 1: I would be careful suggesting that the Eemian is a possible analog to the climate in the near future. The Eemian was warm primarily as a result of increased insolation whereas future climates are warm because of higher greenhouse gas concentrations. The former only plays a direct role during parts of the year (in high latitudes) whereas the latter influence the longwave radiation in all seasons.

We are fully aware of the different causes and impacts between the Eemian and the current/future warming. Still, the Eemian period remains a valuable test bed period for studying the dynamics of the high-latitude climate system for atmospheric/oceanic conditions warmer than present. We have revised the beginning of the introduction to make it clearer.

*The last interglacial (ca. 129-116 ka) also known as the Eemian is often regarded as a possible analogue for future climate as it stands for the most recent period in the past characterized by a warmer than present climate. In contrast to the future year-round warming induced by rising greenhouse gas (GHG) concentrations, the Eemian warming, driven by anomalous orbital forcing, was mostly confined to the summer season and the extra-tropics.*

Page 1, line 19: This time interval contains both warm and cold phases.

We changed the definition of the last interglacial (Eemian) to 129-116 ka which corresponds to the definition in IPCC AR5. The last interglacial is clearly known as warm period (as it was much warmer than in the period before and after). The time interval of 129-116 may include parts of the transition phases with the preceding/following glacial but defining the exact length of any glacial/interglacial period is a challenge on its own.

Page 4, lines 1 & 3: Write out the equivalent grid resolution for T31 and T85.

We have added this information in the manuscript. T31 is equal to a nominal 3.75° resolution whereas T85 corresponds to 1.4°.

Page 4, lines 19-25: This is more of a curious comment than anything else but when you regrid the T31 SST/SIC to the T85 grid, you implicitly introduce an outline of the T31 grid but at the higher resolution. Do you have a feeling for if this will influence the results?

This is probably a partial misunderstanding related to the confusion about the resolution of our simulations that was mentioned above. We extrapolate the SSTs from both types of CCSM3 simulations, i.e. 3° grid (T31x3 simulation) and 1° (T85x1deg), across all land points before regridding this "land-less" new field to the 0.9°x1.25° resolution of CCSM4. The land mask of CCSM4 is added in the last step, so there is no "outline" of the original 3°/1° land mask.

Page 5, line 24: How does the absence of inter-annual variability in the SST/SIC degrade the representation of the storm track? Add a sentence explaining that.

The study Raible and Blender (2004) shows that the Pacific storm track is shifted north in the absence of inter-annual SST/SIC variability (mainly due to the missing ENSO variability). In the Atlantic there are also changes, in particular more storms move zonally and less to the Northeast. Please note that using a mixed ocean model instead leads to similar behavior of the storm track as for simulations with no inter-annual SST/SIC variability.

We have revised the respective sentence in the manuscript to make this point clearer:

*The absence of inter-annual variability in the ocean/sea ice representation, however, can be a drawback with respect to atmospheric dynamics, e.g., causing a too zonally oriented storm track in the North Atlantic (Raible and Blender, 2004).*

Page 6, line 7: The -1.8°C temperature is only used for the SSTs underneath sea ice. The actual temperature of the sea ice is determined by the local surface energy balance, which is generally much lower. It is therefore a bit misleading to use the SST as a measure of the surface temperature and I suggest showing the actual surface temperature instead.

Your comment is completely valid for areas with partial sea ice coverage in terms of that the atmosphere is feeling the surface temperature of the ice according to the local surface energy balance (calculated by the thermodynamic module of the sea ice model CICE). Nevertheless, we prefer to show SSTs in Fig. 1,3,5,6 as it can be shown for both partially ice-free and fully ice-free regions rather than showing the ice temperature for the small areas with partial sea ice coverage what would complicate the illustrations.

Eventually, we are interested in the SSTs rather than the ice temperatures as the SSTs provide information about how much energy from the surface ocean is available for the atmosphere. Further, the exchange of heat between the ice surface/ocean surface and the low-level atmosphere is illustrated by the surface heat fluxes and SAT in Figs. 5&6.

Page 7, end of section 4.1: Determine whether the difference in temperature signal is due to the PI, Eemian or both climate states when going to the lower resolution.

The temperature signal assigned to EEM-PI$_{diff}$ is by definition a combination of both climate states and all four simulations involved. The positive EEM-PI$_{diff}$ temperature signal tells us that the difference between the absolute temperatures in EEM$_{lowRes}$ and EEM$_{highRes}$ is larger than the difference between the absolute temperatures in PI$_{lowRes}$ and PI$_{highRes}$. Physically, EEM-PI$_{diff}$ indicates how diverse the two CCSM3 versions are with respect to climate response to the same (EEM-PI) external forcing. Note also that any differences between the

two CCSM3 versions related to different mean climate biases are removed in EEM-PI$_{diff}$ as we are comparing relative climate signals rather than looking at the absolute (e.g. Eemian) climate.

Page 7, line 12: with and an excessive... -> with an excessive...

Done

Page 7, line 22: CCSM4 and CCSM4 -> CCSM3 and CCSM4

Done

Page 8, line 1: What is the relationship between the SST and the sub-polar gyre?

The circulation of the subpolar gyre influences SSTs in several ways. Firstly, a stronger gyre results in a stronger Irminger Current that transports heat and salt south of Iceland in a westward direction. While this causes a weak direct warming, the salt transport is more important. The enhanced influx of saline waters into the relatively fresh Labrador Sea strengthens deep convection in this region. Since subsurface waters are warmer than the strongly cooled surface waters in this region, this second effect also results in a warming. Lastly, in broader terms, the gyre heat transport dominates over the overturning heat transport in the subpolar latitudes of the North Atlantic. Thus, a stronger subpolar gyre transports more heat northward across the entire width of the ocean basin.

We have revised the manuscript regarding changes of the subpolar gyre to make this point clearer.

Page 8, line 11: Show the PI SST, it is important for the story!

As stated in our response to one of your main comments we focus in our study on EEM-PI changes and sensitivity experiments whereby the absolute PI SSTs (and possible biases) are of low order importance as they are removed by looking at the climate anomalies. Thus, we feel that showing absolute PI SSTs is not of great importance for our results and won't help the reader to better understand the key results of our paper. Note the biases in the CCSM3 SSTs for PI are now better discussed in the new section (Sect. 2.3) dedicated to model validation as well as in the revised Sect. 4.2.

Page 8, line 18: particularly strong on SAT above oceanic grid cells... Don't you use identical SST/SIC in CAM3 and CAM4? If so, you expect to see very similar SAT as it represents the temperature just above the ocean surface.

Yes, the SSTs in the CCSM3 simulations (fully-coupled) and the CCSM4 simulations (atmosphere-land-only) are identical and hence we expect very high similarity.

Page 8, line 19: How much is the winter insolation decreased in winter?

The magnitude of the decrease in insolation depends on the latitude and the month of the year (e.g., see Fig. 1 in Lunt et al., 2013). We have added the number for 50°N in the manuscript.

Page 9: What can we possibly learn from ($\Delta$1 -$\Delta$2) when at least one of the $\Delta$#s have known biases?

Even though both models (1 and 2) might have biases in terms of absolute values, we can investigate the EEM-PI changes in both models (i.e., Δ1 and Δ2) as the model biases are removed by looking at differences between two simulations with the same model. This is a very common approach in climate modeling studies. In this study, we are clearly interested in differences (Δ) rather than absolute values because we want to study the relationship between the changes among different fields (e.g. ΔSSTs, Δsea ice, ΔSAT etc.). These changes (Δ) all are results of the physical principles employed in the climate model. Assuming that the physics in the model are correct, we can learn about the importance of single processes and their interactions with other components in the complex climate system.

Consequently, both Δ1 and Δ2 represent valid estimates for EEM-PI climate anomalies based on the CCSM4 model physics and thus we can use EEM-PIdiff (i.e. Δ1 - Δ2) to assess the impact of the prescribed sea ice and SSTs as in both pairs of simulation (i.e., Δ1, Δ2) all other settings are identical. EEM-PIdiff (i.e. Δ1 - Δ2) can be regarded as the climate response linked to the uncertainty/spread in EEM-PI sea ice and SST changes by the two versions of CCSM3.

Page 9, line 6: surface ocean -> ocean surface

Done

Page 9, line 11: Which terms does Qnet contain? Radiative fluxes? Turbulent fluxes? Internal heat sources in the ocean? A combination of all or a subset of the above?

Qnet refers to the atmospheric energy balance and is defined as the sum of sensible heat, latent heat and longwave net radiation whereas we omit SWnet in the definition due to reason stated in the manuscript (page 10, lines 16-19). We have also added the definition of Qnet to the text of the revised version.

*Qnet is defined here as the sum of sensible heat, latent heat and longwave radiation.*

Page 9, lines 21-29: You have prescribed SST, which means that you easily get artificial turbulent surface fluxes as the ocean temperature acts as an infinite source and sink of energy (sign depends on atmospheric conditions).

In all atmospheric simulations with prescribed SSTs, the SSTs are static and hence the atmosphere finds its own equilibrium given the regional heat input by the ocean surface. In agreement with your comment and as described in the manuscript (page 9, lines 21-29), the surface heat flux response to an initial SST change is therefore stronger than in a fully-coupled simulation run into its equilibrium. However, keep in mind that the purpose of the atmospheric CCSM4 simulation is to mimic the sea ice, SST changes (and consequently also the resulting surface energy flux changes) found in the fully-coupled CCSM3 simulations (Fig. 5). As the SST and sea ice anomalies stem from fully-coupled CCSM3 simulations that each were run into their respective equilibria, these anomalies are based on physical mechanisms.

We further feel that the physical inconsistency in the atmospheric-only simulations is a small price to pay for the flexibility to investigate a specific detail of the coupled system.

Please also refer to our responses to your major comments.

Page 10, line 5: Write out the resolution used in the "Shift" experiments.

We have added a new paragraph at the beginning of Section 5 to better motivate the purpose of this section. This new paragraph includes also a reference to the model

description sections where the reader can look up that all shift experiments were performed with the CCSM4 model using the 0.9°x1.25° resolution.

Page 10: Why do you use the low resolution model when it has known biases?

We use the atmosphere-land-only CCSM4 model which has a nominal 1° (0.9°x1.25°) resolution for all sea ice experiments and prescribe SSTs/sea ice from both the low resolution and the high resolution model as input (see Chapter 5.4 for an overview of all simulations). However, as we are mostly interested in changes between two simulations, the absolute nature of the SSTs/sea ice input fields is of lower order importance. Note that the sea ice shift simulations have been repeated starting from the unperturbed conditions of the high resolution coupled CCSM3 simulation (described in Section 5.4). The results and conclusions from these additional simulations are virtually identical with the shift-experiments starting from the CCSM3 low resolution SSTs/sea ice.

Page 10: Fixed SST is almost certainly the source of the strong turbulent fluxes that are highly artificial as they would never happen in nature in the way described in the manuscript, at least not over a long period of time.

Please see our responses to similar comments above.

The idealized SST and sea ice fields are artificial but they do resemble the regional conditions in the EEM-PI CCSM3 coupled simulations (e.g., compare Fig. 5a and 6a,e) and therefore are not fundamentally at odds with a physically consistent system. Note that strong surface heat fluxes are also found in observations and fully-coupled simulations (Bates et al., 2012, Petrie et al., 2015) as well as shown above in Fig. R1.

Page 11, lines 1-3: Why does the warming spread over Greenland? Comment on changes in atmospheric circulation.

The role of changes in the atmospheric circulation is discussed in Sect. 5.1. It is shown that in the NordS-shift experiment the Greenland anti-cyclone is weakened allowing warm air from the Nordic Seas to be advected towards Greenland's interior. In contrast, for the LabS-shift experiment the Greenland anticyclone remains strong and fosters the cold isolated climate in Greenland (as seen in the PI and EEM simulations).

Page 11, line 8-10: Eq. 1 is written in advective form, not flux form. The terms you refer to are therefore showing temperature advection and not heat flux convergence.

Thank you for spotting this. We have changed it to 'horizontal and vertical temperature advection'.

Page 11, lines 16-19: Are you talking about month to month variability or the climatology? The terms have to be identically equal to zero in the latter if the model is in balance.

See next answer

Page 11, line 20: The temperature tendency has to be identically zero for the model to be in balance. You are looking at a climatology after all, or...?

Yes, we are looking at climatology. Looking at the values of the temperature tendency, those are actually 7-10 magnitudes smaller than the other terms of the energy balance so virtually

zero. We have changed the statement to "the total temperature tendency is zero" as this seems justified by this very small values and avoids confusion.

Page 12, line 6: How much is actually resolved at T31?

Note that all CCSM4 simulations (for which the heat budget calculation is applied) have 0.9°x1.25°(not T31) resolution which corresponds to a grid space in Greenland of ca. 50km.

Page 12, line 13: How does that hang together with the enormous increase in LH flux?I would expect to see a great moistening of the atmosphere when the LH flux increases that much, which in turn increases the cloudiness.

The moisture released by the positive latent heat flux anomaly is constantly transported away by enhanced moisture advection (see Fig. 10). Hence, the increase in atmospheric humidity above the moisture source region is limited as is the increase in cloudiness.

Page 12, line 33-Page 13, line 9: This paragraph is very confusing because you first talk about what you expect to see and then you show that the expected circulation is in fact not true.

We agree that this paragraph indeed was a bit confusing and it has been revised accordingly.

Page 12: What happens to mid- and upper tropospheric winds in these experiments?

The winds in the mid- and upper troposphere for the LabS-shift experiment show no significant changes. For the NordS-shift experiment we observe a high pressure anomaly above the Nordic seas and Greenland leading to anomalous cyclonic flow at these levels. This response is in agreement with previous studies, e.g., Deser et al., 2007.

Page 13, line 20: I don't see a southeastward transport in the figure.

Has been changed to "eastward".

Page 13, lines 20-23: Is this also true in these experiments? Have you done the proper analysis or is it just a conjecture?

We haven't performed a cyclone analysis, which is beyond the scope here, but it is well-known from the literature (e.g., Tsukernik et al., 2007, Hutterli et al., 2005).Note also that the results by Hutterli et al., 2005, which showed the relationship between Greenland precipitation and cyclones/circulation patterns in ERA-40, has been confirmed to be valid as well in the CCSM4 model (Merz et al., 2013).

Page 14, lines 15-19: This is the heart of my concern. Everything in the climate system acts to build sea ice where it has been removed but the prescribed SST/SIC don't allow the sea ice to regrow. Since the summer temperature is higher, there will not be any regrowth in the summer season and you don't see equally outrageous turbulent fluxes.

Please see our responses to your main comments on our thoughts why we feel that the sensitivity experiments are still valid.

Page 14, lines 27-34: This is not very surprising either. There is a prevailing southwesterly flow over the northeastern Atlantic, meaning that warm and moist air is advected over the region where you remove the sea ice. There is thus a smaller "urge" for the climate system to regrow sea ice there and you don't see equally large turbulent fluxes.

Figures 4 and 5 show that in both regions (LabS and NordS) removing sea ice leads to distinct winter heat flux anomalies as in both regions cold air is exposed to a relatively warm sea surface. We agree with the reviewer that the winter temperatures in the Labrador Sea are even colder than over large parts of the North Atlantic due to the local advection of cold air from the American continent in contrast to relatively warmer air masses moving eastward across the Atlantic. Nevertheless, the different magnitudes (in LabS vs. NordS) in winter heat flux anomalies shown in Fig. 11 mostly relate to the chosen boxes across we calculate the averages plotted in Fig. 11. As stated in the manuscript (page 6, lines 5pp) negative heat flux anomalies stemming from the dipole effect (Fig. 5) are included in the NordS box but not in the LabS box. For the definition of the boxes please refer to Fig. 1a,b.

Page 17, line 34: "statistically insignificant warming" sounds strange. Rewrite the sentence in a way that allows you to use something like "not significantly significant".

Done

*All LabS.-shift experiments result in a statistically not significant warming of at most 0.3°C.*

Page 18, line 9-10: Have you adjusted the Greenland elevation in these simulations?

Greenland is set to present-day conditions in all experiments presented in this study. Please refer to Merz et al. 2014a,b for results of another set of CCSM4 simulations which test the impact of a modified Eemian Greenland ice sheet.

Page 18, line 15: A 3.1°C temperature difference could in principle be due to a lowering of the ice sheet. Since the sea level was quite a bit higher in the Eemian, this is not a bad first guess that could be explored in a greater detail in the manuscript.

Please see our response to your previous point. We have also clarified the respective statement in the text to make this point clearer.

*Depending on the actual ice sheet topography this leads to an additional annual mean warming of up to 3.1°C at pNEEM (altitude-corrected) resulting from changes in Greenland's surface energy balance (Merz et al., 2014).*

Page 18, lines 29-34: This section is a bit speculative. Maybe you can extend the discussion to include the importance of precipitation seasonality and the temperature inversion relationship recently discussed by Pausata and Löfverström (2015).

Please note that the statements in our manuscript refer to the study by Sime et al. (2013). We have extended the discussion of the d18O-interpretation to mention the possible effects reported by Pausata and Löfverström (2015).

*However, a meaningful interpretation of the NEEM d18O record is further complicated by the fact that the Eemian warming in Greenland mainly occurs in summer (due to orbital forcing) but d18O is rather tied to winter temperatures (Sjolte et al., 2014). Further, there are possible interferences with changes in precipitation seasonality and the inversion temperature relationship (Pausata and Löfverström, 2015).*

Page 19, lines 20-23: You haven't really shown or discussed any proper atmospheric dynamics in this paper. The main focus is on the turbulent fluxes that no doubt will influence the atmospheric circulation. This has not been shown properly though so this statement is merely a conjecture.

We do not agree that this is valid as parts of Section 5.1 are clearly dedicated to changes in atmospheric circulation (i.e., the Greenland low-level winds).

Figures: Use the same colorscale in all figures showing the same/similar quantities.

Please see our response to the major comment on the same issue.

Figure 1: Consider changing the transect to a different color. It is very hard to see black on top of dark blue.

Done

Figure 2: Validate the model by showing full fields as well as a climate reconstruction.

As stated in our response to your main comment #1 we feel that a lengthy analysis of the full fields and a comparison with climate reconstructions is beyond the scope of this study and has already been done in earlier studies. We have made an effort to better discuss the results of existing studies in the revised version of the manuscript.

Figure 3: The large sensitivity of SIC to the model resolution is curious. Is there any proxy data you can compare this with?

To our knowledge, there is no sea ice proxy available for that period which could be used to judge about either Eemian sea ice mask produced by the two model versions.

Figure 3: What is the purpose of this figure when Fig. 4 shows almost exactly the same thing, though extended to show the response over land as well?

Figure 3 shows sea surface temperature (SST) and sea ice concentration (SIC) whereas Fig. 4 is showing surface air temperature (SAT), so they are not showing the same fields. It is worth showing both the SSTs/SICs (i.e. here used as a forcing as they are prescribed) and the SATs (i.e. the temperature response in the low-level atmosphere). The comparison highlights how strongly the atmospheric temperature response is related to changes in SSTs and sea ice (not only above ocean points but also in Greenland!)

Figure 5: Number labels have not been defined.

The number labels of the contours in Fig. 5a are defined in the caption.

Figure 10: I am curious as to why there are such large differences in e.g. the NorwegianSea and southwestern Greenland?

We are not sure what differences the referee refers to but if this comment concerns the differences between Fig.10c,d it is likely that our calculation of the moisture fluxes (through finite differences) is not able to fully close the moisture budget diagnosed by P-E.

Table 3: Write out the abbreviations and resolutions in the caption.

We have clarified in the caption that this table is showing the results from CCSM4 simulations (which all are performed with the same resolution). We also added a notification that the simulations are explained in Sects. 2.2 & 5.4.

**Additional references used in response (and not included in manuscript)**

Bates, S. C., Fox-Kemper, B., Jayne, S. R., Large, W. G., Stevenson, S., and Yeager, S. G., Mean Biases, Variability, and Trends in Air–Sea Fluxes and Sea Surface Temperature in the CCSM4, Journal of Climate, 25:22, 7781-7801, 2012

Jahn, A., and Coauthors. Late-twentieth-century simulation of Arctic sea ice and ocean properties in the CCSM4. Journal of Climate, 25, 1431–1452, 2012

Deser, C., R. Thomas, and S. Peng, The transient atmospheric circulation response to North Atlantic SST and sea ice anomalies, Journal of Climate, 20, 4751–4767, 2007

[revised manuscript text omitted]
 with respect to their EEM-PI  climate anomaly, i.e., the change in Eemian climate  compared to pre-industrial, which avoids possible caveats associated with mean climate model biases and the calibration of proxies to an absolute level. Equivalently, we focus in this study on the simulated EEM-PI climate anomaly to quantify the Eemian state of any target climate variable. More precisely, we define a set of climate anomalies listed in Table 2. Based on the CCSM3 simulations we compute EEM-PI$_{lowRes}$ and , EEM-PI$_{highRes}$ differing in horizontal resolution as well as other minor settings as explained in Sect. 2.1. Similarly, we calculate two EEM-PI anomalies  based on same atmosphere-land-only CCSM4 setup but differing with respect to the origin of the presribed lower boundaries (either CCSM3$_{lowRes}$ or CCSM3$_{highRes}$). The difference between  these two last EEM-PI anomalies themselves is referred to as EEM-PI$_{diff}$  which stands for the climate response related to the spread/uncertainty in the EEM-PI sea ice and SST changes. Besides, we use the terms LabS-shift and NordS-shift for the comparison of the  EEM experiments including a regional  shift in lower boundary conditions compared to  the reference experiment (i.e., the situation before the sea ice shift).

**3 A new type of  idealized sea ice sensitivity experiment**

Various types of sea ice reduction experiments have been presented in previous studies (e.g., Smith et al., 2003; Deser et al., 2010; Petoukhov and Semenov, 2010). A prominent approach is to implement an observed or simulated minimum sea ice cover (e.g., Smith et al., 2003; Alexander et al., 2004) or an altered sea ice climatology that exhibits a retreated sea ice cover compared to its reference (e.g., Higgins and Cassano, 2009; Deser et al., 2010). An alternative option is to artificially reduce the SIC in a target region to a certain percentage (e.g., Petoukhov and Semenov, 2010). These experimental designs have in common that they use a repeating seasonal cycle of SICs (and SSTs) and thus are not accounting for inter-annual variability. The absence of inter-annual variability in the ocean/sea ice representation, however, can be a drawback with respect to atmospheric dynamics, e.g., causing a  too zonally oriented storm track in the North Atlantic (Raible and Blender, 2004).

To avoid this deficiency and also to be consistent with the pre-industrial and Eemian CCSM4 simulations, which use time-varying SSTs and sea-ice  (including inter-annual variability), the "sea ice shift" approach is applied (illustrated in Fig. 1). We take the monthly varying lower boundary conditions previously used for CCSM4 EEM$_{lowRes1}$ and modify the values in the target region by shifting them along a certain axis. For the EEM$_{LabS}$ simulation we shift all SIC values in the LabS domain northwestward (see Fig. 1a). In technical terms, all values within the solid  green boxes in Fig. 1a are replaced point-by-point by the values within the dashed  boxes. Values in the green shaded area are linearly interpolated to guarantee a smooth transition with the adjacent regions. Similarly, for EEM$_{NordS}$ we shift all SIC values in the NordS domain (dashed  green box in Fig. 1b) northwards. As illustrated by the 50% sea ice contour lines in Fig. 1a,b this approach results in a local sea ice retreat in the perturbed (dashed contour) compared to the reference simulation (solid contour). Note that in all cases we only change the sea ice area, whilst the sea ice thickness is fixed at 2 m throughout the Arctic which is the default for CCSM4 simulations with prescribed lower boundary conditions.

A key consideration in all types of sea ice sensitivity experiments is the prescription of corresponding SST changes. For example, grid cells becoming ice-free are exposed to solar radiation and thus local SSTs likely increase compared to the typical freezing point temperature of $-1.8°C$ of an ocean grid cell completely covered by ice. Vice-versa, the sea ice retreat itself can be caused by a warming of the surface ocean, hence a reduction in SIC is usually accompanied by an increase in SSTs. This strong relationship between SST and SIC in marginal sea ice areas is also found in the input data used for EEM$_{lowRes1}$ ( dashed lines in Fig. 1c,d) along the transects A→B and C→D  in the two target regions. In order to account for this strong link between the sea ice cover and SSTs, we shift the SSTs in the same way as the SICs (see  solid lines in Fig. 1c,d). This approach seems particularly reasonable for the LabS region where we find gradual changes along the transect (Fig. 1c). Hence, the northwestward LabS-shift in EEM$_{LabS}$ can be understood as a warm water inflow into the LabS area (see SSTs in LabS in Fig. 1a compared to Fig. 1b) resulting in a coherent sea ice retreat. In contrast, the situation in the Nordic Seas is more complex ( Fig. 1d) as the northward shift in SSTs corresponds to a displacement of local ocean currents with a nonparallel orientation to the C→D axis along which we apply the shift. For example, the northward NordS-shift results in a removal of the cold East Greenland current in EEM$_{NordS}$ (see SSTs in NordS in Fig. 1b compared to Fig. 1a).

Additionally, we generate a second pair of LabS- and NordS-shift experiments (termed EEM$_{LabS\ ICE}$ and EEM$_{NordS\ ICE}$) for which we only shift the SIC (equivalently to EEM$_{LabS}$ and EEM$_{NordS}$) but not the SSTs. Hence, this second approach avoids a possibly unrealistic warming of the surface ocean but, on the other hand, violates the obvious SST-SIC relationship revealed in Fig. 1c,d. Thus, EEM$_{LabS\ ICE}$ and EEM$_{NordS\ ICE}$ can be understood as experiments providing the lower range in terms of atmospheric response to a prescribed sea ice retreat. A detailed discussion of the atmospheric response to different experimental designs is presented in Sect. 5.4.

In summary, our sea ice shift experiments are of idealized nature but, nevertheless, the resulting SIC and SST anomalies resemble EEM-PI changes simulated by the fully-coupled CCSM3 (discussed in Sect. 5). More precisely, the direction and magnitude of the shift are chosen to locally (either in the LabS or NordS area) result in the same forcing to the atmosphere by

lower boundary conditions as in EEM-PI$_{diff}$. Hence, based on the shift experiments we can further assess the climate response related to the uncertainty in the EEM-PI sea ice and SST changes resulting from the spread among the fully-coupled models.

**4 Simulated Eemian warming: importance of sea ice and SSTs**

**4.1 Atmospheric temperature response in fully-coupled CCSM3 simulations**

5    The  first part of our analysis  assesses the uncertainty of the Eemian warming as suggested by the spread among state-of-the-art climate models. This relates to the model-intercomparison study by Lunt et al. (2013) which showed that the EEM-PI annual mean atmospheric warming (Fig. 5 therein) strongly varies among different  models and even applies to two EEM-PI  simulations with the same climate model but different model versions (denoted as CCSM3_Bremen and

10  CCSM3_NCAR  therein). Here we show a likewise comparison of the EEM-PI temperature response of two versions (EEM-PI$_{highRes}$ Fig. 2a)  and EEM-PI$_{lowRes}$ Fig. 2c)  of fully-coupled CCSM3 simulations

15  , previously introduced in Sect. 2.1.

CCSM3 EEM-PI$_{highRes}$ exhibits a distinct warming in the NH high latitudes with the strongest signal occurring in an area including the Arctic, Greenland and the North Atlantic (Fig. 2a). Significant warming but of smaller magnitude is further found in Europe and most of North America. On the contrary, the CCSM3 EEM-PI$_{lowRes}$ warming is very limited in terms of magnitude and spatial expansion (Fig. 2c). In fact, large areas of the NH experience an annual mean cooling. The difference

20  between the two EEM-PI warming patterns (Fig. 2e) illustrates a stronger warming of EEM-PI$_{highRes}$ than EEM-PI$_{lowRes}$ in almost the entire NH, but most distinctively over the Arctic and the North Atlantic ocean.

The  main reason for the remarkable discrepancy in EEM-PI warming among the two pairs of CCSM3 simulations  is likely the different horizontal resolution as it has been revealed that the low and high resolution versions of CCSM3 show distinct differences for various climatic features even under present-day

25  conditions (Yeager et al., 2006). Hence, though both CCSM3 versions share the majority of their code, they should be regarded as two different models. 
[revised manuscript text omitted]
$_{\text{LabS}}$ and EEM$_{\text{NordS}}$. Comparing the temperature response of EEM$_{\text{LabS}\text{LabS2}}$/EEM$_{\text{LabS}}$ and EEM$_{\text{NordS}\text{NordS2}}$/EEM$_{\text{
[revised manuscript text omitted]
$_2$ eand (,,fwithuseThe top row shows the resultsthe highRes experimentsmiddle row those fromlowRes experimentsbottom row their differences, respectivelythe top and middle row~~ a)-d) denotes EEM-PI changes significant at the 5% level based on t-test statistics applied to respective annual mean SAT time series.

[Figure]

**Figure 3.** CCSM3 Eemian minus pre-industrial (EEM-PI) seasonal mean sea surface temperature change (SST, shaded) and EEM (solid) vs. PI (dashed) 50% sea ice concentration (SIC) contours. The top row is based on the  (1°) simulations and the bottom row on the lowRes (3°) simulations, respectively. Note that these SST/SIC fields are used as lower boundary conditions for the  atmosphere-land-only CCSM4 simulations.

[Figure]

**Figure 4.** CCSM4 Eemian minus pre-industrial (EEM-PI) seasonal mean surface air temperature (SAT) change. The top row shows the result from the  1 experiments, the middle row the  2 experiments and the bottom row their differences, respectively. Stippling in the top and middle row denotes EEM-PI changes significant at the 5% level based on t-test statistics.

[Figure]

**Figure 5.** CCSM4 EEM-PI$_{diff}$ response in winter (DJF) mean a) sea surface temperature (SST, shaded) and sea ice concentration (SIC, contours), b) net surface energy flux (Qnet), c) sensible heat flux (SHF) and d) latent heat flux (LHF). Negative sea ice anomalies in a) are dashed and the contour interval is 10%. Energy fluxes are positive upward.

[Figure]

**Figure 6.** Same as Fig. 5 but for the LabS-shift response (a-d) and the NordS-shift response (e-h), respectively.

[Figure]

**Figure 7.** a) LabS-shift and b) NordS-shift response in winter (DJF) mean surface air temperature (SAT). Stippling denotes values significant at the 5% level based on t-test statistics.

[Figure]

**Figure 8.** LabS-shift and NordS-shift response in winter (DJF) mean CAM4 heat budget components as given in Eq. 2 at the lowest terrain-following model level: temperature tendencies associated with a) and d) heat transport resolved within the CAM4 dynamical core ($HT_{\text{dyn-core}\text{res}}$); b) and e) heat transport due to CAM4 parameterizations ($HT_{\text{par}}$); c) and f) diabatic processes ($\frac{J}{c_p}$).

[Figure]

**Figure 9.** Winter (DJF) mean vertical (shaded) and horizontal (vectors) wind velocities at lowest terrain-following model level for a) EEMlowRes1, b) LabS-shift response, and c) NordS-shift response. Positive (negative) vertical wind velocities denote downward (upward) motion.

[Figure]

**Figure 10.** LabS-shift and NordS-shift response in winter (DJF) mean moisture budget: a) and c) denotes precipitation minus evaporation (*P - E*); b) and d) shows the vertically-integrated moisture fluxes (vectors) and their convergence (-div(**Q**), shaded), respectively. Stippling in a) and c) indicates *P - E* changes significant at the 5% level based on t-test statistics.

[Figure]

**Figure 11.** Annual cycle of sea ice concentration (SIC, blue shading), net surface energy flux (Qnet, green lines) and surface air temperature (SAT, red lines) anomalies: a) response to LabS-shift for the LabS domain, b) response to NordS-shift for the NordS domain, c) EEM-PI$_{diff}$ response for the LabS domain, d) EEM-PI$_{diff}$ response for the NordS domain, and e) Greenland mean SAT response to LabS-shift (dotted), NordS-shift (dashed), and EEM-PI$_{diff}$ (solid). The LabS domain is designated as all oceanic grid points within the solid box in Fig. 1a and the NordS domain is the equivalent in Fig. 1b. Note that all annual cycles are calculated as spatial averages including area weighting, e.g., Greenland mean SAT in e) refers to the area-averaged SAT of whole Greenland.

[Figure]

**Figure 12.** a) LabS-shift, b) NordS-shift, and c) EEM-PI$_{\text{diff}}$ response in winter (DJF) mean temperature shown as longitude–pressure cross section along the $76°$N latitude (i.e., the latitude of pNEEM). Stippling in a) and b) denotes SAT changes significant at the 5% level based on t-test statistics.

**Table 1.** List of the core CCSM4 model  experiments using the  atmosphere-land-only setup and the 0.9°×1.25° horizontal resolution. Present-day levels are denoted as pd  and Eemian (125 ka) as eem, respectively. The orbital parameters are calculated according to Berger (1978). SST and sea ice fields are output of respective fully-coupled CCSM3 simulations described in Sect. 2.1. GHG concentrations are fixed at the attributed level and correspond to Varma et al. (2015). For all simulations, solar forcing, vegetation and ice sheets are held constant at the pre-industrial level.

| Simulation | Orbital parameters | SST/ sea ice | $CO_2$ [ppm] | $CH_4$ [ppb] | $N_2O$ [ppb] |
|---|---|---|---|---|---|
| **Pre-industrial** | | | | | |
| $PI_{lowRes1}$ | pd |  $PI_{lowRes}$ (3°) | 280 | 760 | 270 |
| $PI_{highRes2}$ | pd |  $PI_{highRes}$ (1°) | 280 | 760 | 270 |
| **Eemian** | | | | | |
| $EEM_{lowRes1}$ | eem |  $EEM_{lowRes}$ (3°) | 272 | 622 | 259 |
| $EEM_{highRes2}$ | eem |  $EEM_{highRes}$ (1°) | 272 | 622 | 259 |
| $EEM_{LabS}$ | eem | LabS-shift | 272 | 622 | 259 |
| $EEM_{NordS}$ | eem | Nord-shift | 272 | 622 | 259 |

**Table 2.** Definitions of climate anomalies  used throughout the manuscript. Please refer to Sects. 2.1 and 2.2 for details on individual simulations.

| Abbreviation | Calculation | Description |
|---|---|---|
| $EEM\text{-}PI_{lowRes}$ | $EEM_{lowRes} - PI_{lowRes}$ | Eemian minus pre-industrial climate anomaly based on simulations  with the low resolution (3°) CCSM3 |
| $EEM\text{-}PI_{highRes}$ | $EEM_{highRes} - PI_{highRes}$ | Eemian minus pre-industrial climate anomaly based on simulations  with the high resolution (1°) CCSM3 |
| $EEM\text{-}PI_1$ | $EEM_1 - PI_1$ | Eemian minus pre-industrial climate anomaly based on CCSM4 simulations prescribing SSTs and sea ice from the the lowRes (3°) CCSM3 |
| $EEM\text{-}PI_2$ | $EEM_2 - PI_2$ | Eemian minus pre-industrial climate anomaly based on CCSM4 simulations prescribing SSTs and sea ice from the the highRes (1°) CCSM3 |
| $EEM\text{-}PI_{diff}$ | $EEM\text{-}PI_{highRes2} - EEM\text{-}PI_{lowRes1}$ $= (EEM_{highRes2} - PI_{highRes2}) - (EEM_{lowRes1} - PI_{lowRes1})$ | Difference in Eemian minus pre-industrial climate anomaly due to different (highRes vs. lowRes) SSTs and sea ice |
| LabS-shift | $EEM_{LabS} - EEM_{lowRes1}$ | Climate anomaly due to idealized Labrador Sea shift in CCSM4 |
| NordS-shift | $EEM_{NordS} - EEM_{lowRes1}$ | Climate anomaly due to idealized Nordic Seas shift in CCSM4 |

**Table 3.** Surface air temperature (SAT) anomalies averaged above the Labrador Sea (LabS), Greenland and the Nordic Seas (NordS)  for all CCSM4 sensitivity experiments compared to the respective control experiment (e.g., $EEM_{LabS} = EEM_{LabS} - EEM_1$). Please refer to Sects. 2.2 and 5.4 for details about the simulations. Bold values indicate anomalies significant at the 5% level based on t-test statistics.

| Simulation | LabS ΔSAT [°C] | | Greenland ΔSAT [°C] | | NordS ΔSAT [°C] | |
|---|---|---|---|---|---|---|
| | DJF | annual | DJF | annual | DJF | annual |
| $EEM_{LabS}$ | **6.0** | **3.6** | 0.7 | **0.4** | | |
| $EEM_{\,LabS2}$ | **5.4** | **2.8** | **0.9** | **0.5** | | |
| $EEM_{LabS}$ ICE | **5.3** | **2.9** | 0.7 | **0.5** | | |
| $EEM_{\,LabS2}$ ICE | **5.7** | **2.3** | 0.5 | **0.4** | | |
| $EEM_{NordS}$ | | | **3.8** | **2.1** | **4.6** | **3.1** |
| $EEM_{\,NordS2}$ | | | **3.0** | **2.0** | **3.2** | **2.3** |
| $EEM_{NordS}$ ICE | | | **2.2** | **0.9** | **3.8** | **2.0** |
| $EEM_{\,NordS2}$ ICE | | | **1.1** | **0.6** | **2.3** | **1.2** |
| $EEM\text{-}PI_{diff}$ | **2.9** | **1.8** | **2.8** | **1.5** | **4.4** | **3.3** |

**Table 4.** Surface air temperature (SAT) and accumulation (P-E) anomalies averaged above central Greenland  for all CCSM4 sensitivity experiments compared to the respective control experiment (e.g., $EEM_{LabS} = EEM_{LabS} - EEM_1$). Please refer to Sects. 2.2 and 5.4 for details about the simulations. Note that Central Greenland is defined as 70–77°N, 35–45°W covering the summit area that includes the pNEEM, NGRIP and GRIP ice core sites. Bold values indicate anomalies significant at the 5% level based on t-test statistics.

| Simulation | Central Greenland annual ΔSAT [°C] | Central Greenland annual Δ(P-E) [%] |
|---|---|---|
| $EEM_{LabS}$ | 0.1 | 3 |
| $EEM_{\,LabS2}$ | 0.2 | 2 |
| $EEM_{LabS}$ ICE | 0.2 | 3 |
| $EEM_{\,LabS2}$ ICE | 0.3 | 5 |
| $EEM_{NordS}$ | **2.3** | **12** |
| $EEM_{\,NordS2}$ | **2.3** | **10** |
| $EEM_{NordS}$ ICE | **0.8** | 2 |
| $EEM_{\,NordS2}$ ICE | **0.6** | 1 |
| $EEM\text{-}PI_{diff}$ | **1.6** | 5 |